# ADAPTIVE DECENTRALIZED FEDERATED LEARNING FOR ROBUST OPTIMIZATION

## ABSTRACT

In decentralized federated learning (DFL), the presence of abnormal clients, often caused by noisy or poisoned data, can significantly disrupt the learning process and degrade the overall robustness of the model. Previous methods on this issue often require a sufficiently large number of normal neighboring clients or prior knowledge of reliable clients, which reduces the practical applicability of DFL. To address these limitations, we develop here a novel adaptive DFL (aDFL) approach for robust estimation. The key idea is to adaptively adjust the learning rates of clients. By assigning smaller rates to suspicious clients and larger rates to normal clients, aDFL mitigates the negative impact of abnormal clients on the global model in a fully adaptive way. Our theory does not put any stringent conditions on neighboring nodes and requires no prior knowledge. A rigorous convergence analysis is provided to guarantee the oracle property of aDFL. Extensive numerical experiments demonstrate the superior performance of the aDFL method.

## 1 INTRODUCTION

**Motivation.** Decentralized federated learning (DFL) is an effective solution to handle large-scale datasets by distributing computation across multiple clients (Beltrán et al., 2023) in a decentralized way. Different from centralized federated learning (CFL), DFL eliminates the need for a central server and enables clients to collaborate in a peer-to-peer manner. The decentralized structure improves scalability, robustness, and privacy preservation, making it an appealing solution for large-scale data analysis (Li et al., 2021). Nevertheless, DFL also faces a serious challenge. DFL systems may involve unreliable or undesirable clients that degrade the overall performance of the learning process. One typical example is Byzantine failure, which refers to transmitting incorrect information to the whole network (Blanchard et al., 2017; Yin et al., 2018; Tu et al., 2021b). In addition, some clients may hold low-quality or even corrupted data, contain information irrelevant to the target task, exhibit severe distribution shifts, or suffer from unstable communication and computation. These issues could negatively impact the training process by introducing noise or bias into the model updates, which is especially problematic for DFL, as the lack of a central server makes it more difficult to detect and correct abnormal behaviors (Wu et al., 2023b; Zhang & Wang, 2024).

**Challenge**. To address the abnormal clients in DFL, various robust learning methods have been proposed; see Section 2 for a more detailed discussion. Nevertheless, existing methods suffer from at least one of the following two limitations. First, many existing methods require for each client a sufficiently large number of normal neighborhood clients. Otherwise, a robust and reliable summary (e.g., median) of neighborhood gradients/estimators cannot be obtained (Yang & Bajwa, 2019; Su & Vaidya, 2020; Fang et al., 2022). Second, many other methods require prior knowledge of reliable clients, which is an even more stringent condition for most practical applications (Peng et al., 2023; Wu et al., 2023b; Zhang & Wang, 2024).

**Contributions**. To solve those aforementioned problems, we propose here an adaptive decentralized federated learning (aDFL) approach for robust estimation. The key idea of aDFL is to dynamically adjust the learning rates of individual clients based on their behavior in the DFL process. Intuitively, those clients with suspicious behavior in their estimated gradients should be given smaller learning rates. In contrast, larger learning rates should be given to those clients who behave more normally in their gradients. The consequence is that the negative effect of those abnormal clients can be well controlled and minimized in a fully automatic way. Compared with existing methods, our

theory does not put any stringent conditions on neighboring clients and requires no prior knowledge. In summary, we make the following contributions in this work. **Methodologically,** we develop here a novel aDFL approach for robust learning. This method adapts to diverse DFL settings and data scenarios. Additionally, it does not rely on the assumption of homogeneous data distribution across clients, which overcomes a key limitation of many existing approaches (Yang & Bajwa, 2019; Fang et al., 2022; Peng et al., 2023; Qian et al., 2024) and improves the applicability to real-world heterogeneous scenarios. **Theoretically,** the convergence rate of the aDFL algorithm is rigorously analyzed. Our results show that, aDFL can achieve the oracle property (i.e., the same asymptotic efficiency as the estimator computed by normal clients only) under appropriate regularity conditions.

## 2 RELATED WORK

**Decentralized federated learning.** The literature about DFL can be classified into two categories. The first one is *decentralized consensus optimization methods*, which enforce consensus among neighboring estimators to ensure global consensus. These methods include proximal gradient (Wu et al., 2017; Lü et al., 2020), ADMM (Li et al., 2019b; Liu et al., 2022b) and gradient tracking (Xu et al., 2017; Tang et al., 2018; Li et al., 2019d; Song et al., 2022a;b). The second one is *decentralized gradient descent methods*, which mainly apply (stochastic) gradient descent after obtaining averaged neighborhood estimators. Typical works include Jiang et al. (2017), Sirb & Ye (2018), Li et al. (2019c), Xu et al. (2021), Liu et al. (2022a), and Wu et al. (2023a). More discussions can be found in Beltrán et al. (2023). Note that the proposed aDFL method falls under the second category but can be extended to the first category without difficulty; see Appendix A.4 for detailed discussion.

**Robust centralized learning.** It focuses on minimizing the impact of abnormal participants in a centralized distributed machine learning system (Blanchard et al., 2017; Chen et al., 2017; Yin et al., 2018). The literature in this regard can be classified into two approaches. The first approach aims to mitigate the impact of abnormal clients by designing robust aggregation rules, which are closely related to the robust estimation techniques in statistics (Shi et al., 2022). The most typical technique is to replace the sample mean of the local gradients/estimators by its robust counterpart, such as the trimmed mean (Yin et al., 2018), median (Chen et al., 2017; Yin et al., 2019), and quantile (Tu et al., 2021a). Another approach tries to first identify the abnormal clients by analyzing and detecting abnormal patterns, and then exclude them from the subsequent updating process. The methods include discrepancy comparison (Blanchard et al., 2017), reputation scores (Xia et al., 2019; Xie et al., 2019), and anomaly detection (Li et al., 2019a). Notably, this line of work is also closely related to outlier detection in statistical domain, where various methods have been developed to detect abnormal samples (Filzmoser et al., 2008; Zimek et al., 2012; Ro et al., 2015). One representative work in the federated learning regime is Qian et al. (2024), which leverages false discovery rate (FDR) control and sample splitting techniques to identify abnormal clients.

**Robust decentralized learning**. In DFL, the absence of a central server makes it significantly more difficult to identify and mitigate the influence of abnormal clients. As a result, most existing work on robust DFL extends techniques originally developed for CFL, but often at the cost of stronger assumptions, such as requiring enough trustworthy neighbors. A common line of work includes various robust aggregation rules, such as clipping and trimming (Yang & Bajwa, 2019; He et al., 2022; Su & Vaidya, 2020). Various variance reduction techniques are also used, including the TV-norm regularization and related techniques (Peng et al., 2021; 2023; Hu et al., 2023). Another widely used idea is to evaluate the consistency or credibility of each client by comparing its model with those of its neighbors, and then down-weight or exclude those that behave abnormally. This leads to techniques such as performance-based filtering (Guo et al., 2021; Elkordy et al., 2022) and credibility-aware aggregation (Hou et al., 2022). These methods rely on local information exchange and are tailored to the decentralized setting where global oversight is unavailable.

## 3 STANDARD DECENTRALIZED FEDERATED LEARNING

### 3.1 PROBLEM DESCRIPTION

We begin by introducing the model setup and notation. Due to page limitations, a complete list of notation is provided in Appendix B.1. Assume a total of $N$ instances denoted as $(X_i, Y_i)$ for

$1 \leq i \leq N$. Here, $X_i = (X_{ij}) \in \mathbb{R}^p$ is the feature vector and $Y_i \in \mathbb{R}$ is the associated univariate response. We next consider a total of $M$ clients indexed by $\mathcal{M} = \{1, 2, \dots, M\}$. Let $\mathcal{S}_F = \{1, 2, \dots, N\}$ represent the whole sample set, and let $\mathcal{S}_m$ denote the sample collected by the $m$th client. We then should have $\mathcal{S}_F = \bigcup_m \mathcal{S}_m$ and $\mathcal{S}_{m_1} \bigcap \mathcal{S}_{m_2} = \emptyset$ for any $m_1 \neq m_2$. For simplicity, we assume that $|\mathcal{S}_m| = N/M = n$ for every $1 \leq m \leq M$. In federated learning, data across different clients often exhibit considerable heterogeneity, which may arise from varied data collection environments. Despite the heterogeneity here, we assume that all normal clients share a common underlying regression relationship. To be more precise, denote the joint distribution of $(x, y) \in \mathcal{X} \times \mathcal{Y}$ by $\mathcal{P}(x, y)$. Then we allow the marginal distributions $\mathcal{P}(x)$ and $\mathcal{P}(y)$ to be heterogeneous but basically require the conditional distribution $\mathcal{P}(y \mid x)$ must be the same across different clients. The common parameter of interest is denoted as $\theta_0 \in \mathbb{R}^p$. Next, let $\ell(x, y; \theta)$ be a loss function with parameter $\theta \in \mathbb{R}^p$. Define a global loss function as $\mathcal{L}(\theta) = N^{-1} \sum_{i=1}^{N} \ell(X_i, Y_i; \theta)$. It can be decomposed as $\mathcal{L}(\theta) = M^{-1} \sum_{m=1}^{M} \mathcal{L}_m(\theta)$, where $\mathcal{L}_m(\theta) = n^{-1} \sum_{i \in \mathcal{S}_m} \ell(X_i, Y_i; \theta)$ is the loss function defined on the $m$th client. Next define $\widehat{\theta} = \arg\min_\theta \mathcal{L}(\theta)$ as the whole sample estimator and $\widehat{\theta}_m = \arg\min_\theta \mathcal{L}_m(\theta)$ as the local estimator computed on the $m$th client.

In this work, we consider the *data-contaminated adversary* setting, where all of the clients are assumed to follow the learning protocol but the local data on the abnormal client may be corrupted (Biggio et al., 2012; Fang et al., 2020; Jagielski et al., 2018; Li et al., 2016). Specifically, define for each client a binary variable $a_m \in \{1, 0\}$ to indicate whether the $m$th client is abnormal or not. Collect the indices of abnormal clients by $\mathcal{A} = \{m : a_m = 1\}$. Let $\varrho = |\mathcal{A}|/M \in [0, 1/2)$ be the fraction of abnormal clients. Accordingly, we assume that as $n \to \infty$,

$$\begin{cases} \sqrt{n}(\widehat{\theta}_m - \theta_0) \to_d N(0, \Sigma_m) & \text{if } m \notin \mathcal{A}, \\ \sqrt{n}(\widehat{\theta}_m - \theta_m) \to_d N(0, \Sigma_m) \text{ with } \theta_m \neq \theta_0 & \text{if } m \in \mathcal{A}. \end{cases}$$

for some positive definite matrix $\Sigma_m \in \mathbb{R}^{p \times p}$. Since $\theta_m \neq \theta_0$ for any $m \in \mathcal{A}$, including those abnormal clients $\mathcal{A}$ in DFL without effective control should cause seriously biased results.

## 3.2 THE DFL FRAMEWORK

We start with a standard DFL framework involving two key steps (Yuan et al., 2016; Wu et al., 2023a). First, each client aggregates information from its neighbors to derive a neighborhood-averaged parameter estimator. Next, it updates this estimator by the method of gradient descent based on the data placed on the local client. Specifically, assume $M$ clients are connected through a communication network represented by an adjacency matrix $A = (a_{m_1 m_2}) \in \mathbb{R}^{M \times M}$. Here, $a_{m_1 m_2} = 1$ if client $m_1$ can receive information from client $m_2$, and $a_{m_1 m_2} = 0$ otherwise. Define in-degree $d_{m_1} = \sum_{m_2} a_{m_1 m_2}$. We assume that $d_{m_1} > 0$ for every $1 \leq m_1 \leq M$. Define the weighting matrix $W = (w_{m_1 m_2}) \in \mathbb{R}^{M \times M}$ with $w_{m_1 m_2} = a_{m_1 m_2}/d_{m_1}$. Let $\widehat{\theta}^{(t,m)}$ be the estimator obtained on the $m$th client at the $t$th iteration. Then, the update formula at the $(t+1)$th iteration is:

$$\widetilde{\theta}^{(t,m)} = \sum_{k=1}^{M} w_{mk} \widehat{\theta}^{(t,k)}; \qquad \widehat{\theta}^{(t+1,m)} = \widetilde{\theta}^{(t,m)} - \alpha \dot{\mathcal{L}}_m \big( \widetilde{\theta}^{(t,m)} \big). \tag{3.1}$$

Here $\dot{\mathcal{L}}_m(\theta) \in \mathbb{R}^p$ denotes the first order derivative of $\mathcal{L}_m(\cdot)$ with respect to $\theta$, and $\alpha \in \mathbb{R}^+$ denotes the learning rate. Under appropriate regularity assumptions and assuming $\varrho = 0$, Wu et al. (2023a) showed that, with a sufficiently small $\alpha$ and a relatively balanced network structure $W$, $\widehat{\theta}^{(t,m)}$ should converge numerically to an asymptotically efficient estimator of $\theta_0$. However, it is unclear what would happen if some of the clients are abnormal (i.e., $\varrho > 0$). We are thus inspired to study the theoretical properties of $\widehat{\theta}^{(t,m)}$ under the assumption with $\varrho > 0$.

To this end, define $\text{SE}^2(W) = M^{-1} \| W^\top \mathbf{1}_M - \mathbf{1}_M \|^2$ which measures the balance of network structures. In the most ideal situation with doubly stochastic $W$ in the sense that $\mathbf{1}_M^\top W = \mathbf{1}_M^\top$ (Lian et al., 2018; Li et al., 2019d), we have $\text{SE}(W) = 0$. Then we have the following regularity conditions.

**Assumption 1** (Parameter space)**.** *Assume the parameter space $\Theta$ is a compact and convex subset of $\mathbb{R}^p$. Let $\text{int}(\Theta)$ be the set of interior points of $\Theta$. Assume $\theta_m \in \text{int}(\Theta)$ for $m \in \mathcal{A} \cup \{0\}$. Moreover, define $r = \sup_{\theta \in \Theta} \max_m \|\theta - \theta_m\| > 0$ as a rough measure for the radius of $\Theta$.*

**Assumption 2** (Covariates distribution). *Assume $(X_i, Y_i)$ from the $m$th client, i.e., $i \in \mathcal{S}_m$ are independently and identically generated from a probability distribution $\mathcal{P}_m$.*

**Assumption 3** (Local strong convexity). *Define $\Omega_m(\theta) = \mathbb{E}_m[\ddot{\ell}(X_i, Y_i; \theta)]$ for $m \in \mathcal{M}$, where $\ddot{\ell}(x, y; \theta) \in \mathbb{R}^{p \times p}$ denotes the second order derivative of $\ell(x, y; \theta)$ with respect to $\theta$. Assume that for $m \in \mathcal{M}$, we have $\lambda_{\min}\{\Omega_m(\theta_m)\} \geq \lambda_{\min}$ for some positive constant $\lambda_{\min}$, and $\min_m \inf_{\theta \in \Theta} \lambda_{\min}\{\ddot{\mathcal{L}}_m(\theta)\} \geq 0$.*

**Assumption 4** (Smoothness). *Assume that there exists some constant $C_{\max} > 0$ such that for $m \in \mathcal{M}$, $\sup_{\theta \in \Theta} \mathbb{E}_m\{\|\dot{\ell}(X_i, Y_i; \theta) - \mathbb{E}_m\{\dot{\ell}(X_i, Y_i; \theta)\}\|_2^8\} \leq C_{\max}^8$, and $\sup_{\theta \in \Theta} \mathbb{E}_m\{\|\ddot{\ell}(X_i, Y_i; \theta) - \Omega_m(\theta)\|_2^8\} \leq C_{\max}^8$. Moreover, for any $(X_i, Y_i) \in \mathcal{S}_m$, $\dot{\ell}(X_i, Y_i; \theta)$ and $\ddot{\ell}(X_i, Y_i; \theta)$ are Lipschitz continuous in the sense that for any $\theta', \theta'' \in \Theta$, the following inequality holds*

$$\|\dot{\ell}(X_i, Y_i; \theta') - \dot{\ell}(X_i, Y_i; \theta'')\| \leq L(X_i, Y_i)\|\theta' - \theta''\|,$$
$$\|\ddot{\ell}(X_i, Y_i; \theta') - \ddot{\ell}(X_i, Y_i; \theta'')\| \leq L(X_i, Y_i)\|\theta' - \theta''\|$$

*for some positive function $L(X_i, Y_i)$ and constant $L_{\max}$ such that $\mathbb{E}_m\{L^8(X_i, Y_i)\} \leq L_{\max}^8$.*

**Assumption 5** (Network structure). *There exists some constant $\rho \in (0, 1)$ such that $\|W^\top(I_M - M^{-1}\mathbf{1}_M\mathbf{1}_M^\top)W\| + \mathrm{SE}(W) \leq \rho$.*

**Assumption 6** (Non-vanishing bias). *Define $\flat_m = \mathbb{E}_m\{\dot{\ell}(X_i, Y_i; \theta_0)\}$, Assume $\min_{m \in \mathcal{A}} \|\flat_m\| \geq \flat_{\min}$ for some constant $\flat_{\min} > 0$.*

**Remark 1.** *Assumption 1 defines the parameter space for $\theta_m$ with $m \in \mathcal{A} \cup \{0\}$. Similar conditions are also used in Zhang et al. (2013) and Jordan et al. (2019). Assumption 2 addresses the distribution of the data $\{(X_i, Y_i) : i \in \mathcal{S}_k\}$, allowing the data distributions to vary across different clients. This relaxes the homogeneous data condition commonly assumed in existing approaches (Fang et al., 2022; Qian et al., 2024). Assumption 3 requires only local strong convexity of the loss functions rather than the global strong convexity typically assumed in existing literature (Karimireddy et al., 2021; Kuwaranancharoen & Sundaram, 2023; Zhang & Wang, 2024). This makes our theoretical results applicable to a broader class of loss functions. For completeness, we also provide theoretical results for our proposed method under the standard global strong convexity assumption; see Appendix A.2 for details. Assumption 4 requires the local loss functions to be sufficiently smooth, which is a classical regularity condition in convex optimization (Jordan et al., 2019) and federated learning (Zhang & Wang, 2024). Assumption 5 is a condition about the network structure. This assumption is weaker than the commonly assumed doubly stochastic assumption in the literature (Li et al., 2019d; Song et al., 2023). Assumption 6 forces abnormal clients to be distinguishable from normal clients since $\|\flat_m\| = 0$ for any $m \notin A$.*

We start with the properties of the whole-sample estimator $\widehat{\theta}$ with $\varrho > 0$. This leads to the following Theorem 3.1 about the mean-squared error (MSE) of $\theta$.

**Theorem 3.1** (MSE of $\widehat{\theta}$). *Assume Assumptions $1 - 6$ hold. Further assume that $\varrho < \epsilon$ for some sufficiently small but fixed $\epsilon$ depending on $(L_{\max}, \lambda_{\min}, \rho)$. Then we have $\mathbb{E}\|\widehat{\theta} - \theta_0\|^2 = V(\widehat{\theta}) + \|\bar{\flat}_{\mathcal{A}}\| B(\widehat{\theta})$, where $V(\widehat{\theta}) \lesssim L_{\max}^2/[\{(1 - \varrho)\lambda_{\min}\}^2 N] + O(N^{-2})$, $NV(\widehat{\theta}) \to \mathrm{tr}\{\Omega_{\mathcal{A}}^{-1}\Sigma_{\mathcal{A}}\Omega_{\mathcal{A}}^{-1}\}$ as $N \to \infty$, and*

$$\frac{\varrho^2\|\bar{\flat}_{\mathcal{A}}\|}{L_{\max}^2} - C\Big(\frac{\varrho}{N} + \frac{1}{N^3}\Big) \leq B(\widehat{\theta}) \leq O\Big(\varrho^2\|\bar{\flat}_{\mathcal{A}}\| + \frac{\varrho}{N} + \frac{1}{N^3}\Big).$$

*Here $\bar{\flat}_{\mathcal{A}} = |\mathcal{A}|^{-1} \sum_{m \in \mathcal{A}} \flat_m$. The detailed formulas of $\Omega_{\mathcal{A}}$ and $\Sigma_{\mathcal{A}}$ are given in Appendix B.1.*

By Theorem 3.1, we know that the MSE of $\widehat{\theta}$ is mainly determined by two terms. The first term $V(\widehat{\theta})$ reflects the variance with its leading term given by $N^{-1} \mathrm{tr}\{\Omega_{\mathcal{A}}^{-1}\Sigma_{\mathcal{A}}\Omega_{\mathcal{A}}^{-1}\}$. The second term reflects the bias with its leading term of the same order as $\varrho^2\|\bar{\flat}_{\mathcal{A}}\|^2$. If $\varrho \to 0$, $\widehat{\theta}$ remains to be a consistent estimator for $\theta_0$. However, for $\widehat{\theta}$ to achieve a root-$N$ convergence rate, we need to have $\varrho^2 = o(N^{-1})$. This leads to $n/M = o(|\mathcal{A}|^{-2})$. Otherwise, $\widehat{\theta}$ may exhibit a non-negligible bias. However, this condition is not always achievable in practice. Consider for example a situation with

each client representing a local hospital. In this case, each client (e.g., a hospital) might hold a sufficiently large amount of data. Nevertheless, the total number of clients (hospitals) is typically quite limited. According to classical results on DFL, under suitable assumptions, the difference between the standard DFL estimator and $\widehat{\theta}$ is statistically ignorable; see Proposition A.1 in Appendix A.1 for details. However, Theorem 3.1 indicates that $\widehat{\theta}$ itself might be biased. Consequently, the DFL estimator is also expected to suffer from the same bias. This motivates us to develop a robust DFL algorithm, so that the negative effects due to the abnormal clients can be well controlled.

## 4 ROBUST DFL

### 4.1 WEIGHTED DECENTRALIZED FEDERATED LEARNING

By Theorem 3.1, we know that the key reason responsible for the poor performance of the standard DFL estimator is the existence of the abnormal clients (i.e., $\mathcal{A}$). Unfortunately, a standard DFL algorithm treats those abnormal clients and normal clients equally without differentiating their relative trustworthiness. One natural solution is to revise $\alpha$ slightly so that different learning rates can be used for different clients according to their trustworthiness. Intuitively, larger learning rates should be given to clients, which are more likely to have $a_m = 0$. In contrast, significantly reduced learning rates should be given to those which are more likely to have $a_m = 1$. Accordingly, the bias due to those abnormal clients in $\mathcal{A}$ can be greatly reduced.

Let $\widehat{\theta}_{\mathcal{A}}^{(t,m)}$ be an estimator obtained on the $m$th client at the $t$th iteration. We are motivated to modify the standard DFL updating formula (3.1) as:

$$\widehat{\theta}_{\mathcal{A}}^{(t+1,m)} \;=\; \widetilde{\theta}_{\mathcal{A}}^{(t,m)} - \alpha\omega_m \dot{\mathcal{L}}_{(m)}\big(\widetilde{\theta}_{\mathcal{A}}^{(t,m)}\big) \tag{4.1}$$

with $\widetilde{\theta}_{\mathcal{A}}^{(t,m)} = \sum_k w_{mk}\widehat{\theta}_{\mathcal{A}}^{(t,k)}$. Here, $\omega_m \in [0,1]$ is a data-driven weight that reflects the trustworthiness of the $m$th client. Intuitively, $\omega_m$ should be larger for trustworthy clients. Conversely, $\omega_m$ should be smaller for those suspicious clients.

Subsequently, we analyze the theoretical properties of the algorithm (4.1) with general $\omega_m$s. In particular, we are eager to understand the role played by the adaptive weights $\omega_m$s. To this end, define $\bar{\Delta}_2^2 = M^{-1}\sum_{m=1}^M \big\{\omega_m - (1 - a_m)\big\}^2$ as the mean squared distance between $\omega_m$ and the oracle weight $1 - a_m$. Write $\bar{\omega}^{\mathcal{G}} = M^{-1}\sum_{m\notin\mathcal{A}}\omega_m$, $\overline{\omega}_2^{\mathcal{A}} = (|\mathcal{A}|^{-1}\sum_{m\in\mathcal{A}}\omega_m^2)^{1/2}$ and $\bar{\flat}_2^{\mathcal{A}} = (|\mathcal{A}|^{-1}\sum_{m\in\mathcal{A}}\|\flat_m\|^2)^{1/2}$. Further denote $\widehat{\theta}_{\mathcal{A}}^{*(t)} = \{(\widehat{\theta}_{\mathcal{A}}^{(t,1)})^\top, \ldots, (\widehat{\theta}_{\mathcal{A}}^{(t,M)})^\top\}^\top \in \mathbb{R}^{Mp}$ as the stacked estimator obtained from Equation (4.1) at the $t$th iteration. For theoretical purposes, define an oracle estimator as the estimator obtained by using data from the trustworthy clients only. Denote this oracle estimator by $\widehat{\theta}_{\mathcal{A}} = arg\,min_\theta \sum_{m\notin\mathcal{A}}\mathcal{L}_m(\theta)$. Let $\widehat{\delta}_0^{\mathcal{A}} = \max_m \|\widehat{\theta}_{\mathcal{A}}^{(0,m)} - \widehat{\theta}_{\mathcal{A}}\|$. We then have the following Theorem 4.1.

**Theorem 4.1** (Convergence property of $\widehat{\theta}_{\mathcal{A}}^{*(t)}$). *Assume that Assumptions $1 - 6$ hold, Further assume that $\alpha + \mathrm{SE}(W) < \epsilon$ and the initial value $\widehat{\theta}_{\mathcal{A}}^{*(0)}$ is sufficiently close to $I^*\widehat{\theta}_{\mathcal{A}}$ in the sense that $\|\widehat{\theta}_{\mathcal{A}}^{*(0)} - I^*\widehat{\theta}_{\mathcal{A}}\| \leq \epsilon$ for some sufficiently small but fixed $\epsilon$ depending on $(L_{\max}, \lambda_{\min}, \rho)$. Then, with probability at least $1 - O\big(M/n^4 + 1/(\log n)^4\big)$, we have $M^{-1/2}\|\widehat{\theta}_{\mathcal{A}}^{*(t)} - I^*\widehat{\theta}_{\mathcal{A}}\| \lesssim \mathrm{Err}_1 + \mathrm{Err}_2 + \mathrm{Err}_3$, where*

$$\mathrm{Err}_1 = \Big(1 - \frac{\alpha\bar{\omega}^{\mathcal{G}}\lambda_{\min}}{8}\Big)^t\widehat{\delta}_0^{\mathcal{A}}, \;\; \mathrm{Err}_2 = \frac{\alpha L_{\max} + \mathrm{SE}(W)}{(1-\rho)\lambda_{\min}\bar{\omega}^{\mathcal{G}}}\Big\{\Big(\frac{\log n}{n}\Big)^{1/2}L_{\max} + \varrho^{1/2}\bar{\flat}_2^{\mathcal{A}}\Big\},$$

$$\mathrm{Err}_3 = \frac{1}{\bar{\omega}^{\mathcal{G}}\lambda_{\min}}\Big[\varrho\bar{\flat}_2^{\mathcal{A}}\overline{\omega}_2^{\mathcal{A}} + \bar{\Delta}_2\Big\{\Big(\frac{\log N}{N}\Big)^{\frac{1}{2}} + L_{\max}\|\widehat{\theta}_{\mathcal{A}} - \theta_0\|\Big\}\Big]. \tag{4.2}$$

*Assume $M = o(n^4)$ as $n \to \infty$. Then with probability tending to 1, we have $M^{-1/2}\big\|\widehat{\theta}_{\mathcal{A}}^{*(\infty)} - I^*\widehat{\theta}_{\mathcal{A}}\big\|$ upper bounded by*

$$\frac{C}{\bar{\omega}^{\mathcal{G}}}\Big[\big\{\alpha + \mathrm{SE}(W)\big\}\big(n^{-1/2} + \varrho^{1/2}\big) + \Big(\varrho\overline{\omega}_2^{\mathcal{A}} + \frac{\bar{\Delta}_2}{\sqrt{N}}\Big)\Big]. \tag{4.3}$$

Compared to the classical results on DFL (see Proposition A.1 for details), the main difference of Theorem 4.1 is the inclusion of an additional statistical error term $\mathrm{Err}_3$. If oracle weights $(1 - a_m)$s

are employed, we then have $\bar{\omega}^{\mathcal{G}} = 1 - \varrho$ and $\bar{\Delta}_2 = \overline{\omega}_2^{\mathcal{A}} \equiv 0$. Accordingly, the influence of abnormal clients on $\widehat{\theta}_{\mathcal{A}}^{*(t)}$ can be eliminated completely, as long as the learning rate $\alpha$ is sufficiently small and the network structure $W$ is sufficiently balanced.

For $\widehat{\theta}_{\mathcal{A}}$ to achieve the oracle property, the right-hand side of Equation (4.3) should be of an $o_p(1/\sqrt{N})$ order. This conclusion holds if the following three conditions can be satisfied. They are, respectively, (1) $\{\alpha + \mathrm{SE}(W)\}\{1/\sqrt{n} + \varrho^{1/2}\}/\bar{\omega}^{\mathcal{G}} = o(1/\sqrt{N})$, (2) $\overline{\omega}_2^{\mathcal{A}}/\bar{\omega}^{\mathcal{G}} = o_p\big(1/(\varrho\sqrt{N})\big)$, and (3) $\bar{\Delta}_2/\bar{\omega}^{\mathcal{G}} = o_p(1)$. The first condition can be satisfied by setting a reasonable $\bar{\omega}^{\mathcal{G}}$, and a sufficiently small $\alpha$ and $\mathrm{SE}(W)$. Both conditions (2) and (3) require $\omega_m$ to approximate the oracle weights $(1 - a_m)$ closely. However, since the status of the clients is unknown in advance, we need to develop an effective estimator for $\omega_m$ so that both conditions (2) and (3) can be practically satisfied.

## 4.2 Adaptive Decentralized Federated Learning

To this end, an effective measure for the trustworthiness of a client is necessarily needed. Note that a trustworthy client should have a small gradient norm at a reasonably accurate parameter estimator. In contrast, an abnormal client tends to exhibit a larger gradient norm. Thus, the size of the gradient norm might serve as a natural indicator of trustworthiness. Based on this idea, we develop below a two-stage algorithm.

STAGE 1. We start with assuming for each client $m$ an initial estimator, denoted by $\widehat{\theta}_{\text{init}}^{(m)}$, which may not be statistically efficient but must be consistent. For example, one might use the standard DFL estimator as described in Section 3.2 to serve this purpose, if condition $\varrho \to 0$ can be well satisfied.

STAGE 2. Once the initial estimator $\widehat{\theta}_{\text{init}}^{(m)}$ is obtained, the adaptive weight for the $m$th client can be computed as

$$\widehat{\omega}_m = \pi\big\{\lambda_n\|\dot{\mathcal{L}}_{(m)}(\widehat{\theta}_{\text{init}}^{(m)})\|\big\}, \tag{4.4}$$

where $\pi(\cdot) \in [0,1]$ is an appropriately selected and monotonously decreasing mapping function. For example, we use $\pi(x) = \exp(-x)$ in this work. Moreover, $\lambda_n$ is a positive tuning parameter, which controls the gradient scale. It is important to note that the selection of $\lambda_n$ plays a critical role in this algorithm. Specifically, $\lambda_n\|\dot{\mathcal{L}}_{(m)}(\widehat{\theta}_{\text{init}}^{(m)})\|$ should not be too low. Otherwise, $\widehat{\omega}_m$ cannot shrink to 0 quickly for those abnormal clients. Conversely, this product should not be too large either. Otherwise, $\widehat{\omega}_m$ might not give sufficient trust to those trustworthy clients. Subsequently, the updating step in Equation (4.1) can be executed by replacing $\omega_m$ with $\widehat{\omega}_m$ in (4.4). This leads to a practically feasible aDFL estimator $\widehat{\theta}_{\text{aDFL}}^{(t,m)}$ for the $m$th client at the $t$th iteration with $\widehat{\theta}_{\text{aDFL}}^{(0,m)} = \widehat{\theta}_{\text{init}}^{(m)}$. The pseudo code for the aDFL algorithm is described below in Algorithm 1.

---

Algorithm 1: Adaptive Decentralized Federated Learning

---

**Require:** Network $W$, learning rate $\alpha$, max iteration $T$.
**Ensure:** aDFL estimator $\{\widehat{\theta}_{\text{aDFL}}^{(T,m)}\}_{m=1}^{M}$.
  1: Compute initial estimators $\{\widehat{\theta}_{\text{init}}^{(m)}\}_{m=1}^{M}$, and set $\widehat{\theta}_{\text{aDFL}}^{(0,m)} = \widehat{\theta}_{\text{init}}^{(m)}$ for $1 \le m \le M$.
  2: **for** $0 \le t \le T - 1$ **do**
  3:   **for** $1 \le m \le M$ (distributedly) **do**
  4:     Compute the neighborhood-averaged estimator $\widetilde{\theta}_{\text{aDFL}}^{(t,m)} = \sum_k w_{mk}\widehat{\theta}_{\text{aDFL}}^{(t,k)}$.
  5:     Compute $\widehat{\theta}_{\text{aDFL}}^{(t+1,m)} = \widetilde{\theta}_{\text{aDFL}}^{(t,m)} - \alpha\widehat{\omega}_m\dot{\mathcal{L}}_{(m)}(\widetilde{\theta}_{\text{aDFL}}^{(t,m)})$, where $\widehat{\omega}_m$ is given by (4.4).
  6:   **end for**
  7: **end for**

---

We next study the theoretical properties of the proposed aDFL estimator $\widehat{\theta}_{\text{aDFL}}^{(t,m)}$. Denote the stacked aDFL estimator at iteration $t$ as $\widehat{\theta}_{\text{aDFL}}^{*(t)} = \big\{(\widehat{\theta}_{\text{aDFL}}^{(t,1)})^\top, \ldots, (\widehat{\theta}_{\text{aDFL}}^{(t,M)})^\top\big\}^\top \in \mathbb{R}^{Mp}$. Write the corresponding estimators of $\overline{\omega}_2^{\mathcal{A}}$, $\bar{\Delta}_2$, and $\bar{\omega}^{\mathcal{G}}$ based on Equation (4.4) as $\hat{\overline{\omega}}_2^{\mathcal{A}}$, $\hat{\bar{\Delta}}_2$ and $\hat{\bar{\omega}}^{\mathcal{G}}$, respectively. We then have the following Theorem 4.2.

**Theorem 4.2** (Convergence rate of the aDFL). *Assume that Assumptions $1 - 6$ hold. Let $\pi(x) = \exp(-x)$, and set the initial value $\widehat{\theta}_r^{(m)}$ as the standard DFL estimator. Assume that $\log N \lesssim \lambda_n \lesssim$*

$\sqrt{n}M^{-1/8}$. *Then, with probability at least* $1 - O(M/n^4 + 1/\log N)$, *we have: (1)* $\hat{\hat{\omega}}_2^{\mathcal{A}} \lesssim 1/\sqrt{N}$, *(2)* $\hat{\hat{\Delta}}_2^2 \lesssim \varrho/\sqrt{N} + \lambda_n(1/\sqrt{n} + \|\widehat{\theta} - \theta_0\|)$, *and (3)* $1/\hat{\hat{\omega}}^{\mathcal{G}} \lesssim \exp(c\lambda_n\|\widehat{\theta} - \theta_0\|)$. *Further assume* $M \to \infty$ *with* $M = o(n^4)$, $\varrho = o(1)$ *and* $\alpha + \mathrm{SE}(W)$ *is sufficiently small as* $n \to \infty$. *Then, with probability tending to 1, we have* $M^{-1/2}\|\widehat{\theta}_{\mathrm{aDFL}}^{*(\infty)} - I^*\widehat{\theta}_{\mathcal{A}}\|$ *upper bounded by*

$$C \exp\left(c\lambda_n\|\widehat{\theta} - \theta_0\|\right)\left\{\frac{\lambda_n\|\widehat{\theta} - \theta_0\|}{\sqrt{N}} + o\left(\frac{1}{\sqrt{N}}\right)\right\}. \tag{4.5}$$

From Theorem 4.2, the statistical error introduced by abnormal clients (4.5) can be further reduced to be of the order $o_p(1/\sqrt{N})$, if we can further assume that $\lambda_n\|\widehat{\theta} - \theta_0\| = o_p(1)$. Here, recall that $\widehat{\theta}$ denotes the whole-sample estimator. This result implies that the aDFL estimator achieves the same asymptotic efficiency as $\widehat{\theta}_{\mathcal{A}}$, as long as a suitable tuning parameter $\lambda_n$ can be used.

The validity of Theorem 4.2 relies on the assumption that an initial estimator of a reasonable quantity be provided. It can be easily satisfied as long as there exists a statistically consistent (but not necessarily efficient) initial estimator. As shown by our Theorem 3.1 and Proposition A.1, a standard DFL estimator can serve the purpose with $\varrho = o(1)$. In practice, one might also consider other decentralized robust estimators (Karimireddy et al., 2021; Fang et al., 2022; Zhang & Wang, 2024) as initial estimators $\widehat{\theta}_{\mathrm{init}}^{(m)}$ with $\varrho \in [0, 1/2)$ under appropriate regularity conditions. We summarize this result in the following Corollary 4.1.

**Corollary 4.1** (General initial estimator). *Assume that Assumptions 1 − 6, and A.1 hold. Let* $\pi(x) = \exp(-x)$. *Assume* $M \to \infty$ *with* $M = o(n^4)$ *and* $\alpha + \mathrm{SE}(W)$ *is sufficiently small as* $n \to \infty$. *Further assume* $\log N \lesssim \lambda_n \lesssim \sqrt{n}M^{-1/8}$, *Then, with probability tending to 1, we have* $M^{-1/2}\|\widehat{\theta}_{\mathrm{aDFL}}^{*(\infty)} - I^*\widehat{\theta}_{\mathcal{A}}\|$ *upper bounded by* $C \exp\left(c\lambda_n\|\bar{\theta}_{\mathrm{init}} - \theta_0\|\right)\left\{\lambda_n\|\bar{\theta}_{\mathrm{init}} - \theta_0\|/\sqrt{N} + o\left(1/\sqrt{N}\right)\right\}$ *with* $\bar{\theta}_{\mathrm{init}} = M^{-1}\sum_{m=1}^{M}\widehat{\theta}_{\mathrm{init}}^{(m)}$.

We find that aDFL estimator should have the oracle property as long as $\lambda_n\|\bar{\theta}_{\mathrm{init}} - \theta_0\| = o_p(1)$.

**Remark 2.** *The numerical convergence speed and statistical efficiency of Algorithm 1 can be improved in two ways. First, Theorem 4.2 indicates a convergence rate of* $1 - O(\hat{\hat{\omega}}^{\mathcal{G}})$. *Thus, after computing* $\widehat{\omega}_m$, *each client can obtain* $\omega_{\max} = \max_m \widehat{\omega}_m$ *by a DFL algorithm and then update* $\widehat{\omega}_m \leftarrow \widehat{\omega}_m/\omega_{\max}$ *so that* $\hat{\hat{\omega}}^{\mathcal{G}}$ *can be increased. Second, both Theorem 4.2 and Corollary 4.1 reveal that the error bound depends on* $\|\bar{\theta}_{\mathrm{init}} - \theta_0\|$. *Then the aDFL estimator can be used as a new initial estimator for Algorithm 1 repeatedly. The multi-stage aDFL algorithm is provided in Algorithm A.1.*

## 5 EXPERIMENTS

In this section, we examine the finite-sample performance of the proposed aDFL method. We compare our aDFL algorithm with the following alternatives: DFL (Wu et al., 2023a), BRIDGE-M, BRIDGE-T (Fang et al., 2022), SLBRN-M, SLBRN-T (Zhang & Wang, 2024) and ClippedGossip (Karimireddy et al., 2021). In aDFL method, we use cross-validation for the practical selection of $\lambda_n$. To investigate the effect of the number of neighboring nodes, we further consider two different network structures: the Directed Circle Network Wu et al. (2023a) with varying in-degree $D$, and the Undirected Erdős–Rényi Graph (Erdős & Rényi, 1959) with varying link probability $q$. Complete implementation details of the algorithms and network structures are provided in Appendix D.1.

### 5.1 SIMULATION EXPERIMENTS ON SYNTHETIC DATA

Following Qian et al. (2024), we consider the linear regression model $Y_i = X_i^\top\theta_0 + \varepsilon_i$, where $\varepsilon_i \sim N(0,1)$ and $\theta_0 = (\mathbf{1}_s^\top, 0, \ldots, 0)^\top \in \mathbb{R}^p$ with $s = \lfloor 0.2p \rfloor$. For the distribution of $X_i$, we study two scenarios: a homogeneous scenario with $X_i \sim N_p(0, I_p)$, and a heterogeneous scenario in which each client generates $X_i$ from distinct multivariate normal distributions. See Appendix D.2 for details. We consider the case where the data on abnormal clients is corrupted. Inspired by Karimireddy et al. (2021), Zhang & Wang (2024) and Qian et al. (2024), three types of data corruption are investigated:

- **Bit-Flipping (BF)**: The response variables $Y_i$'s on abnormal clients are replaced by $\widetilde{Y}_i = -Y_i$.

- **Out-of-Distribution (OOD)**: Features $X_i$'s on abnormal clients are replaced by $\widetilde{X}_i = 0.7X_i + V_p$, where entries of $V_p \in \mathbb{R}^p$ are independently generated from a uniform distribution $\mathcal{U}(0, 1)$.

- **Model-Parameter Corruption (MP)**: The parameters on abnormal clients are set as $\theta_c = (\mathbf{1}_{s_c}, 0, \ldots, 0)^\top \in \mathbb{R}^p$ with $s_c = \lfloor 0.1p \rfloor$.

We fix the feature dimension as $p = 50$, the number of clients as $M = 100$, and the local sample size as $n = 100$. Thus, the total sample size is given by $N = M \times n = 10,000$. We randomly select $\lfloor \varrho M \rfloor$ clients as abnormal clients. We then use MSE on normal clients to assess the performance of estimators computed by different algorithms. Specifically, the MSE is defined as $\text{MSE} = |\mathcal{A}^c|^{-1} \sum_{m \in \mathcal{A}^c} \|\widehat{\theta}^{(m)} - \theta_0\|^2$, where $\widehat{\theta}^{(m)} \in \mathbb{R}^p$ is the resulting estimator obtained on the $m$th client. For all algorithms, we replicate the experiments 20 times in each setting. The averaged values and confidence bands of these MSEs under the Directed Circle Network are shown in Figure 1, while those under the Undirected Erdős–Rényi Graph are present in Appendix D.2. The additional simulation results of the heterogeneous scenario can also be found in Appendix D.2. Moreover, to further strengthen our simulation study, we explore additional experiments involving (1) two more realistic network structures, (2) two more complex data corruption types, and (3) a dynamic corruption scenario under specific settings. Across these settings, the results consistently demonstrate the robustness and effectiveness of our approach; see Appendix D.2 for details.

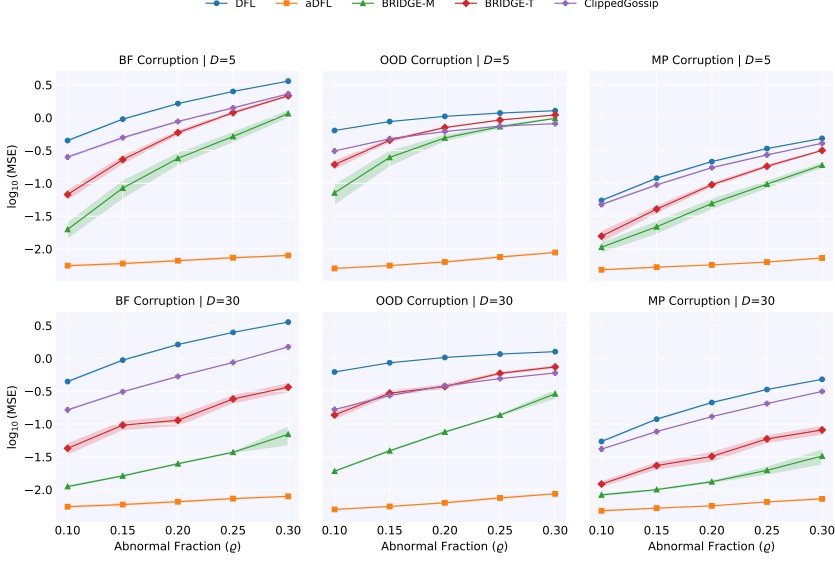

Figure 1: The logarithm of MSE values versus the fraction of abnormal clients ($\varrho$) under the Directed Circle Network and the homogeneous scenario. Different algorithms are evaluated under different corruption types and two in-degrees ($D$).

Generally, the results under the two network structures show similar patterns, from which we obtain the following observations. First, under this directed network structure, we find that the two SLBRN algorithms fail to converge, so the corresponding results are not reported. Second, as the abnormal fraction ($\varrho$) increases, the MSE of all algorithms increases significantly except for the aDFL algorithm. Furthermore, various abnormal robust algorithms exhibit a smaller MSE compared to the standard DFL algorithm. Moreover, the aDFL algorithm achieves the smallest MSE among all these algorithms. Lastly, we find that the performances of various robust algorithms improve in terms of MSE under the same Byzantine corruption type when the $D$ increases from 5 to 30. This is expected because more information can be transmitted with a larger number of neighboring clients.

## 5.2 APPLICATION TO REAL DATA

In this section, we empirically evaluate the effectiveness of our proposed aDFL method on two classical datasets: MNIST (LeCun et al., 1998) and CIFAR10 (Krizhevsky et al., 2009). MNIST contains 60,000 training and 10,000 testing images, whereas CIFAR10 contains 50,000 training

and 10,000 testing images. In this experiment, we distribute all training data equally to $M = 50$ clients. We consider two data distribution scenarios: (1) a homogeneous scenario, where images are randomly distributed; and (2) a heterogeneous scenario, where each client holds images from only a subset of labels. For abnormal clients, we implement both OOD and label-flipping (LF) corruption (Karimireddy et al., 2021). We train LeNet5 (LeCun et al., 1998) on MNIST using Xavier uniform initializer, and fine-tune a pre-trained VGG16 (Simonyan, 2014) on CIFAR10. To speed up convergence, we adopt a constant-and-cut learning-rate scheduling strategy (Lang et al., 2019). Further implementation details are provided in Appendix D.3. We also extend the real data analysis by exploring (i) a more heterogeneous scenario and (ii) the more challenging CINIC10 dataset (Darlow et al., 2018). The results again confirm the robustness and effectiveness of our approach; see Appendix D.3 and Figure D.12 for details.

At the $t$th iteration, we evaluate the performance of the $m$th client on the testing set. We then evaluate the performance of the $m$th client at the $t$th iteration using testing loss and accuracy. We plot the averaged values and confidence bands of these results on normal clients. In addition to the competing methods discussed above, we include the oracle estimator as a reference. In the main text, we present results for the CIFAR10 dataset under the heterogeneous scenario with LF corruption using a Directed Circle Network; see Figure 2. Additional results are provided in Appendix D.3.

From Figure 2, we find that as the fraction of abnormal clients increases or the number of neighbors decreases, the performances of competing methods decline significantly. Compared to competitors, our aDFL method achieves the best performance, which is comparable to that of the oracle across all situations. This highlights aDFL's strong ability to be adaptive to different scenarios.

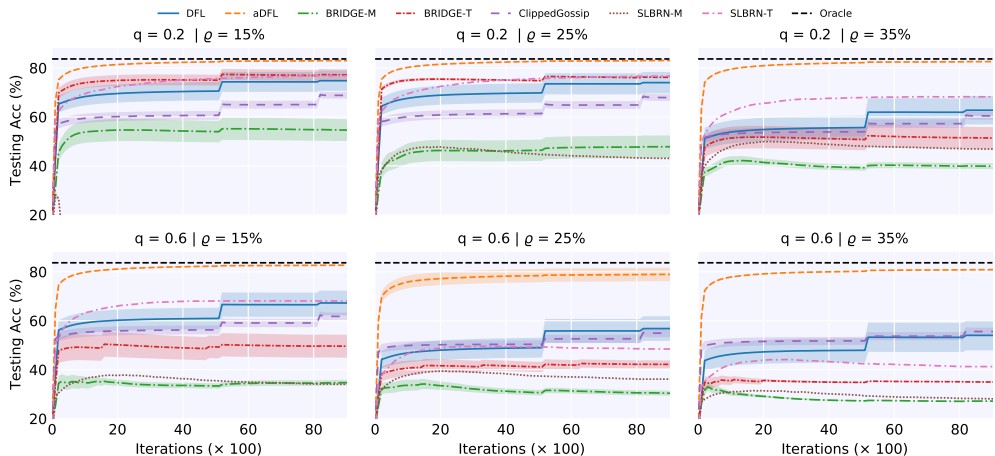

Figure 2: The testing accuracy over iterations for CIFAR10 in the heterogeneous scenario. Different methods are evaluated with varying link probabilities ($q$) and the fraction of abnormal clients ($\varrho$) under the LF corruption and Erdős–Rényi Graph.

## 6 CONCLUSION

In this work, we propose aDFL, a robust decentralized federated learning method that dynamically adjusts each client's learning rate based on training behavior. It preserves the original network topology and requires no stringent assumptions on neighbors or prior knowledge. We provide theoretical guarantees, and extensive experiments corroborate its effectiveness. However, several limitations remain. First, the current design primarily targets noisy/poisoned data; extending it to more general settings reqruies further study. Second, aDFL communicates every training round, which can be costly in large networks. Alleviating this via combining with local updating techniques is a key direction. Moreover, privacy mechanisms are not yet integrated, but our key technique (the introduction of $w_m$) can be easily extended to the existing privacy-preserving DFL methods. Lastly, our analysis assumes bounded gradients, which may not always hold; future work could consider using gradient clipping (Pascanu et al., 2013; Zhang et al., 2019) to relax this assumption.

## REPRODUCIBILITY STATEMENT

All numerical experiments and real-data analyses are fully reproducible using the code provided in the anonymized supplementary materials.

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

## A  ADDITIONAL DISCUSSIONS AND RESULTS

### A.1  THE PROPERTIES OF THE STANDARD DFL ESTIMATOR

Denote $\widehat{\theta}^{*(t)} = \{(\widehat{\theta}^{(t,1)})^\top, \ldots, (\widehat{\theta}^{(t,M)})^\top\}^\top \in \mathbb{R}^{Mp}$ be the stacked standard DFL estimator obtained at the $t$th iteration, and let $\widehat{\delta}_0 = \max_m \|\widehat{\theta}^{(0,m)} - \widehat{\theta}\|$ be the initial distance. The numerical convergence property of the standard DFL algorithm is elaborated by the following Proposition A.1.

**Proposition A.1** (Convergence Property of the Standard DFL). *Assume that Assumptions 1 – 6 hold. Further assume that $\alpha + \mathrm{SE}(W) < \epsilon$ and $\widehat{\delta}_0 < \epsilon$ for some sufficiently small but fixed $\epsilon$ depending on $(L_{\max}, \lambda_{\min}, \rho)$. Then, with probability at least $1 - O(M/n^4)$, the following relationship holds.*

$$M^{-1/2}\left\|\widehat{\theta}^{*(t)} - I^*\widehat{\theta}\right\| \lesssim \left(1 - \frac{\alpha(1-\varrho)\lambda_{\min}}{8}\right)^t \widehat{\delta}_0 + \frac{\alpha + \mathrm{SE}(W)}{(1-\rho)(1-\varrho)\lambda_{\min}}.$$

Proposition A.1 suggests that the discrepancy between the DFL estimator $\widehat{\theta}^{*(t)}$ obtained in the $t$th step and the whole-sample estimator $I^*\widehat{\theta}$ is upper bounded by: (1) the optimization error $\{1 - (\alpha\lambda_{\min})/8\}^t$ and (2) the statistical error $\{\alpha + \mathrm{SE}(W)\}/\{\lambda_{\min}(1-\rho)\}$. By the time of numerical convergence with $t \to \infty$, we obtain $M^{-1/2}\|\widehat{\theta}^{*(\infty)} - I^*\widehat{\theta}\| \leq \{\alpha + \mathrm{SE}(W)\}/\{\lambda_{\min}(1-\rho)\}$. Therefore, to have the difference between $\widehat{\theta}^{*(\infty)}$ and $\widehat{\theta}$ to be statistically ignorable, we should have $\alpha + \mathrm{SE}(W) = o(1/N)$. Moreover, we can combine the conclusions of Theorem 3.1 and Proposition A.1 to obtain an explicit bound on $\|\widehat{\theta}^{*(\infty)} - I^*\theta_0^*\|$ in terms of $n, M$, and $\rho$ as

$$M^{-1/2}\left\|\widehat{\theta}^{*(\infty)} - I^*\theta_0^*\right\| \lesssim \frac{\alpha + \mathrm{SE}(W)}{(1-\rho)(1-\varrho)} + \frac{1}{\sqrt{\delta}}\left\{\frac{1}{(1-\varrho)\sqrt{nM}} + \varrho\|\bar{\mathfrak{b}}_{\mathcal{A}}\| + \frac{(\varrho\|\bar{\mathfrak{b}}_{\mathcal{A}}\|)^{1/2}}{\sqrt{N}} + \frac{1}{nM}\right\}$$

with probability at least $1 - \delta$ for some small constant $\delta > 0$.

### A.2  THE THEORETICAL RESULTS UNDER GLOBAL STRONG CONVEXITY ASSUMPTION

**Assumption 3$'$** (Global Strong Convexity). *Assume there exists a fixed positive constant $\lambda_{\min}$, such that $\min_m \inf_{\theta \in \Theta} \lambda_{\min}\{\Omega_m(\theta)\} \geq \lambda_{\min}$.*

**Theorem A.1** (MSE of $\widehat{\theta}$). *Assume Assumptions 1 – 3$'$, and 4 – 6 hold. Then we have $\mathbb{E}\|\widehat{\theta} - \theta_0\|^2 = V(\widehat{\theta}) + \|\bar{\mathfrak{b}}_{\mathcal{A}}\| \quad B(\widehat{\theta})$, where $V(\widehat{\theta}) \lesssim L_{\max}^2/(\lambda_{\min}^2 N) + O(N^{-2})$, $NV(\widehat{\theta}) \to \mathrm{tr}\{\Omega_{\mathcal{A}}^{-1}\Sigma_{\mathcal{A}}\Omega_{\mathcal{A}}^{-1}\}$ as $N \to \infty$, and*

$$\frac{\varrho^2\|\bar{\mathfrak{b}}_{\mathcal{A}}\|}{L_{\max}^2} - C\left(\frac{\varrho}{N} + \frac{1}{N^3}\right) \leq B(\widehat{\theta}) \leq O\left(\varrho^2\|\bar{\mathfrak{b}}_{\mathcal{A}}\| + \frac{\varrho}{N} + \frac{1}{N^3}\right).$$

*Here $\bar{\mathfrak{b}}_{\mathcal{A}} = |\mathcal{A}|^{-1}\sum_{m \in \mathcal{A}}\mathfrak{b}_m$. The detailed formulas of $\Omega_{\mathcal{A}}$ and $\Sigma_{\mathcal{A}}$ are given in Appendix B.1.*

Since $\widehat{\theta}_{\mathcal{A}}$ is computed based solely on trust-able data, its properties can be directly derived by extending the classical properties of $M$-estimators (Van der Vaart, 2000; Serfling, 2009; Zhang et al., 2013). We then have the following proposition.

**Proposition A.2** (MSE of $\widehat{\theta}_{\mathcal{A}}$). *Assume that Assumptions 1 – 3$'$, and 4 hold. Then we have $\mathbb{E}\|\widehat{\theta}_{\mathcal{A}} - \theta_0\|^2 = V(\widehat{\theta}_{\mathcal{A}}) + B(\widehat{\theta}_{\mathcal{A}})$, where $V(\widehat{\theta}_{\mathcal{A}}) \leq \{\lambda_{\min}^2 N(1-\varrho)\}^{-1}2L_{\max}^2$, $NV(\widehat{\theta}_{\mathcal{A}}) \to \mathrm{tr}[\{(M - |\mathcal{A}|)^{-1}\sum_{m \notin \mathcal{A}}\Omega_m(\theta_0)\}^{-1}\{(M - |\mathcal{A}|)^{-1}\sum_{m \notin \mathcal{A}}\Sigma_m(\theta_0)\}\{(M - |\mathcal{A}|)^{-1}\sum_{m \notin \mathcal{A}}\Omega_m(\theta_0)\}^{-1}]$, and $B(\widehat{\theta}_{\mathcal{A}}) = O(N^{-2}(1-\varrho)^{-2})$.*

By comparing Proposition A.2 with Theorem A.1, we find that using only trustworthy clients should result in a superior estimator, as compared with the estimator computed based on all clients.

**Theorem A.2** (Convergence Property of $\widehat{\theta}_{\mathcal{A}}^{*(t)}$). *Assume that Assumptions 1 – 3′, and 4 – 6 hold, and $\alpha + \mathrm{SE}(W) < \epsilon$, for some sufficiently small but fixed $\epsilon$ depending on $(L_{\max}, \lambda_{\min}, \rho)$. Then, with probability at least $1 - O\big(M/n^4 + 1/(\log n)^4\big)$, we have $M^{-1/2}\|\widehat{\theta}_{\mathcal{A}}^{*(t)} - I^*\widehat{\theta}_{\mathcal{A}}\| \lesssim \mathrm{Err}_1 + \mathrm{Err}_2 + \mathrm{Err}_3$, where*

$$\mathrm{Err}_1 = \Big(1 - \frac{\alpha\bar{\omega}\lambda_{\min}}{8}\Big)^t \widehat{\delta}_0^{\mathcal{A}}, \ \mathrm{Err}_2 = \frac{\alpha L_{\max} + \mathrm{SE}(W)}{(1-\rho)\lambda_{\min}\bar{\omega}}\Big\{\Big(\frac{\log n}{n}\Big)^{1/2} L_{\max} + \varrho^{1/2}\bar{b}_2^{\mathcal{A}}\Big\},$$

$$\mathrm{Err}_3 = \frac{1}{\bar{\omega}\lambda_{\min}}\Big[\varrho\bar{b}_2^{\mathcal{A}}\bar{\omega}_2^{\mathcal{A}} + \bar{\Delta}_2\Big\{\Big(\frac{\log N}{N}\Big)^{\frac{1}{2}} + L_{\max}\|\widehat{\theta}_{\mathcal{A}} - \theta_0\|\Big\}\Big].$$

*Assume $M = o(n^4)$ as $n \to \infty$. Then with probability tending to 1, we have $M^{-1/2}\|\widehat{\theta}_{\mathcal{A}}^{*(\infty)} - I^*\widehat{\theta}_{\mathcal{A}}\|$ upper bounded by*

$$\frac{C}{\bar{\omega}}\Big[\big\{\alpha + \mathrm{SE}(W)\big\}\big(n^{-1/2} + \varrho^{1/2}\big) + \big(\varrho\bar{\omega}_2^{\mathcal{A}} + \frac{\bar{\Delta}_2}{\sqrt{N}}\big)\Big].$$

**Theorem A.3** (Convergence Rate of the aDFL). *Assume that Assumptions 1 – 3′, and 4 – 6 hold. Let $\pi(x) = \exp(-x)$, and set the initial value $\widehat{\theta}_r^{(m)}$ as the standard DFL estimator. Assume that $\log N \lesssim \lambda_n \lesssim \sqrt{n}M^{-1/8}$. Then, with probability at least $1 - O(M/n^4 + 1/\log N)$, we have: (1) $\hat{\bar{\omega}}_2^{\mathcal{A}} \lesssim 1/\sqrt{N}$, (2) $\hat{\bar{\Delta}}_2^2 \lesssim \varrho/\sqrt{N} + \lambda_n(1/\sqrt{n} + \|\widehat{\theta} - \theta_0\|)$, and (3) $1/\hat{\omega} \lesssim \exp(c\lambda_n\|\widehat{\theta} - \theta_0\|)$. Further assume $M \to \infty$ with $M = o(n^4)$, $\varrho = o(1)$ and $\alpha + \mathrm{SE}(W)$ is sufficiently small as $n \to \infty$. Then, with probability tending to 1, we have $M^{-1/2}\|\widehat{\theta}_{\mathrm{aDFL}}^{*(\infty)} - I^*\widehat{\theta}_{\mathcal{A}}\|$ upper bounded by*

$$C\exp\big(c\lambda_n\|\widehat{\theta} - \theta_0\|\big)\Big\{\frac{\lambda_n\|\widehat{\theta} - \theta_0\|}{\sqrt{N}} + o\Big(\frac{1}{\sqrt{N}}\Big)\Big\}.$$

### A.3 THE ASSUMPTION FOR COROLLARY 4.1.

**Assumption A.1** (Consensus Convergence). *Denote $\bar{\theta}_{\mathrm{init}} = M^{-1}\sum_{m=1}^M \widehat{\theta}_{\mathrm{init}}^{(m)}$. Assume that (1) $\lambda_n\|\bar{\theta}_{\mathrm{init}} - \theta_0\| = O_p(1)$, and (2) $\lambda_n M^{-1}\sum_{m=1}^M \|\widehat{\theta}_{\mathrm{init}}^{(m)} - \bar{\theta}_{\mathrm{init}}\|^2 = o_p(1/N^2)$.*

**Remark 3.** *Assumption A.1 requires that the initial estimators $\{\widehat{\theta}_{\mathrm{init}}^{(m)}\}_m$ have a clear consensus in the sense that their sample variance is of the order $o_p(1/(\sqrt{\lambda_n}N))$. Moreover, their consensus should be of a reasonable quality in the sense that $\lambda_n\|\bar{\theta}_{\mathrm{init}} - \theta_0\| = O_p(1)$. Such types of initial estimators can be easily obtained by, for example, (1) a standard DFL algorithm with a sufficiently small $\alpha + \mathrm{SE}(W)$ value; or (2) a gradient tracking algorithm of Shi et al. (2015) on a symmetric doubly stochastic $W$.*

### A.4 MULTI-STAGE ADFL ALGORITHM

It is worth noting that in Algorithm A.1, the key step of the aDFL method involves computing adaptive weights using Equation (4.4). These weights adjust each client's training contribution based on its behavior. Consequently, our proposed aDFL method can be smoothly extended to many existing DFL frameworks. Moreover, with an appropriate choice of $\lambda_n$, one can expect to achieve nearly oracle performance.

### A.5 THE CHOICE OF $\lambda_n$

We propose here a method of decentralized cross validation (DCV) for an automatic selection of $\lambda_n$. Specifically, for any $1 \leq m \leq M$, we split $\mathcal{S}_m$ into a training set $\mathcal{S}_m^{\mathrm{train}}$ and a validation set $\mathcal{S}_m^{\mathrm{val}}$. The training set $\cup_m\mathcal{S}_m^{\mathrm{train}}$ is then used to obtain the aDFL estimator under different $\lambda_n$ values, while the validation set $\cup_m\mathcal{S}_m^{\mathrm{val}}$ is used to evaluate the aDFL estimator's performance. In the presence of Byzantine attacks in DFL, the most ideal global metric should be $\mathcal{L}^{\mathrm{val}}(\{\theta^{(m)}\}_m) = |\mathcal{A}^c|^{-1}\sum_{m=1}^M (1 - a_m)\sum_{i\in\mathcal{S}_m^{\mathrm{val}}} \ell(X_i, Y_i; \theta^{(m)})$, which is the validation losses computed on all trustworthy clients. By leveraging the aDFL estimator $\widehat{\theta}_{\mathrm{aDFL}}^{*(T)}$ and the adaptive

---

### Algorithm A.1: Multi-stage Adaptive Decentralized Federated Learning

---

**Require:** initial estimator $\{\widehat{\theta}_{\text{init}}^{(m)}\}_{m=1}^{M}$; max iteration $T$; number of stages $S$;
**Ensure:** aDFL estimator $\{\widehat{\theta}_{\text{aDFL}}^{(T,m)}\}_{m=1}^{M}$

  1: **for** $s \leq S$ **do**
  2:    Set $\widehat{\theta}_{\text{aDFL}}^{(0,m)} = \widehat{\theta}_{\text{init}}^{(m)}$ for $1 \leq m \leq M$
  3:    **for** $0 \leq t \leq T-1$ **do**
  4:      **for** $1 \leq m \leq M$ (distributedly) **do**
  5:        Compute the neighborhood-averaged estimator $\widetilde{\theta}_{\text{aDFL}}^{(t,m)} = \sum_k w_{mk}\widehat{\theta}_{\text{aDFL}}^{(t,k)}$
  6:        Update parameter estimator by

$$\widehat{\theta}_{\text{aDFL}}^{(t+1,m)} = \widetilde{\theta}_{\text{aDFL}}^{(t,m)} - \alpha\widehat{\omega}_m \dot{\mathcal{L}}_{(m)}(\widetilde{\theta}_{\text{aDFL}}^{(t,m)}),$$

        where $\widehat{\omega}_m$ is computed as (4.4)
  7:      **end for**
  8:    **end for**
  9: **end for**

---

weights $\{\widehat{\omega}_m\}_m$ produced by the aDFL algorithm on the training data, we can construct an estimator for the ideal loss function. Specifically, it is defined as

$$\widehat{\mathcal{L}}^{\text{val}}(\widehat{\theta}_{\text{aDFL}}^{*(T)}) = (M\widehat{\omega})^{-1}\sum_{m=1}^{M}\widehat{\omega}_m \sum_{i\in\mathcal{S}_m^{\text{val}}} \ell(X_i, Y_i; \widehat{\theta}_{\text{aDFL}}^{(T,m)}) + (M\widehat{\omega})^{-1}\lambda n_{\text{val}}^{-1/2}\Big(\sum_{m=1}^{M}\widehat{\omega}_m^2\Big)^{1/2}.$$

Here $n_{\text{val}} = |\mathcal{S}_{\text{val}}|$ and $\lambda$ represents a penalty on the adaptive weights to balance the estimation value and the variance of $\widehat{\mathcal{L}}^{\text{val}}(\widehat{\theta}_{\text{aDFL}}^{*(T)})$. We observe that $\lambda$ is not highly sensitive in our algorithm. In practice, we set $\lambda = 1.64$. Then the DCV algorithm can be executed for each $\lambda_n^k$ in a two stage manner:

In the first stage, each client $m$ executes Algorithm 1 to obtain the aDFL estimator $\widehat{\theta}_{\text{aDFL}}^{(T,m)}$ and $\{\widehat{\omega}_m\}_m$. Based on this estimator, every client calculates its validation loss, denoted as $\widehat{\mathcal{L}}_{\text{val}}^{(0,m)}$ and recorded as the initial loss value. We also define the initial averaged adaptive weight as $\hat{\hat{\omega}}^{(0,m)} = \widehat{\omega}_m$.

In the second stage, an iterative algorithm should be executed on the decentralized network, so that an estimator for $\widehat{\mathcal{L}}^{\text{val}}(\widehat{\theta}_{\text{aDFL}}^{*(T)})$ with consensus can be obtained. To this end, assume that client $m$ has obtained the loss value $\widehat{\mathcal{L}}_{\text{val}}^{(t,m)}$ and averaged adaptive weight $\hat{\hat{\omega}}^{(t,m)}$ at iteration $t$. At the $(t+1)$th iteration, the loss value and averaged adaptive weight are updated as follows:

$$\widehat{\mathcal{L}}_{\text{val}}^{(t+1,m)} = \sum_{k=1}^{M} w_{mk}\widehat{\mathcal{L}}_{\text{val}}^{(t,k)}; \quad \hat{\hat{\omega}}^{(t+1,m)} = \sum_{k=1}^{M} w_{mk}\hat{\hat{\omega}}^{(t,k)}.$$

Then, by similar technique of Yuan et al. (2016) and Wu et al. (2023a), it can be proved that $\widehat{\mathcal{L}}_{\text{val}}^{(t,m)} \to \widehat{\mathcal{L}}_{\text{val}}^{\infty} \approx M^{-1}\sum_{m=1}^{M} \widehat{\mathcal{L}}_{\text{val}}^{(0,m)}$ and $\hat{\hat{\omega}}^{(t,m)} \to \hat{\hat{\omega}}^{\infty} \approx \hat{\hat{\omega}}$ as $t \to \infty$ under appropriate regularity conditions. Finally, the optimal $\lambda_n$ is selected as $\lambda_n^{\text{opt}} = arg\,min_{\lambda_n \in \{\lambda_n^k\}_k} \mathcal{L}_{\text{val}}^{\infty}/\hat{\hat{\omega}}^{\infty}$. Subsequently, we present the complete DCV algorithm in Algorithm A.2.

## B  PROOF OF THE MAIN THEORETICAL RESULTS

We first show that under the assumption of global strong convexity, the theorem presented in Section A.2 holds. We then extend this result to the setting of local strong convexity.

### B.1  NOTATIONS AND PRELIMITS

Let $I_p$ be the $p \times p$ identity matrix. Define $\mathbf{1}_M = (1,\ldots,1)^\top \in \mathbb{R}^M$ and $I^* = \mathbf{1}_M \otimes I_p \in \mathbb{R}^{Mp \times p}$. For a sequence $\{a^{(t)}\}$, define $a^{(\infty)} = \lim_{t\to\infty} a^{(t)}$. For two positive sequences $\{a_n\}$ and $\{b_n\}$,

---

Algorithm A.2: DCV for Choosing $\lambda_n$

---

**Require:** candidate sets $\{\lambda_n^k\}_k$, training set $\{\mathcal{S}_m^{\text{train}}\}$ and validation set $\{\mathcal{S}_m^{\text{val}}\}$, tuning parameter $\lambda$;
**Ensure:** $\lambda_n^{\text{opt}}$

1: **for** $\lambda_n^k$ in $\{\lambda_n^k\}_k$ **do**
2:     Obtain $\widehat{\theta}_{\text{aDFL}}^{*(T)}$ and $\{\widehat{\omega}_m\}_m$ by Algorithm 1 on $\{\mathcal{S}_m^{\text{train}}\}_m$.
3:     **for** $1 \leq m \leq M$ (distributedly) **do**
4:         Compute initial estimators $\hat{\hat{\omega}}^{2(0,m)} = \widehat{\omega}_m^2$, $\hat{\hat{\omega}}^{(0,m)} = \widehat{\omega}_m$ and $\widehat{\mathcal{L}}_{\text{val}}^{(0,m)} = \widehat{\omega}_m \sum_{i \in \mathcal{S}_m^{\text{val}}} \ell(X_i, Y_i; \widehat{\theta}_{\text{aDFL}}^{(T,m)})$.
5:     **end for**
6:     **for** $0 \leq t \leq T-1$ **do**
7:         **for** $1 \leq m \leq M$ (distributedly) **do**
8:             Update estimators by $\hat{\hat{\omega}}^{2(t+1,m)} = \sum_{k=1}^M w_{mk} \hat{\hat{\omega}}^{2(t,m)}$; $\hat{\hat{\omega}}^{(t+1,m)} = \sum_{k=1}^M w_{mk} \hat{\hat{\omega}}^{(t,m)}$ and $\widehat{\mathcal{L}}_{\text{val}}^{(t+1,m)} = \sum_{k=1}^M w_{mk} \widehat{\mathcal{L}}_{\text{val}}^{(t,k)}$.
9:         **end for**
10:     **end for**
11:     For each $m$, we have $\hat{\hat{\omega}}^{2(T,m)} \equiv \hat{\hat{\omega}}^{2(T)}$, $\hat{\hat{\omega}}^{(T,m)} \equiv \hat{\hat{\omega}}^{(T)}$ and $\widehat{\mathcal{L}}_{\text{val}}^{(T,m)} \equiv \widehat{\mathcal{L}}_{\text{val}}^{(T)}$ for sufficiently large $T$. Denote $\hat{\hat{\omega}}^{2(T)}, \hat{\hat{\omega}}^{(T)}, \widehat{\mathcal{L}}_{\text{val}}^{(T)}$ as $\hat{\hat{\omega}}^{2(T)}(\lambda_n^k), \hat{\hat{\omega}}^{(T)}(\lambda_n^k), \widehat{\mathcal{L}}_{\text{val}}^{(T)}(\lambda_n^k)$.
12: **end for**
13: Obtain $\lambda_n^{\text{opt}} = arg\,min_{\lambda_n^k \in \{\lambda_n^k\}_k} (M\hat{\hat{\omega}}^{(T)}(\lambda_n^k))^{-1}\{\widehat{\mathcal{L}}_{\text{val}}^{(T)}(\lambda_n^k) + \lambda n_{\text{val}}^{-1/2}(M\hat{\hat{\omega}}^{2(T)}(\lambda_n^k))^{1/2}\}$.

---

write $a_n \ll b_n$ or $a_n = o(b_n)$ if $a_n/b_n \to 0$ as $n \to \infty$. Write $a_n \lesssim b_n$ or $a_n = O(b_n)$ if $a_n/b_n \leq C < \infty$ as $n \to \infty$. For a vector $x \in \mathbb{R}^p$, denote its Euclidean norm by $\|x\|$. For a symmetric matrix $B \in \mathbb{R}^{p \times p}$, denote its smallest and largest eigenvalues by $\lambda_{\min}(B)$ and $\lambda_{\max}(B)$, respectively. For an arbitrary matrix $B \in \mathbb{R}^{p_1 \times p_2}$, define its $\ell_2$-norm as $\|B\| = \lambda_{\max}^{1/2}(B^\top B)$. For a set $S$, denote its cardinality by $|S|$ and represent its complement by $S^c$. Denote $\mathbb{E}_m(\cdot)$ stands for the expectation with respect to a probability distribution $\mathcal{P}_m$. The generic absolute constants $c$ and $C$ may vary from line to line.

For simplicity of notation, write $\ell(X_i, Y_i; \theta)$ as $\ell_i(\theta)$, denote $e_m^{\mathcal{A}}(\theta) = \mathbb{E}_m\{\ell_i(\theta)\}$. Define $\theta_{\mathcal{A}} = arg\,min_\theta M^{-1} \sum_{m=1}^M e_m^{\mathcal{A}}(\theta)$ represents pseudo-true parameter. Denote $\bar{\omega} = M^{-1} \sum_{m=1}^M \omega_m$, $\Sigma_{\mathcal{A}}(\theta) = M^{-1} \sum_{m=1}^M [\mathbb{E}_m\{\dot{\ell}_i(\theta) - \mathbb{E}_m(\dot{\ell}_i(\theta))\}\{\dot{\ell}_i(\theta) - \mathbb{E}_m(\dot{\ell}_i(\theta))\}^\top]$ and $\Omega_{\mathcal{A}}(\theta) = M^{-1} \sum_{m=1}^M \{\Omega_m(\theta)\}$.

### B.2 Proof of Theorem A.1 and Proposition A.2

**Proof of Theorem A.1:** We decompose $\mathbb{E}\|\widehat{\theta} - \theta_0\|^2 = V(\widehat{\theta}) + B(\widehat{\theta})$, where $V(\widehat{\theta}) = \mathbb{E}\|\widehat{\theta} - \theta_{\mathcal{A}}\|^2$, and $B(\widehat{\theta}) = 2\mathbb{E}(\widehat{\theta} - \theta_{\mathcal{A}})^\top(\theta_{\mathcal{A}} - \theta_0) + \|\theta_{\mathcal{A}} - \theta_0\|^2$. We next investigate the two terms separately.

STEP 1. We first study $V(\widehat{\theta})$. Note that $\mathbb{E}\{\mathcal{L}(\theta)\} = M^{-1} \sum_{m=1}^M e_m^{\mathcal{A}}(\theta)$. Additionally, since $X_i$ and $Y_i$ are independently generated (though not identically distributed), based on Assumptions 1 – 4, along with the events $\mathcal{E}_1$ and $\mathcal{E}_2$ defined in Lemma C.1, we can apply Equation (23) and employ similar proof techniques from Appendix B of Zhang et al. (2013), specifically Sections B.01 and B.02, then we obtain:

$$V(\widehat{\theta}) = \mathbb{E}\|\widehat{\theta} - \theta_{\mathcal{A}}\|^2 \lesssim \frac{L_{\max}^2}{\lambda_{\min}^2 N} + O\left(\frac{1}{N^2}\right). \tag{B.1}$$

$$\mathbb{E}\|\widehat{\theta} - \theta_{\mathcal{A}}\| \lesssim \frac{L_{\max}^3}{N\lambda_{\min}^3} + O\left(\frac{1}{N^3}\right). \tag{B.2}$$

In addition, we have $V(\widehat{\theta}) = \mathbb{E}\| - \Omega_{\mathcal{A}}^{-1}(\theta_{\mathcal{A}})\dot{\mathcal{L}}(\theta_{\mathcal{A}})\|^2\{1 + o(1)\}$. Then it could be verified that $\mathbb{E}\|\Omega_{\mathcal{A}}^{-1}(\theta_{\mathcal{A}})\dot{\mathcal{L}}(\theta_{\mathcal{A}})\|^2$

$$= \text{tr}\left[\Omega_{\mathcal{A}}^{-1}(\theta_{\mathcal{A}})\mathbb{E}\left\{N^{-1}\sum_{i=1}^N \dot{\ell}_i(\theta_{\mathcal{A}})\right\}\left\{N^{-1}\sum_{i=1}^N \ddot{\ell}_i^\top(\theta_{\mathcal{A}})\right\}\Omega_{\mathcal{A}}^{-1}(\theta_{\mathcal{A}})\right]$$

$$
= \quad \mathrm{tr}\left[\Omega_{\mathcal{A}}^{-1}(\theta_{\mathcal{A}})\mathbb{E}\left\{N^{-1}\sum_{i=1}^{N}\left(\dot{\ell}_i(\theta_{\mathcal{A}}) - \mathbb{E}\dot{\ell}_i(\theta_{\mathcal{A}})\right)\right\}\left\{N^{-1}\sum_{i=1}^{N}\left(\dot{\ell}_i(\theta_{\mathcal{A}}) - \mathbb{E}\dot{\ell}_i(\theta_{\mathcal{A}})\right)^{\top}\right\}\Omega_{\mathcal{A}}^{-1}(\theta_{\mathcal{A}})\right]
$$

$$
= \quad N^{-1}\,\mathrm{tr}\left\{\Omega_{\mathcal{A}}^{-1}(\theta_{\mathcal{A}})\Sigma_{\mathcal{A}}(\theta_{\mathcal{A}})\Omega_{\mathcal{A}}^{-1}(\theta_{\mathcal{A}})\right\}.
$$

This yields

$$
NV(\widehat{\theta}) \to \mathrm{tr}\left\{\Omega_{\mathcal{A}}^{-1}(\theta_{\mathcal{A}})\Sigma_{\mathcal{A}}(\theta_{\mathcal{A}})\Omega_{\mathcal{A}}^{-1}(\theta_{\mathcal{A}})\right\} \text{ as } N \to \infty.
$$

STEP 2. We next investigate $(\theta_{\mathcal{A}} - \theta_0)^{\top}\mathbb{E}(\widehat{\theta} - \theta_{\mathcal{A}})$ and $\|\theta_{\mathcal{A}} - \theta_0\|^2$. By definition, we know that $M^{-1}\sum_{m=1}^{M}\dot{e}_m^{\mathcal{A}}(\theta_{\mathcal{A}}) = 0$, this leads to

$$
0 \quad = \quad M^{-1}\sum_{m=1}^{M}\left\{\dot{e}_m^{\mathcal{A}}(\theta_{\mathcal{A}}) - \dot{e}_m^{\mathcal{A}}(\theta_0)\right\} + M^{-1}\sum_{m=1}^{M}\dot{e}_m^{\mathcal{A}}(\theta_0).
$$

By definition, we know that $\mathbb{E}_m\{\dot{\ell}_i(\theta_0)\} = 0$ for $m \notin \mathcal{A}$, then it can be verified that

$$
M^{-1}\sum_{m=1}^{M}\dot{e}_m^{\mathcal{A}}(\theta_0) = \frac{|\mathcal{A}|}{M}|\mathcal{A}|^{-1}\sum_{m\in\mathcal{A}}\flat_m = \frac{|\mathcal{A}|\bar{\flat}_{\mathcal{A}}}{M}. \tag{B.3}
$$

We subsequently establish both the upper and lower bounds for $\|\theta_{\mathcal{A}} - \theta_0\|$. First, it could be verified that for any $\theta \in \Theta$, $\lambda_{\min}\{\ddot{e}_m^{\mathcal{A}}(\theta)\} = \lambda_{\min}\{\Omega_m(\theta)\} \geq \lambda_{\min}$ by Assumption $3'$. Then we have

$$
0 \quad = \quad M^{-1}\sum_{m=1}^{M}\left\{\dot{e}_m^{\mathcal{A}}(\theta_{\mathcal{A}}) - \dot{e}_m^{\mathcal{A}}(\theta_0)\right\}^{\top}(\theta_{\mathcal{A}} - \theta_0) + \frac{|\mathcal{A}|}{M}(\bar{\flat}_{\mathcal{A}})^{\top}(\theta_{\mathcal{A}} - \theta_0)
$$

$$
\geq \quad \lambda_{\min}\|\theta_{\mathcal{A}} - \theta_0\|^2 + \frac{|\mathcal{A}|}{M}(\bar{\flat}_{\mathcal{A}})^{\top}(\theta_{\mathcal{A}} - \theta_0).
$$

This yields $\|\theta_{\mathcal{A}} - \theta_0\| \leq \lambda_{\min}^{-1}|\mathcal{A}|M^{-1}\|\bar{\flat}_{\mathcal{A}}\|$. In addition, it could be proved that for any $\theta', \theta'' \in \Theta$, $\|\dot{e}_m^{\mathcal{A}}(\theta') - \dot{e}_m^{\mathcal{A}}(\theta'')\| \leq L_{\max}\|\theta' - \theta''\|$ by definition and Assumption 4. Then we have

$$
\left\|M^{-1}\sum_{m=1}^{M}\left\{\dot{e}_m^{\mathcal{A}}(\theta_{\mathcal{A}}) - \dot{e}_m^{\mathcal{A}}(\theta_0)\right\}\right\| \quad = \quad \left\|\frac{|\mathcal{A}|}{M}\bar{\flat}_{\mathcal{A}}\right\| \implies \frac{|\mathcal{A}|}{M}\|\bar{\flat}_{\mathcal{A}}\|
$$

$$
\leq \quad M^{-1}\sum_{m=1}^{M}L_{\max}\|\theta_{\mathcal{A}} - \theta_0\| = L_{\max}\|\theta_{\mathcal{A}} - \theta_0\|.
$$

As a consequence, we can obtain

$$
\frac{|\mathcal{A}|}{M}L_{\max}^{-1}\|\bar{\flat}_{\mathcal{A}}\| \leq \|\theta_{\mathcal{A}} - \theta_0\| \leq \frac{|\mathcal{A}|}{M}\lambda_{\min}^{-1}\|\bar{\flat}_{\mathcal{A}}\|. \tag{B.4}
$$

Combining the results of (B.2) and (B.4), we know

$$
\left\|(\theta_{\mathcal{A}} - \theta_0)^{\top}\mathbb{E}(\widehat{\theta} - \theta_{\mathcal{A}})\right\| \lesssim \frac{|\mathcal{A}|}{NM}\|\bar{\flat}_{\mathcal{A}}\| + O\left(\frac{\|\bar{\flat}_{\mathcal{A}}\|}{N^3}\right). \tag{B.5}
$$

Combining the results of (B.4) and (B.5), we have

$$
\frac{|\mathcal{A}|^2}{M^2}L_{\max}^{-2}\|\bar{\flat}_{\mathcal{A}}\|^2 - 2\left\|(\theta_{\mathcal{A}} - \theta_0)^{\top}\mathbb{E}(\widehat{\theta} - \theta_{\mathcal{A}})\right\| \leq B(\widehat{\theta}) \leq \frac{|\mathcal{A}|^2}{M^2}\lambda_{\min}^{-2}\|\bar{\flat}_{\mathcal{A}}\|^2 + 2\left\|(\theta_{\mathcal{A}} - \theta_0)^{\top}\mathbb{E}(\widehat{\theta} - \theta_{\mathcal{A}})\right\|.
$$

Simplify $\Omega_{\mathcal{A}}(\theta_{\mathcal{A}})$ and $\Sigma_{\mathcal{A}}(\theta_{\mathcal{A}})$ to $\Omega_{\mathcal{A}}$ and $\Sigma_{\mathcal{A}}$. Further note that $\varrho = |\mathcal{A}|/M$, this finishes the theorem proof.

**Proof of Proposition A.2:** The proof of Proposition A.2 is similar to that of Theorem A.1. Thus, we omit the detailed proof here.

## B.3 PROOF OF THEOREM A.2 AND PROPOSITION A.1

We first introduce some notations. Denote $\omega_m^* = 1 - a_m$ represents the oracle weight, $\Delta_m = \omega_m - \omega_m^*$, $\widehat{\text{SE}}^2 = M^{-1} \sum_{m=1}^M \|\dot{\mathcal{L}}_{(m)}(\widehat{\theta}_{\mathcal{A}})\|^2$, and $\overline{\text{SE}}_\omega = \|M^{-1} \sum_{m=1}^M (\omega_m - \omega^*)\dot{\mathcal{L}}_m(\widehat{\theta}_{\mathcal{A}})\|$. Recall that $\bar{\omega} = M^{-1} \sum_{m=1}^M \omega_m$, $\overline{\omega}_2^{\mathcal{A}} = (|\mathcal{A}|^{-1} \sum_{m \in \mathcal{A}} \omega_m^2)^{1/2}$, $\bar{b}_2^{\mathcal{A}} = (|\mathcal{A}|^{-1} \sum_{m \in \mathcal{A}} \|b_m\|^2)^{1/2}$. $\bar{\theta}_{\mathcal{A}}^{(t)} = M^{-1} \sum_{m=1}^M \widehat{\theta}_{\mathcal{A}}^{(t,m)}$ represent the averaged estimator in the $t$th iteration. Let $\widehat{\theta}_{\mathcal{A}}^* = I^* \widehat{\theta}_{\mathcal{A}}$, $\bar{\theta}_{\mathcal{A}}^{*(t)} = I^* \bar{\theta}_{\mathcal{A}}^{(t)}$.

We will work with the weighted loss function and weighted Hessian matrix defined as follows:

$$\mathcal{L}_w(\theta) = M^{-1} \sum_{m=1}^M \omega_m \mathcal{L}_m(\theta)$$

$$\Omega_{\mathcal{A}}^w(\theta) = \mathbb{E}\{\ddot{\mathcal{L}}_w(\theta)\} = M^{-1} \sum_{m=1}^M \omega_m \Omega_m(\theta).$$

By Lemma C.1, to prove Theorem 4.1, it suffices to study the upper bound of $\widehat{\theta}_{\mathcal{A}}^{*(t+1)} - \widehat{\theta}_{\mathcal{A}}^*$ under the good events $\bigcap_m \mathcal{E}_{1,m} \bigcap \mathcal{E}_2^\omega \bigcap_m \mathcal{E}_{3,m} \bigcap \mathcal{E}_4 \bigcap \mathcal{E}_5$. Then the proof of this theorem is divided into three steps. In the first and second steps, we decompose $\widehat{\theta}_{\mathcal{A}}^{*(t+1)} - \widehat{\theta}_{\mathcal{A}}^*$ to $\widehat{\theta}_{\mathcal{A}}^{*(t+1)} - \bar{\theta}_{\mathcal{A}}^{*(t+1)}$ and $\bar{\theta}_{\mathcal{A}}^{*(t+1)} - \widehat{\theta}_{\mathcal{A}}^*$ and analyze these two terms separately. In the third step, we combine the results from the first two steps to derive the final theorem.

STEP 1. We first study $\|\bar{\theta}_{\mathcal{A}}^{*(t+1)} - \widehat{\theta}_{\mathcal{A}}^*\|$. It could be verified that

$$\|\bar{\theta}_{\mathcal{A}}^{(t+1)} - \widehat{\theta}_{\mathcal{A}}\| \leq \|\bar{\theta}_{\mathcal{A}}^{(t)} - \widehat{\theta}_{\mathcal{A}} - \alpha M^{-1} \sum_{m=1}^M \omega_m \dot{\mathcal{L}}_m(\bar{\theta}_{\mathcal{A}}^{(t)})\| + \alpha M^{-1} \sum_{m=1}^M \|\omega_m \dot{\mathcal{L}}_m(\widetilde{\theta}_{\mathcal{A}}^{(t,m)})$$

$$-\omega_m \dot{\mathcal{L}}_m(\bar{\theta}_{\mathcal{A}}^{(t)})\| + \|M^{-1} \sum_{m=1}^M \widetilde{\theta}_{\mathcal{A}}^{(t,m)} - \bar{\theta}_{\mathcal{A}}^{(t)}\| = \Delta_{(1)} + \Delta_{(2)} + \Delta_{(3)}.$$

**(i) Analysis of $\Delta_{(1)}$**: Note that $\lambda_{\min}(\Omega_{\mathcal{A}}^\omega(\theta)) \geq \bar{\omega}\lambda_{\min}$ by definition and Assumption 3'. Then we have

$$\Delta_{(1)} = \|\bar{\theta}_{\mathcal{A}}^{(t)} - \widehat{\theta}_{\mathcal{A}} - \alpha \dot{\mathcal{L}}_\omega(\bar{\theta}_{\mathcal{A}}^{(t)})\| \leq \|\bar{\theta}_{\mathcal{A}}^{(t)} - \widehat{\theta}_{\mathcal{A}} - \alpha \dot{\mathcal{L}}_\omega(\bar{\theta}_{\mathcal{A}}^{(t)}) - \alpha \dot{\mathcal{L}}_\omega(\widehat{\theta}_{\mathcal{A}})\|$$

$$+\alpha \|M^{-1} \sum_{m=1}^M \omega_m \dot{\mathcal{L}}_m(\widehat{\theta}_{\mathcal{A}}) - M^{-1} \sum_{m=1}^M \omega_m^* \dot{\mathcal{L}}_m(\widehat{\theta}_{\mathcal{A}})\|$$

$$\leq (1 - \alpha\bar{\omega}\lambda_{\min}/2)\|\bar{\theta}_{\mathcal{A}}^{(t)} - \widehat{\theta}_{\mathcal{A}}\| + \alpha\overline{\text{SE}}_\omega.$$

The second inequality holds since $\ddot{\mathcal{L}}_w(\theta) \geq \bar{\omega}\lambda_{\min}/2$ under $\mathcal{E}_2^\omega$ and $\sum_{m=1}^M \omega_m^* \dot{\mathcal{L}}_m(\widehat{\theta}_{\mathcal{A}}) = 0$.

**(ii) Analysis of $\Delta_{(2)}$**:

$$M\Delta_{(2)}/\alpha = \sum_{m=1}^M \omega_m \|\dot{\mathcal{L}}_m(\widetilde{\theta}_{\mathcal{A}}^{(t,m)}) - \dot{\mathcal{L}}_m(\bar{\theta}_{\mathcal{A}}^{(t)})\| \leq 2 \sum_{m=1}^M \omega_m L_{\max}\|\widetilde{\theta}_{\mathcal{A}}^{(t,m)} - \bar{\theta}_{\mathcal{A}}^{(t)}\|$$

$$\leq 2\sqrt{M}L_{\max}\|(W \otimes I_p)\widehat{\theta}_{\mathcal{A}}^{*(t)} - \bar{\theta}_{\mathcal{A}}^{*(t)}\| \leq 2\sqrt{M}L_{\max}\rho\|\widehat{\theta}_{\mathcal{A}}^{*(t)} - \bar{\theta}_{\mathcal{A}}^{*(t)}\|.$$

The first inequality holds under $\cap_m \mathcal{E}_{1,m}$, and the last inequality holds by Lemma C.2 (i). Finally, we have $\Delta_{(3)} \leq M^{-1/2} \text{SE}(W)\|\widehat{\theta}_{\mathcal{A}}^{*(t)} - \bar{\theta}_{\mathcal{A}}^{*(t)}\|$ by Lemma C.2 (ii). This leads to

$$\|\bar{\theta}_{\mathcal{A}}^{(t+1)} - \widehat{\theta}_{\mathcal{A}}\| \leq \left(1 - \frac{\alpha\bar{\omega}\lambda_{\min}}{2}\right)\|\bar{\theta}_{\mathcal{A}}^{(t)} - \widehat{\theta}_{\mathcal{A}}\| + M^{-1/2}\big\{2L_{\max}\alpha\rho + \text{SE}(W)\big\}\|\widehat{\theta}_{\mathcal{A}}^{*(t)} - \bar{\theta}_{\mathcal{A}}^{*(t)}\| + \alpha\overline{\text{SE}}_\omega.$$

As a result, we have

$$\|\bar{\theta}_{\mathcal{A}}^{*(t+1)} - \widehat{\theta}_{\mathcal{A}}^*\| \leq \left(1 - \frac{\alpha\bar{\omega}\lambda_{\min}}{2}\right)\|\bar{\theta}_{\mathcal{A}}^{*(t)} - \widehat{\theta}_{\mathcal{A}}^*\| + \big\{2L_{\max}\alpha\rho + \text{SE}(W)\big\}\|\widehat{\theta}_{\mathcal{A}}^{*(t)} - \bar{\theta}_{\mathcal{A}}^{*(t)}\| + \alpha\sqrt{M}\overline{\text{SE}}_\omega.$$

STEP 2. We next study $\|\widehat{\theta}_{\mathcal{A}}^{*(t+1)} - \bar{\theta}_{\mathcal{A}}^{*(t+1)}\|$. It could be verified that $\|\widehat{\theta}_{\mathcal{A}}^{*(t+1)} - \bar{\theta}_{\mathcal{A}}^{*(t+1)}\|$

$$\leq \|(W \otimes I_p)(\widehat{\theta}_{\mathcal{A}}^{*(t)} - \bar{\theta}_{\mathcal{A}}^{*(t)})\| + \alpha \Big\{ \sum_{m=1}^{M} \|\omega_m \dot{\mathcal{L}}_m(\widetilde{\theta}_{\mathcal{A}}^{(t,m)}) - M^{-1} \sum_{m=1}^{M} \omega_m \dot{\mathcal{L}}_m(\widetilde{\theta}_{\mathcal{A}}^{(t,m)})\|^2 \Big\}^{1/2} + \sqrt{M} \Delta_{(3)}$$

$$\leq \rho\|\widehat{\theta}_{\mathcal{A}}^{*(t)} - \bar{\theta}_{\mathcal{A}}^{*(t)}\| + \alpha\|\dot{\mathcal{L}}_\omega^*(\widetilde{\theta}_{\mathcal{A}}^{*(t)})\| + \sqrt{M}\Delta_{(3)}.$$

Here $\dot{\mathcal{L}}_\omega^*(\widetilde{\theta}_{\mathcal{A}}^{*(t)}) = \{\omega_1 \dot{\mathcal{L}}_1^\top(\widetilde{\theta}_{\mathcal{A}}^{(t,1)}), \ldots, \omega_m \dot{\mathcal{L}}_m^\top(\widetilde{\theta}_{\mathcal{A}}^{(t,m)})\}^\top$. The first inequality holds because $\|\widehat{\theta}_{\mathcal{A}}^{*(t+1)} - \bar{\theta}_{\mathcal{A}}^{*(t+1)}\| \leq \|(W \otimes I_p)\widehat{\theta}_{\mathcal{A}}^{*(t)} - \alpha \dot{\mathcal{L}}_w^*(\widetilde{\theta}_{\mathcal{A}}^{*(t)}) - \bar{\theta}_{\mathcal{A}}^{*(t)} + \alpha I^* M^{-1} \sum_{m=1}^{M} \omega_m \dot{\mathcal{L}}_m(\widetilde{\theta}_{\mathcal{A}}^{(t,m)})\| + \|I^*(M^{-1} \sum_{m=1}^{M} \widetilde{\theta}_{\mathcal{A}}^{(t,m)} - \bar{\theta}_{\mathcal{A}}^{(t)})\|$, and the second inequality holds because $\sum_{m=1}^{M} \|\omega_m \dot{\mathcal{L}}_m(\widetilde{\theta}_{\mathcal{A}}^{(t,m)}) - M^{-1} \sum_{m=1}^{M} \omega_m \dot{\mathcal{L}}_m(\widetilde{\theta}_{\mathcal{A}}^{(t,m)})\|^2 \leq \sum_{m=1}^{M} \|\omega_m \dot{\mathcal{L}}_m(\widetilde{\theta}_{\mathcal{A}}^{(t,m)})\|^2$. Next, note that

$$\|\dot{\mathcal{L}}_m(\widetilde{\theta}_{\mathcal{A}}^{(t,m)})\| \leq 2L_{\max}\|\widetilde{\theta}_{\mathcal{A}}^{(t,m)} - \bar{\theta}_{\mathcal{A}}^{(t)}\| + 2L_{\max}\|\bar{\theta}_{\mathcal{A}}^{(t)} - \widehat{\theta}_{\mathcal{A}}\| + \|\dot{\mathcal{L}}_m(\widehat{\theta}_{\mathcal{A}})\|.$$

Then it could be verified that $\|\dot{\mathcal{L}}_\omega^*(\widetilde{\theta}^{*(t)})\|$

$$\leq \Big\{ 4 \sum_{m=1}^{M} \omega_m^2 L_{\max}^2 \|\widetilde{\theta}_{\mathcal{A}}^{(t,m)} - \bar{\theta}_{\mathcal{A}}^{(t)}\|^2 \Big\}^{1/2} + \Big\{ 4 \sum_{m=1}^{M} \omega_m^2 L_{\max}^2 \|\bar{\theta}_{\mathcal{A}}^{(t)} - \widehat{\theta}_{\mathcal{A}}\|^2 \Big\}^{1/2} + \Big\{ \sum_{m=1}^{M} \omega_m^2 \|\dot{\mathcal{L}}_m(\widehat{\theta}_{\mathcal{A}})\|^2 \Big\}^{1/2}$$

$$\leq 2L_{\max} \Big\{ \sum_{m=1}^{M} \|\widetilde{\theta}_{\mathcal{A}}^{(t,m)} - \bar{\theta}_{\mathcal{A}}^{(t)}\|^2 \Big\}^{1/2} + 2L_{\max}\|\bar{\theta}_{\mathcal{A}}^{*(t)} - \widehat{\theta}_{\mathcal{A}}^*\| + \sqrt{M}\widehat{\mathrm{SE}}.$$

By combining the above results with Lemma C.2, we have $\|\widehat{\theta}_{\mathcal{A}}^{*(t+1)} - \bar{\theta}_{\mathcal{A}}^{*(t)}\|$

$$\leq \rho\|\widehat{\theta}_{\mathcal{A}}^{*(t)} - \bar{\theta}_{\mathcal{A}}^{*(t)}\| + 2\alpha L_{\max}\Big[ \Big\{ \sum_{m=1}^{M} \|\widetilde{\theta}_{\mathcal{A}}^{(t,m)} - \bar{\theta}_{\mathcal{A}}^{(t)}\|^2 \Big\}^{1/2} + \|\bar{\theta}_{\mathcal{A}}^{*(t)} - \widehat{\theta}_{\mathcal{A}}^*\| \Big] + \alpha\sqrt{M}\widehat{\mathrm{SE}} + \sqrt{M}\Delta_{(3)}$$

$$\leq \Big\{ \rho + 2\rho L_{\max}\alpha + \mathrm{SE}(W) \Big\}\|\widehat{\theta}_{\mathcal{A}}^{*(t)} - \bar{\theta}_{\mathcal{A}}^{*(t)}\| + 2\alpha L_{\max}\|\bar{\theta}_{\mathcal{A}}^{*(t)} - \widehat{\theta}_{\mathcal{A}}^*\| + \alpha\sqrt{M}\widehat{\mathrm{SE}}.$$

STEP 3. Finally, we establish the upper bound of $\widehat{\theta}_{\mathcal{A}}^{*(t+1)} - \widehat{\theta}_{\mathcal{A}}^*$. To achieve this, we combine the results from Steps 1 and 2. Let $\bar{\delta}^{*(t+1)} = \|\bar{\theta}_{\mathcal{A}}^{*(t+1)} - \widehat{\theta}_{\mathcal{A}}^*\|$ and $\widehat{\delta}^{*(t+1)} = \|\widehat{\theta}_{\mathcal{A}}^{*(t+1)} - \bar{\theta}_{\mathcal{A}}^{*(t+1)}\|$. Using these definitions, we obtain:

$$\begin{pmatrix} \widehat{\delta}^{*(t+1)} \\ \bar{\delta}^{*(t+1)} \end{pmatrix} \leq \begin{bmatrix} \rho + 2\alpha\rho L_{\max} + \mathrm{SE}(W) & 2\alpha L_{\max} \\ 2\alpha\rho L_{\max} + \mathrm{SE}(W) & 1 - \alpha\bar{\omega}\lambda_{\min}/2 \end{bmatrix} \begin{pmatrix} \widehat{\delta}^{*(t)} \\ \bar{\delta}^{*(t)} \end{pmatrix} + \alpha\sqrt{M} \begin{pmatrix} \widehat{\mathrm{SE}} \\ \mathrm{SE}_\omega \end{pmatrix}. \tag{B.6}$$

Denote

$$\mathbf{H} = [h_{ij}]_{2 \times 2} = \begin{bmatrix} \rho + 2\alpha\rho L_{\max} + \mathrm{SE}(W) & 2\alpha L_{\max} \\ 2\alpha\rho L_{\max} + \mathrm{SE}(W) & 1 - \alpha\bar{\omega}\lambda_{\min}/2 \end{bmatrix},$$

and $\rho_H = \max|\lambda(\mathbf{H})|$ represents the spectral radius of $\mathbf{H}$. By Lemma C.3, we have $0 < \rho_H < 1 - (\alpha\bar{\omega}\lambda_{\min})/8$. Thus, the linear system in (B.6) converges. By recursion and noting that $h_{ij} > 0$ for sufficiently small $\alpha$, we derive the following.

$$\begin{pmatrix} \widehat{\delta}^{*(t+1)} \\ \bar{\delta}^{*(t+1)} \end{pmatrix} \leq \mathbf{H}^{t+1} \begin{pmatrix} \widehat{\delta}^{*(0)} \\ \bar{\delta}^{*(0)} \end{pmatrix} + \alpha\sqrt{M}(I_2 - \mathbf{H})^{-1} \begin{pmatrix} \widehat{\mathrm{SE}} \\ \mathrm{SE}_\omega \end{pmatrix}. \tag{B.7}$$

It could be calculated that,

$$(I_2 - \mathbf{H})^{-1} = c_0 \begin{bmatrix} \alpha\bar{\omega}\lambda_{\min}/2 & 2\alpha L_{\max} \\ 2\alpha\rho L_{\max} + \mathrm{SE}(W) & 1 - \rho - 2\alpha\rho L_{\max} - \mathrm{SE}(W) \end{bmatrix},$$

Here $c_0 > 0$ and

$$\frac{1}{c_0} = \alpha\Big[ \Big\{ 1 - \rho - 2\alpha\rho L_{\max} - \mathrm{SE}(W) \Big\}\frac{\bar{\omega}\lambda_{\min}}{2} - 2L_{\max}\Big\{ 2L_{\max}\alpha\rho + \mathrm{SE}(W) \Big\} \Big]$$

$$\geq \alpha\Big[ \frac{\bar{w}\lambda_{\min}}{2}(1 - \rho) - \frac{(1-\rho)(\bar{w}\lambda_{\min})^2}{16L_{\max}} - 2L_{\max}\frac{(1-\rho)\bar{w}\lambda_{\min}}{8L_{\max}} \Big]$$

$$\geq \alpha\Big( \frac{(1-\rho)\omega\lambda_{\min}}{4} - \frac{(1-\rho)\omega\lambda_{\min}}{8}\frac{\bar{\omega}\lambda_{\min}}{8L_{\max}} \Big) \geq \alpha\frac{(1-\rho)\bar{\omega}\lambda_{\min}}{8},$$

Substituting the above results into equation (B.7), we obtain:

$$\begin{pmatrix}\widehat{\delta}^{*(t+1)} \\ \overline{\delta}^{*(t+1)}\end{pmatrix} \leq \mathbf{H}^{t+1}\begin{pmatrix}\widehat{\delta}^{*(0)} \\ \overline{\delta}^{*(0)}\end{pmatrix} + O\left(\frac{\sqrt{M}}{(1-\rho)\bar{\omega}\lambda_{\min}}\right)\begin{pmatrix}\alpha\big(\bar{\omega}\lambda_{\min}\widehat{\mathrm{SE}}/2 + 2L_{\max}\overline{\mathrm{SE}}_\omega\big) \\ \big\{2\alpha L_{\max} + \mathrm{SE}(W)\big\}\widehat{\mathrm{SE}} + (1-\rho)\overline{\mathrm{SE}}_\omega\end{pmatrix}. \quad \text{(B.8)}$$

Then, for $\rho_\omega = 1 - (\alpha\bar{\omega}\lambda_{\min})/8 \in (\rho_H, 1)$, by Gelfand's formula (Johnson & Horn, 1985, Corollary 5.6.14), there exists some $t_0 \in \mathbb{N}$ such that for all $t \geq t_0$:

$$\|\widehat{\theta}_{\mathcal{A}}^{*(t+1)} - \bar{\theta}_{\mathcal{A}}^{*(t+1)}\| \quad \lesssim \quad \rho_\omega^{t+1}(\|\widehat{\delta}^{*(0)}\| + \|\bar{\delta}^{*(0)}\|) + O\left(\frac{\sqrt{M}}{(1-\rho)\bar{\omega}\lambda_{\min}}\right)\alpha\big(\bar{\omega}\lambda_{\min}\widehat{\mathrm{SE}}/2$$
$$+2L_{\max}\overline{\mathrm{SE}}_\omega\big)$$

$$\|\bar{\theta}_{\mathcal{A}}^{*(t+1)} - \widehat{\theta}_{\mathcal{A}}^*\| \quad \lesssim \quad \rho_\omega^{t+1}(\|\widehat{\delta}^{*(0)}\| + \|\bar{\delta}^{*(0)}\|) + O\left(\frac{\sqrt{M}}{(1-\rho)\bar{\omega}\lambda_{\min}}\right)\Big[\big\{2\alpha L_{\max} + \mathrm{SE}(W)\big\}\widehat{\mathrm{SE}}$$
$$+(1-\rho)\overline{\mathrm{SE}}_\omega\Big]. \quad \text{(B.9)}$$

By reorganizing the results in (B.9), we obtain the following inequality:

$$\|\widehat{\theta}_{\mathcal{A}}^{*(t+1)} - \widehat{\theta}_{\mathcal{A}}^*\| \quad \lesssim \quad 2\rho_\omega^{t+1}(\|\widehat{\delta}^{*(0)}\| + \|\bar{\delta}^{*(0)}\|) + O\left(\frac{\sqrt{M}}{(1-\rho)\lambda_{\min}\bar{\omega}}\right)\Big[\big\{\alpha L_{\max} + \mathrm{SE}(W)\big\}\widehat{\mathrm{SE}}$$
$$+(1-\rho)\overline{\mathrm{SE}}_\omega\Big]. \quad \text{(B.10)}$$

The inequality holds because $(i) : \alpha\bar{\omega}\lambda_{\min}/2 + 2\alpha L_{\max} + \mathrm{SE}(W) \lesssim \big\{2\alpha L_{\max} + \mathrm{SE}(W)\big\}$, and $(ii) : (1 - \rho + 2\alpha L_{\max}) \lesssim (1 - \rho)$. Next, we specify the forms of $\widehat{\mathrm{SE}}$ and $\overline{\mathrm{SE}}_\omega$ to simplify equation (B.10).

STEP 3.1. We first analyze $\overline{\mathrm{SE}}_\omega$, recall that $\bar{\Delta}_2^2 = M^{-1}\sum_{m=1}^M \big\{\omega_m - (1 - a_m)\big\}^2$ we obtain:

$$\overline{\mathrm{SE}}_\omega \quad \leq \quad \|M^{-1}\sum_{m=1}^M (\omega_m - \omega_m^*)\dot{\mathcal{L}}_m(\theta_0)\| + \|M^{-1}\sum_{m=1}^M (\omega_m - \omega_m^*)\big\{\dot{\mathcal{L}}_m(\widehat{\theta}_{\mathcal{A}}) - \dot{\mathcal{L}}_m(\theta_0)\big\}\|$$

$$\leq \quad \|M^{-1}\sum_{m=1}^M (\omega_m - \omega_m^*)\dot{\mathcal{L}}_m(\theta_0)\| + 2\bar{\Delta}_2 L_{\max}\|\widehat{\theta}_{\mathcal{A}} - \theta_0\|.$$

It could be verified that $\|M^{-1}\sum_{m=1}^M \Delta_m \dot{\mathcal{L}}_m(\theta_0)\| = \|M^{-1}\sum_{m=1}^M \Delta_m\big[\dot{\mathcal{L}}_{(m)}(\theta_0) - \mathbb{E}\{\dot{\mathcal{L}}_{(m)}(\theta_0)\}\big] + M^{-1}\sum_{m=1}^M \Delta_m \flat_m\|$.

**Analysis of the second term:** $\|M^{-1}\sum_{m=1}^M \Delta_m \flat_m\| = \|M^{-1}\sum_{m\in\mathcal{A}} \omega_m \flat_m\| \leq M^{-1}\sum_{m\in\mathcal{A}} \omega_m \|\flat_m\|$.

**Analysis of the first term:** Note that $M^{-1}\sum_{m=1}^M \Delta_m\big[\dot{\mathcal{L}}_{(m)}(\theta_0) - \mathbb{E}\{\dot{\mathcal{L}}_{(m)}(\theta_0)\}\big] = M^{-1}\sum_{m\in\mathcal{A}} \omega_m\big[\dot{\mathcal{L}}_{(m)}(\theta_0) - \mathbb{E}\{\dot{\mathcal{L}}_{(m)}(\theta_0)\}\big] + M^{-1}\sum_{m\notin\mathcal{A}} \Delta_m \dot{\mathcal{L}}_{(m)}(\theta_0)$. First, under events $\bigcap_{m\in\mathcal{A}} \mathcal{E}_{3,m}$, we have $\|M^{-1}\sum_{m\in\mathcal{A}} \omega_m\big[\dot{\mathcal{L}}_{(m)}(\theta_0) - \mathbb{E}\{\dot{\mathcal{L}}_{(m)}(\theta_0)\}\big]\| \leq M^{-1}\sum_{m\in\mathcal{A}} \omega_m \|\flat_m\|$. Second, it could be proved that under events $\mathcal{E}_4$,

$$\|M^{-1}\sum_{m\notin\mathcal{A}} \Delta_m \dot{\mathcal{L}}_{(m)}(\theta_0)\| \lesssim \frac{M - |\mathcal{A}|}{M}\left(\frac{\log(N - n|\mathcal{A}|)}{(N - n|\mathcal{A}|)}\right)^{1/2}\bar{\Delta}_{\mathcal{A}}^c.$$

Here $\bar{\Delta}_{\mathcal{A}}^c = \big\{(M - |\mathcal{A}|)^{-1}\sum_{m\notin\mathcal{A}}(\omega_m - \omega^*)^2\big\}^{1/2}$. Combining the above results, we have

$$\overline{\mathrm{SE}}_\omega \quad \leq \quad M^{-1}\sum_{m\in\mathcal{A}} \omega_m \|\flat_m\| + O\left(\frac{\sqrt{\log(N - n|\mathcal{A}|)}}{\sqrt{N}}\big\{M^{-1}\sum_{m\notin\mathcal{A}}(\omega_m - \omega^*)^2\big\}^{1/2}\right)$$
$$+2\bar{\Delta}_2 L_{\max}\|\widehat{\theta}_{\mathcal{A}} - \theta_0\|.$$

STEP 3.2. We next study $\widehat{\text{SE}}$. By the Cauchy-Schwarz inequality, we have

$$
\widehat{\text{SE}}^2 \;\lesssim\; \Big\{ M^{-1} \sum_{m=1}^{M} \big\| \dot{\mathcal{L}}_{(m)}(\widehat{\theta}_{\mathcal{A}}) - \dot{\mathcal{L}}_{(m)}(\theta_0) \big\|^2 + M^{-1} \sum_{m=1}^{M} \big\| \dot{\mathcal{L}}_{(m)}(\theta_0) - \mathbb{E}\dot{\ell}(x,y;\theta_0) \big\|^2
$$

$$
+ M^{-1} \sum_{m=1}^{M} \big\| \mathbb{E}\dot{\ell}(x,y;\theta_0) \big\|^2 \Big\}.
$$

We study the three terms separately. First, under events $\cap_m \mathcal{E}_{m,1}$, we have $M^{-1} \sum_{m=1}^{M} \big\| \dot{\mathcal{L}}_{(m)}(\widehat{\theta}_{\mathcal{A}}) - \dot{\mathcal{L}}_{(m)}(\theta_0) \big\|^2 \leq 4 L_{\max}^2 \|\widehat{\theta}_{\mathcal{A}} - \theta_0\|^2$. Second, under event $\mathcal{E}_5$, we have $M^{-1} \sum_{m=1}^{M} \big\| \dot{\mathcal{L}}_{(m)}(\theta_0) - \mathbb{E}\dot{\ell}(x,y;\theta_0) \big\|^2 \leq \log n / n$. Finally, it is obvious to show that $M^{-1} \sum_{m=1}^{M} \| \mathbb{E}\dot{\ell}(x,y;\theta_0) \|^2 = M^{-1} \sum_{m \in \mathcal{A}} \|\flat_m\|^2$. Combining the above results, we have

$$
\widehat{\text{SE}} \;\lesssim\; L_{\max} \|\widehat{\theta}_{\mathcal{A}} - \theta_0\| + \frac{\sqrt{\log n}}{\sqrt{n}} + \frac{\sqrt{|\mathcal{A}|}}{\sqrt{M}} \Big\{ \frac{1}{|\mathcal{A}|} \sum_{m \in \mathcal{A}} \|\flat_m\|^2 \Big\}^{1/2}.
$$

Recall that $\overline{\omega}_2^{\mathcal{A}} = \big( |\mathcal{A}|^{-1} \sum_{m \in \mathcal{A}} \omega_m^2 \big)^{1/2}$, $\bar{\flat}_2^{\mathcal{A}} = \big( |\mathcal{A}|^{-1} \sum_{m \in \mathcal{A}} \|\flat_m\|^2 \big)^{1/2}$. Substituting the result obtained from Step 3 into equation (B.10), and define $\widehat{\delta}_0 = \max_m \|\widehat{\theta}_{\mathcal{A}}^{(0,m)} - \widehat{\theta}_{\mathcal{A}}\|$, we obtain

$$
M^{-1/2} \|\widehat{\theta}_{\mathcal{A}}^{*(t)} - \widehat{\theta}_{\mathcal{A}}^{*}\| \;\lesssim\; \rho_\omega^t \widehat{\delta}_0 + \frac{\alpha L_{\max} + \text{SE}(W)}{(1-\rho)\lambda_{\min}\bar{\omega}} \Big\{ \Big( \frac{\log n}{n} \Big)^{1/2} L_{\max} + \Big( \frac{|\mathcal{A}|}{M} \Big)^{1/2} \bar{\flat}_2^{\mathcal{A}} \Big\}
$$

$$
+ (\bar{\omega}\lambda_{\min})^{-1} \Big\{ \frac{|\mathcal{A}|}{M} \bar{\flat}_2^{\mathcal{A}} \overline{\omega}_2^{\mathcal{A}} + \bar{\Delta}_2 \Big( \frac{\log N}{N} \Big)^{1/2} + L_{\max} \bar{\Delta}_2 \|\widehat{\theta}_{\mathcal{A}} - \theta_0\| \Big\}.
$$

The first inequality holds because it could be proved that $\|\widehat{\theta}_{\mathcal{A}} - \theta_0\| \lesssim (\log N / N)^{1/2}$ with probability at least $1 - O(1/(\log N)^4)$ by Lemma 6 in Zhang et al. (2013). Furthermore, as $n \to \infty$, applying Markov's inequality readily demonstrates that

$$
M^{-1/2} \lim_{t \to \infty} \|\widehat{\theta}_{\mathcal{A}}^{*(t)} - I^* \widehat{\theta}_{\mathcal{A}}\| \;\lesssim\; \frac{1}{\bar{\omega}} \Bigg[ \Big\{ \alpha + \text{SE}(W) \Big\} \Big\{ \frac{1}{\sqrt{n}} + \Big( \frac{|\mathcal{A}|}{M} \Big)^{1/2} \Big\} + \Big( \frac{|\mathcal{A}|}{M} \overline{\omega}_2^{\mathcal{A}} + \frac{\bar{\Delta}_2}{\sqrt{N}} \Big) \Bigg]
$$

with probability tending to 1. The disappearance of $\log n$ and $\log N$ occurs because Markov's inequality can be directly applied in this context to derive upper bounds for $\|\widehat{\theta}_{\mathcal{A}} - \theta_0\|$, $M^{-1} \sum_{m=1}^{M} \| \dot{\mathcal{L}}_{(m)}(\theta_0) - \mathbb{E}\dot{\ell}(x,y;\theta_0) \|^2$, and $\|M^{-1} \sum_{m \in \mathcal{A}} \Delta_m \dot{\mathcal{L}}_{(m)}(\theta_0)\|$. This finishes the theorem proof.

**Proof of Proposition A.1**: The proof of Proposition A.1 can be found in Wu et al. (2023a). Thus, we omit the details here.

B.4 PROOF OF THEOREM A.3

With a slight abuse of notation, we define (1) $\bar{\omega} = M^{-1} \sum_{m=1}^{M} \widehat{\omega}_m$, (2) $(\overline{\omega}_2^{\mathcal{A}})^2 = |\mathcal{A}|^{-1} \sum_{m \in \mathcal{A}} \widehat{\omega}_m^2$, and (3) $\bar{\Delta}_2^2 = M^{-1} \sum_{m=1}^{M} \big\{ \widehat{\omega}_m - (1 - a_m) \big\}^2$. Furthermore, note that $\widehat{\theta}_{\text{init}}^{(m)} = \widehat{\theta}^{(\infty,m)}$ corresponds to the standard DFL estimator, which we simply denote as $\widehat{\theta}^{(m)}$ throughout this section. By applying the sample-splitting technique (Balakrishnan et al., 2017; Chernozhukov et al., 2018), we can separately study the training process and the estimation of $\widehat{\omega}_m$. As a result, it remains to analyze (1) – (3) in the following 3 parts, respectively.

PART 1. We first investigate $(\overline{\omega}_2^{\mathcal{A}})^2$. Recall that $\widehat{\theta}$ denotes the whole sample estimator. By definition, we have

$$
(\overline{\omega}_2^{\mathcal{A}})^2 = \frac{1}{|\mathcal{A}|} \sum_{m \in \mathcal{A}} \exp\big( -2\lambda_n \|\dot{\mathcal{L}}_m(\widehat{\theta}^{(m)})\| \big) \leq \frac{1}{|\mathcal{A}|} \sum_{m \in \mathcal{A}} \Big| \exp\big( -2\lambda_n \|\dot{\mathcal{L}}_m(\widehat{\theta}^{(m)})\| \big)
$$

$$
- \exp\big( -2\lambda_n \|\dot{\mathcal{L}}_m(\widehat{\theta})\| \big) \Big| + \frac{1}{|\mathcal{A}|} \sum_{m \in \mathcal{A}} \exp\big( -2\lambda_n \|\dot{\mathcal{L}}_m(\widehat{\theta})\| \big) = \Delta_1^{(1)} + \Delta_2^{(1)}.
$$

Note that by mean value theorem, Lemma C.1, and under the events $\bigcap_m \mathcal{E}_{1,m}$, the following holds with probability at least $1 - O\left(M/n^4\right)$.

$$
\begin{aligned}
\Delta_1^{(1)} &\leq 2\lambda_n \frac{1}{|\mathcal{A}|} \sum_{m \in \mathcal{A}} \exp(-\xi_m)\Big|\|\dot{\mathcal{L}}_m(\widehat{\theta}^{(m)})\| - \|\dot{\mathcal{L}}_m(\widehat{\theta})\|\Big| \\
&\leq 2\lambda_n \frac{1}{|\mathcal{A}|} \sum_{m \in \mathcal{A}} 2L_{\max}\|\widehat{\theta}^{(m)} - \widehat{\theta}\| \leq 4L_{\max}\lambda_n \sqrt{\frac{1}{|\mathcal{A}|} \sum_{m \in \mathcal{A}} \|\widehat{\theta}^{(m)} - \widehat{\theta}\|^2} \\
&\leq 4L_{\max}\lambda_n \sqrt{\frac{M}{|\mathcal{A}|}} \sqrt{\frac{1}{M} \sum_{m=1}^{M} \|\widehat{\theta}^{(m)} - \widehat{\theta}\|^2} \lesssim \lambda_n \left(\frac{M}{|\mathcal{A}|}\right)^{1/2} \{\alpha + \mathrm{SE}(W)\}.
\end{aligned}
$$

Here $\xi_m$ is some positive constant for any $1 \leq m \leq M$ and $\exp(-\xi_m) \leq 1$. The second inequality holds under assumption 4. The last equality holds by Proposition A.1. Then we can set a sufficiently small $\alpha$ and $\mathrm{SE}(W)$ such that $(\lambda_n/\bar{\omega})\left(M/|\mathcal{A}|\right)^{1/2}\{\alpha + \mathrm{SE}(W)\} \ll M^2/(|\mathcal{A}|^2 N)$. As a result, $\Delta_1^{(1)}$ is an ignorable higher order term.

We next investigate $\Delta_2^{(1)}$. Recall that $\theta_{\mathcal{A}} = arg\,min_\theta M^{-1} \sum_{m=1}^{M} e_m^{\mathcal{A}}(\theta)$ represents pseudo-true parameter and $\theta_m = arg\,min_\theta \mathbb{E}_m \ell(X_i, Y_i; \theta)$. Then by triangle-inequality, we have

$$
\begin{aligned}
\Delta_2^{(1)} &\leq \frac{1}{|\mathcal{A}|} \sum_{m \in \mathcal{A}} \exp\left(-2\lambda_n \|\dot{\mathcal{L}}_m(\theta_{\mathcal{A}}) - \dot{\mathcal{L}}_m(\theta_m)\|\right) \times \exp\left(2\lambda_n \|\dot{\mathcal{L}}_m(\theta_m)\|\right) \\
&\quad \times \exp\left(2\lambda_n \|\dot{\mathcal{L}}_m(\widehat{\theta}) - \dot{\mathcal{L}}_m(\theta_{\mathcal{A}})\|\right) \\
&\leq \left\{\frac{1}{|\mathcal{A}|} \sum_{m \in \mathcal{A}} \exp\left(-6\lambda_n \|\dot{\mathcal{L}}_m(\theta_{\mathcal{A}}) - \dot{\mathcal{L}}_m(\theta_m)\|\right)\right\}^{1/3} \left\{\frac{1}{|\mathcal{A}|} \sum_{m \in \mathcal{A}} \exp\left(6\lambda_n \|\dot{\mathcal{L}}_m(\theta_m)\|\right)\right\}^{1/3} \\
&\quad \times \left\{\frac{1}{|\mathcal{A}|} \sum_{m \in \mathcal{A}} \exp\left(6\lambda_n \|\dot{\mathcal{L}}_m(\widehat{\theta}) - \dot{\mathcal{L}}_m(\theta_{\mathcal{A}})\|\right)\right\}^{1/3} = \Delta_1^{(2)} \Delta_2^{(2)} \Delta_3^{(2)}. \quad (B.11)
\end{aligned}
$$

The second inequality holds by Cauchy-Schwartz inequality. We then study the three terms in equation (B.11) separately.

**Analysis of $\Delta_1^{(2)}$:** Define good events: $\mathcal{E}_{2,m} = \left\{\|\ddot{\mathcal{L}}_{(m)}(\theta) - \Omega_{\mathcal{A}}(\theta)\| \leq \lambda_{\min}/4\right\}$. Using similar techniques in Lemma C.1, it is easy to prove that $\mathbb{P}(\bigcup_{m \in \mathcal{A}} \mathcal{E}_{2,m}^c) \lesssim |\mathcal{A}| L_{\max}^8/(\lambda_{\min}^8 n^4)$. Further define good event: $\mathcal{E}_6 = \left\{\|\theta_{\mathcal{A}} - \theta_0\| \leq \min_{m \in \mathcal{A}} \|\theta_m - \theta_0\|/2\right\}$. Using equation (B.4) in Appendix B.2, we can show that $\mathcal{E}_6$ holds when $M \geq 2|\mathcal{A}|\|\bar{b}_{\mathcal{A}}\|/(\min_{m \in \mathcal{A}} \|\theta_m - \theta_0\|\lambda_{\min})$. Then it suffices to study $\Delta_1^{(2)}$ under events $\bigcap \mathcal{E}_{2,m} \bigcap \mathcal{E}_6$. It could be verified that with probability at least $1 - O(M/n^4)$,

$$
\Delta_1^{(2)} \leq \left[\frac{1}{|\mathcal{A}|} \sum_{m \in \mathcal{A}} \exp\left\{-3\lambda_n(\lambda_{\min}/2)\|\theta_m - \theta_{\mathcal{A}}\|\right\}\right]^{1/3} \leq \exp\left\{-\frac{\lambda_n \lambda_{\min}}{2} \min_{m \in \mathcal{A}} \|\theta_m - \theta_0\|/2\right\}.
$$

**Analysis of $\Delta_2^{(2)}$:** Define good events: $\mathcal{E}_{3,m}^* = \left\{\lambda_n \|\dot{\mathcal{L}}_{(m)}(\theta_m)\| \leq C\right\}$ for some sufficient large $C > 0$. Note that $\mathbb{E}\{\dot{\mathcal{L}}_{(m)}(\theta_m)\} = 0$. Then using similar techniques in proof of Lemma C.1 and under Assumptions $1 - 4$, we have

$$
\mathbb{P}\left(\bigcup_{m \in \mathcal{A}} (\mathcal{E}_{3,m}^*)^c\right) \leq |\mathcal{A}| \frac{\lambda_n^8}{C^8 n^4}.
$$

Consequently, as long as $\lambda_n \lesssim \sqrt{n}(|\mathcal{A}|)^{-1/8}$, we have $\max_m \lambda_n \|\dot{\mathcal{L}}_m(\theta_m)\| = O(1)$ under $\bigcap_{m \in \mathcal{A}} \mathcal{E}_{3,m}^*$, this leads to $\Delta_2^{(2)} = O(1)$ with probability at least $1 - O(M/n^4)$.

**Analysis of $\Delta_3^{(2)}$:** Under events $\bigcap \mathcal{E}_{1,m}$ defined in (C.1), it could be verified that

$$
\Delta_3^{(2)} \leq \left[\frac{1}{|\mathcal{A}|} \sum_{m \in \mathcal{A}} \exp\left\{12\lambda_n L_{\max}\|\widehat{\theta} - \theta_{\mathcal{A}}\|\right\}\right]^{1/3} = \exp\left\{4\lambda_n L_{\max}\|\widehat{\theta} - \theta_{\mathcal{A}}\|\right\}.
$$

The first inequality holds under Assumption 4. As a result, as long as $\lambda_n \lesssim \sqrt{N}$, by equation (B.1) in Appendix B.2, we have $\Delta_3^{(2)} = O(1)$ with probability at least $1 - O(M/n^4)$.

Combining the above results, we know that as long as $\lambda_n \lesssim \sqrt{n}M^{-1/8}$, we have $\overline{\omega}_2^{\mathcal{A}} \lesssim \exp\Big( - \lambda_n\lambda_{\min}\min\|\theta_m - \theta_0\|/8 \Big)$ with probability at least $1 - O(M/n^4)$, this finishes the first part.

PART 2. We next study $\bar{\Delta}_2^2 = M^{-1}\sum_{m=1}^M \{\widehat{\omega}_m - (1 - a_m)\}^2$. Define $\mathcal{G} = \mathcal{A}^c$ represents the indices of the trustworthy clients. By definition, we have

$$\bar{\Delta}_2^2 = \frac{1}{M}\sum_{m\in\mathcal{A}}\widehat{\omega}_m^2 + \frac{1}{M}\sum_{m\in\mathcal{G}}(\widehat{\omega}_m - 1)^2.$$

First, by PART 1, we have $M^{-1}\sum_{m\in\mathcal{A}}\widehat{\omega}_m^2 \lesssim |\mathcal{A}|M^{-1}\exp\big( - \lambda_n\lambda_{\min}\min\|\theta_m - \theta_0\|/4 \big)$. Then it remains to study $M^{-1}\sum_{m\in\mathcal{G}}(\widehat{\omega}_m - 1)^2$.

By Cauchy-Schwartz inequality, it could be proved that $M^{-1}\sum_{m\in\mathcal{G}}(\widehat{\omega}_m - 1)^2$

$$
\begin{aligned}
&= \frac{1}{M}\sum_{m\in\mathcal{G}}\Big\{\exp(-\lambda_n\|\dot{\mathcal{L}}_m(\widehat{\theta}^{(m)})\|) - 1\Big\}^2 \\
&\leq \frac{2}{M}\sum_{m\in\mathcal{G}}\Big[\Big\{\exp\big( - \lambda_n\|\dot{\mathcal{L}}_m(\widehat{\theta}^{(m)})\|\big) - \exp\big( - \lambda_n\|\dot{\mathcal{L}}_m(\theta_0)\|\big)\Big\}^2 \\
&\quad + \Big\{\exp\big( - \lambda_n\|\dot{\mathcal{L}}_m(\theta_0)\|\big) - \exp\big( - \lambda_n\|\mathbb{E}\dot{\mathcal{L}}_m(\theta_0)\|\big)\Big\}^2\Big] \\
&\leq \frac{4}{M}\sum_{m\in\mathcal{G}}\Big\{\exp\big( - \xi_m\big)\lambda_n\big|\|\dot{\mathcal{L}}_m(\widehat{\theta}^{(m)})\| - \|\dot{\mathcal{L}}_m(\theta_0)\|\big|\Big\} + \frac{4}{M}\sum_{m\in\mathcal{G}}\Big\{\exp(-\xi_m)\lambda_n\|\dot{\mathcal{L}}_m(\theta_0)\|\Big\}.
\end{aligned}
$$
(B.12)

Here $\xi_m$ is some positive constant for any $1 \leq m \leq M$ and $\exp(-\xi_m) \leq 1$. The first inequality holds because $\exp(-\lambda_n\|\mathbb{E}(\dot{\mathcal{L}}_m(\theta_0))\|) = 1$. The second inequality holds because (B.12) is upper bounded by $4M^{-1}\sum_{m\in\mathcal{G}}\Big[\big|\exp(-\lambda_n\|\dot{\mathcal{L}}_m(\widehat{\theta}^{(m)})\|) - \exp(-\lambda_n\|\dot{\mathcal{L}}_m(\theta_0)\|)\big| + \big|\exp(-\lambda_n\|\dot{\mathcal{L}}_m(\theta_0)\|) - \exp(-\lambda_n\|\mathbb{E}\dot{\mathcal{L}}_m(\theta_0)\|)\big|\Big]$. The last inequality holds by mean value theorem. Then under good events $\cap_{m\in\mathcal{G}}\mathcal{E}_{1,m}$, it could be proved that

$$\frac{1}{M}\sum_{m\in\mathcal{G}}(\omega_m - 1)^2 \leq 8L_{\max}\lambda_n\Big\{\frac{1}{M}\sum_{m\in\mathcal{G}}\|\widehat{\theta}^{(m)} - \widehat{\theta}\| + \|\widehat{\theta} - \theta_0\|\Big\} + \frac{4\lambda_n}{M}\sum_{m\in\mathcal{G}}\|\dot{\mathcal{L}}_m(\theta_0)\|.$$

First, note that by Markov's inequality, we have $M^{-1}\sum_{m\in\mathcal{G}}\big(\|\dot{\mathcal{L}}_m(\theta_0)\| - \mathbb{E}\|\dot{\mathcal{L}}_m(\theta_0)\|\big) \lesssim (\sqrt{\log N}/\sqrt{N})$ with probability at least $1 - O(1/\log N)$. Assume $\log N \lesssim M$, this yields

$$\frac{1}{M}\sum_{m\in\mathcal{G}}\|\dot{\mathcal{L}}_m(\theta_0)\| \lesssim \mathbb{E}\|\dot{\mathcal{L}}_{m\in\mathcal{G}}(\theta_0)\| + \Big(\frac{\log N}{N}\Big)^{1/2} \lesssim \frac{1}{\sqrt{n}}.$$

As a results, as long as $\lambda_n \lesssim \sqrt{n}$, which holds under the assumption $\lambda_n \lesssim \sqrt{n}M^{-1/8}$ in PART 1, we have $4\lambda_n M^{-1}\sum_{m\in\mathcal{G}}\|\dot{\mathcal{L}}_m(\theta_0)\| \lesssim \lambda_n/\sqrt{n}$. Furthermore, as long as $\alpha + \widehat{\text{SE}}(W)$ are sufficiently small such that $\lambda_n\{\alpha + \text{SE}(W)\} = o(1)$, by Proposition A.1, we can obtain that $\lambda_n M^{-1}\sum_{m\in\mathcal{G}}\|\widehat{\theta}^{(m)} - \widehat{\theta}\| \leq \lambda_n\sqrt{M^{-1}\sum_{m=1}^M\|\widehat{\theta}^{(m)} - \widehat{\theta}\|^2}$ is an ignorable higher order term.

Combining the above results, we know that as long as $\lambda_n \lesssim \sqrt{n}M^{-1/8}$, we have $\bar{\Delta}_2^2 \lesssim |\mathcal{A}|M^{-1}\exp\big( - \lambda_n\lambda_{\min}\min\|\theta_m - \theta_0\|/4 \big) + \lambda_n/\sqrt{n} + \lambda_n\|\widehat{\theta} - \theta_0\|$ with probability at least $1 - O(M/n^4 + 1/\log N)$, this finishes the second part.

PART 3. Finally, we next study $\bar{\omega} = M^{-1}\sum_{m=1}^M\widehat{\omega}_m$. Similar to the analysis in the previous two parts, since we can make the $\widehat{\theta}^{(m)}$ arbitrarily close to $\widehat{\theta}$ by choosing a sufficiently small $\alpha + \text{SE}(W)$,

it remains to study $M^{-1} \sum_{m=1}^{M} \exp(-\lambda_n \|\dot{\mathcal{L}}_m(\widehat{\theta})\|)$. It could be verified that under good events $\cap_{m \in \mathcal{G}} \mathcal{E}_{1,m}$, we have

$$M^{-1} \sum_{m=1}^{M} \exp(-\lambda_n \|\dot{\mathcal{L}}_m(\widehat{\theta})\|) \geq \frac{|\mathcal{G}|}{M} \frac{1}{|\mathcal{G}|} \sum_{m \in \mathcal{G}} \exp(-\lambda_n \|\dot{\mathcal{L}}_m(\widehat{\theta})\|)$$

$$\geq \frac{|\mathcal{G}|}{M} \left\{ \frac{1}{|\mathcal{G}|} \sum_{m \in \mathcal{G}} \exp(-\lambda_n \|\dot{\mathcal{L}}_m(\widehat{\theta}) - \dot{\mathcal{L}}_m(\theta_0)\|) \exp(\lambda_n \|\dot{\mathcal{L}}_m(\theta_0)\|) \right\}$$

$$\geq \frac{|\mathcal{G}|}{M} \exp(-2L_{\max}\lambda_n \|\widehat{\theta} - \theta_0\|) \frac{1}{|\mathcal{G}|} \sum_{m \in \mathcal{G}} \exp(\lambda_n \|\dot{\mathcal{L}}_m(\theta_0)\|) \geq \frac{|\mathcal{G}|}{M} \exp\left(-2L_{\max}\lambda_n \|\widehat{\theta} - \theta_0\|\right).$$

As a result, $\frac{1}{\bar{\omega}} \lesssim \exp\left(2L_{\max}\lambda_n \|\widehat{\theta} - \theta_0\|\right)$ with probability at least $1 - O(M/n^4)$. This finishes the third part.

Combining the results from the three parts, we know that when $\lambda_n \lesssim \sqrt{n} M^{-1/8}$, we have with probability at least $1 - O(M/n^4 + 1/\log N)$: (1) $(|\mathcal{A}|^{-1} \sum_{m \in \mathcal{A}} \widehat{\omega}_m^2)^{1/2} \lesssim \exp\left(-\lambda_n \lambda_{\min} \min_{m \in \mathcal{A}} \|\theta_m - \theta_0\|/8\right)$, (2) $M^{-1} \sum_{m=1}^{M} \left\{\widehat{\omega}_m - (1 - a_m)\right\}^2 \lesssim |\mathcal{A}| M^{-1} \exp\left(-\lambda_n \lambda_{\min} \min \|\theta_m - \theta_0\|/4\right) + \lambda_n/\sqrt{n} + \lambda_n \|\widehat{\theta} - \theta_0\|$, and (3) $(M^{-1} \sum_{m=1}^{M} \widehat{\omega}_m)^{-1} \lesssim \exp\left(2L_{\max}\lambda_n \|\widehat{\theta} - \theta_0\|\right)$. Substituting the above equations into (4.3), let $\alpha + \mathrm{SE}(W) = O(1/\sqrt{N})$, there exists some positive constants $c_1, c_2$ such that $M^{-1/2} \|\widehat{\theta}_{\mathcal{A}}^{*(\infty)} - I^* \widehat{\theta}_{\mathcal{A}}\|$ is upper bounded with probability tending to 1 by

$$C \exp\left(c_1 \times \lambda_n \|\widehat{\theta} - \theta_0\|\right) \left\{ o_p\left(1/\sqrt{N}\right) + o_p\left\{\exp\left(-c_2 \times \lambda_n\right)\right\} + \frac{\lambda_n}{\sqrt{N}} \|\widehat{\theta} - \theta_0\| \right\}.$$

### B.5 PROOF OF COROLLARY 4.1

The proof of the corollary 4.1 is similar to that of Theorem A.3 by replacing $\widehat{\theta}^{(m)}$ with $\widehat{\theta}_{\mathrm{init}}^{(m)}$ and $\widehat{\theta}$ with $\bar{\theta}_{\mathrm{init}}$. Thus, we omit the detailed proof here.

### B.6 PROOF OF THEOREM 3.1

Similarly to the proof of Theorem A.1, to prove Theorem 3.1, we aim to verify that (1) The parameter $\theta_{\mathcal{A}}$ lies in a neighborhood of $\theta_0$. (2) Loss function $M^{-1} \sum_{m=1}^{M} e_m^{\mathcal{A}}(\theta)$ is strongly convex at $\theta_{\mathcal{A}}$. Once these conditions are established, we can mimic Steps 1 and 2 of Proof B.2 to complete the proof of the theorem.

**Proof of (1):** Recall that $M^{-1} \sum_{m=1}^{M} \dot{e}_m^{\mathcal{A}}(\theta_{\mathcal{A}}) = 0$, and $M^{-1} \sum_{m=1}^{M} \dot{e}_m^{\mathcal{A}}(\theta_0) = \varrho \bar{b}_{\mathcal{A}}$ from equation (B.3). Applying the integral form of the mean value theorem, we obtain:

$$M^{-1} \sum_{m=1}^{M} \left\{ \dot{e}_m^{\mathcal{A}}(\theta_0) - \dot{e}_m^{\mathcal{A}}(\theta_{\mathcal{A}}) \right\} = M^{-1} \sum_{m=1}^{M} \int_0^1 \ddot{e}_m^{\mathcal{A}}(\theta_0 + t(\theta_{\mathcal{A}} - \theta_0))(\theta_0 - \theta_{\mathcal{A}})dt = \varrho \bar{b}_{\mathcal{A}}.$$

Denote $\Delta_\theta = \theta_{\mathcal{A}} - \theta_0$, and $\Omega_{\mathrm{avg}} = M^{-1} \sum_{m=1}^{M} \int_0^1 \ddot{e}_m^{\mathcal{A}}(\theta_0 + t\Delta_\theta)dt$. Then the above relation simplifies to:

$$\varrho \bar{b}_{\mathcal{A}} = -\Omega_{\mathrm{avg}} \Delta_\theta. \tag{B.13}$$

We now analyze $\Omega_{\mathrm{avg}}$. First, for each $m \in \mathcal{A}$ and any vector $x \in \mathbb{R}^p$ with $x^\top x = 1$, by Assumption 3, we have

$$x^\top \left( \int_0^1 \ddot{e}_m^{\mathcal{A}}(\theta_0 + t\Delta_\theta)dt \right) x = \int_0^1 x^\top \ddot{e}_m^{\mathcal{A}}(\theta_0 + t\Delta_\theta)x \, dt \geq 0.$$

Similarly, for each $m \notin \mathcal{A}$, using Assumption 3 again, we have

$$x^\top \left( \int_0^1 \ddot{e}_m^{\mathcal{A}}(\theta_0 + t\Delta_\theta)dt \right) x = \int_0^1 x^\top \Omega_m(\theta_0 + t\Delta_\theta)x \, dt.$$

Furthermore, according to Assumptions 3, 4, and Weyl's inequality, for any $\theta \in \mathbb{R}^p$, it follows that for each $m \notin \mathcal{A}$:

$$\lambda_{\min}(\Omega_m(\theta)) \geq \lambda_{\min} - L_{\max}||\theta - \theta_0||.$$

In particular, if $||\theta - \theta_0|| \leq \lambda_{\min}/(2rL_{\max})$, then $\lambda_{\min}(\Omega_m(\theta)) \geq \lambda_{\min}/2$. Hence, we have for $m \notin \mathcal{A}$:

$$
\begin{aligned}
x^\top \Big( \int_0^1 \ddot{e}_m^{\mathcal{A}}(\theta_0 + t\Delta_\theta)dt \Big)x &= \int_0^{\frac{\lambda_{\min}}{2rL_{\max}}} x^\top \Omega_m(\theta_0 + t\Delta_\theta)xdt + x^\top \int_{\frac{\lambda_{\min}}{2rL_{\max}}}^1 x^\top \Omega_m(\theta_0 + t\Delta_\theta)xdt \\
&\geq \quad \frac{\lambda_{\min}}{2}\frac{\lambda_{\min}}{2rL_{\max}} + 0 = \frac{\lambda_{\min}^2}{4rL_{\max}}.
\end{aligned}
$$

Combining the results above, the following equation holds for any $x \in \mathbb{R}^p$ satisfying $x^\top x = 1$,

$$
\begin{aligned}
x^\top \Omega_{\text{avg}} x &= M^{-1} \sum_{m \notin \mathcal{A}} x^\top \Big( \int_0^1 \ddot{e}_m^{\mathcal{A}}(\theta_0 + t\Delta_\theta)dt \Big)x + \sum_{m \in \mathcal{A}} x^\top \Big( \int_0^1 \ddot{e}_m^{\mathcal{A}}(\theta_0 + t\Delta_\theta)dt \Big)x \\
&\geq \quad (1-\varrho)\frac{\lambda_{\min}^2}{4rL_{\max}} + 0 = (1-\varrho)\frac{\lambda_{\min}^2}{4rL_{\max}}.
\end{aligned}
$$

This yields $\lambda_{\min}(\Omega_{\text{avg}}) \geq (1-\varrho)\lambda_{\min}^2/(4rL_{\max})$. Substituting this result back into equation (B.13), we obtain:

$$||\theta_{\mathcal{A}} - \theta_0|| = ||\Delta_\theta|| \leq \lambda_{\min}^{-1}(\Omega_{\text{avg}})||\varrho\bar{b}_{\mathcal{A}}|| \leq \frac{4rL_{\max}}{\lambda_{\min}^2}(1-\varrho)^{-1}\varrho||\bar{b}_{\mathcal{A}}||. \tag{B.14}$$

This completes the proof of (1).

**Proof of (2):** It could be proved that under Assumption 3, we have

$$\lambda_{\min}\Big( M^{-1} \sum_{m=1}^M \ddot{e}_m^{\mathcal{A}}(\theta_0) \Big) \geq M^{-1} \sum_{m \in \mathcal{A}} \lambda_{\min} = (1-\varrho)\lambda_{\min}.$$

In addition, for any $\theta$, it could be shown that

$$
\begin{aligned}
\lambda_{\min}\Big( M^{-1} \sum_{m=1}^M \ddot{e}_m^{\mathcal{A}}(\theta_{\mathcal{A}}) \Big) &\geq \lambda_{\min}\Big( M^{-1} \sum_{m=1}^M \ddot{e}_m^{\mathcal{A}}(\theta_0) \Big) - L_{\max}||\theta_{\mathcal{A}} - \theta_0|| \\
&\geq (1-\rho)\lambda_{\min} - \frac{4rL_{\max}^2}{\lambda_{\min}^2}(1-\varrho)^{-1}\varrho||\bar{b}_{\mathcal{A}}||.
\end{aligned}
$$

Assume that $\varrho$ is sufficiently small, such that $\varrho \leq \{8L_{\max}^2 r^2\}^{-1}\{(1-\rho)^2\lambda_{\min}^3\}$. Then we have

$$\lambda_{\min}\Big( M^{-1} \sum_{m=1}^M \ddot{e}_m^{\mathcal{A}}(\theta_{\mathcal{A}}) \Big) \geq (1-\rho)\lambda_{\min}/2. \tag{B.15}$$

Applying equations (B.14) and (B.15). Using similar techniques as those employed in Appendix B.2, we have

$$V(\widehat{\theta}) \lesssim \frac{L_{\max}^2}{(1-\rho)^2\lambda_{\min}^2 N} + O\Big(\frac{1}{N^2}\Big).$$

$$NV(\widehat{\theta}) \to \text{tr}\big\{\Omega_{\mathcal{A}}^{-1}(\theta_{\mathcal{A}})\Sigma_{\mathcal{A}}(\theta_{\mathcal{A}})\Omega_{\mathcal{A}}^{-1}(\theta_{\mathcal{A}})\big\} \text{ as } N \to \infty.$$

$$\frac{\varrho^2||\bar{b}_{\mathcal{A}}||}{L_{\max}^2} - C\Big(\frac{\varrho}{N} + \frac{1}{N^3}\Big) \leq B(\widehat{\theta}) \leq \frac{16r^2 L_{\max}^2}{\lambda_{\min}^2(1-\varrho)^2}\frac{\varrho^2||\bar{b}_{\mathcal{A}}||}{\lambda_{\min}^2} + O\Big(\frac{\varrho}{N} + \frac{1}{N^3}\Big).$$

This finishes the proof.

### B.7 PROOF OF THEOREM 4.1

Similarly to the proof of Theorem A.2, to prove Theorem 4.1, we aim to verify that with high probability: (1) the eigenvalues of $\ddot{\mathcal{L}}_\omega(\theta)$ are still bounded (by positive constants) when $\theta$ lies in a neighborhood of $\theta_0$; and (2) the sequence $\{\widehat{\theta}_{\mathcal{A}}^{*(t)}\}$ lies in the neighborhood of $\theta_0^*$ for any $t$. Once these conditions are established, we can mimic Proof B.3 to complete the proof of the theorem.

**Proof of (1):** Define $\delta = (\bar{\omega}_N^\varrho/\bar{\omega})\lambda_{\min}/(4L_{\max})$, $\bar{\omega}_N^\varrho = M^{-1}\sum_{m\in\mathcal{A}}\omega_m$, and event

$$\mathcal{E}_2^{\omega\prime} = \left\{\|\ddot{\mathcal{L}}_w(\theta) - \Omega_{\mathcal{A}}^w(\theta)\| \leq \frac{\bar{\omega}_N^\varrho\lambda_{\min}}{4}\right\}.$$

Using similar technique skills in the proof of Lemma C.1, we have $\mathbb{P}\{(\mathcal{E}_2^{\omega\prime})^c\} \lesssim C_{\max}^8/((\bar{\omega}_N^\varrho)^8\lambda_{\min}^8 N^4)$. We first prove that

$$\ddot{\mathcal{L}}_\omega(\theta) \geq \frac{1}{2}\bar{\omega}_N^\varrho\lambda_{\min}$$

under $\mathcal{E}_2^{\omega\prime}$ for $\theta \in B(\theta_0, \delta)$ with $\delta = (\bar{\omega}_N^\varrho/\bar{\omega})\lambda_{\min}/(4L_{\max})$.

Recall that

$$\Omega_{\mathcal{A}}^\omega(\theta) = M^{-1}\sum_{m=1}^M \omega_m\Omega_m(\theta)$$

Under Assumption 3, we have

$$\Omega_{\mathcal{A}}^\omega(\theta_0) \geq M^{-1}\sum_{m=1}^M \omega_m\big(0 + (1 - a_m)\lambda_{\min}\big) = M^{-1}\sum_{m=1}^M \omega_m(1 - a_m)\lambda_{\min} = \bar{\omega}_N^\varrho\lambda_{\min}.$$

Note that $\Omega_{\mathcal{A}}^\omega(\theta) - \Omega_{\mathcal{A}}^\omega(\theta_0) = M^{-1}\sum_{m=1}^M \omega_m\big\{\Omega_m(\theta) - \Omega_m(\theta_0)\big\}$. Then it could be verified that under Assumption 4,

$$\|\Omega_{\mathcal{A}}^\omega(\theta) - \Omega_{\mathcal{A}}^\omega(\theta_0)\| \leq \bar{\omega}L_{\max}\|\theta - \theta_0\|.$$

For any $\|\theta - \theta_0\| \leq (\bar{\omega}_N^\varrho/\bar{\omega})\lambda_{\min}/(4L_{\max})$, under Assumptions 3, 4 and Wely's inequality, we have

$$\lambda_{\min}\big(\Omega_{\mathcal{A}}^\omega(\theta)\big) \geq \lambda_{\min}\big(\Omega_{\mathcal{A}}^\omega(\theta_0)\big) - \bar{\omega}L_{\max}\|\theta - \theta_0\| \geq \frac{3}{4}\bar{\omega}_N^\varrho\lambda_{\min}.$$

In addition, recall $\ddot{\mathcal{L}}_\omega(\theta) = M^{-1}\sum_{m=1}^M \omega_m\ddot{\mathcal{L}}_m(\theta)$. This yields

$$\ddot{\mathcal{L}}_\omega(\theta) \geq \frac{1}{2}\bar{\omega}_N^\varrho\lambda_{\min}.$$

under $\mathcal{E}_2^{\omega\prime}$ for $\theta \in B(\theta_0, \delta)$.

**Proof of (2):** For any $\|\theta - \widehat{\theta}_{\mathcal{A}}\| \leq \delta/2$, we have

$$\|\theta - \theta_0\| \leq \|\theta - \widehat{\theta}_{\mathcal{A}}\| + \|\widehat{\theta}_{\mathcal{A}} - \theta_0\| \leq \delta$$

for sufficiently large $N$ with probability at least $1 - O(1/N^4)$ by Lemma 6 in Zhang et al. (2013). Denote $R = \delta/(4\sqrt{M})$, we are going to verify that if $\|\bar{\theta}_{\mathcal{A}}^{*(t)} - \widehat{\theta}_{\mathcal{A}}^*\| \leq \sqrt{M}R$ and $\|\widehat{\theta}_{\mathcal{A}}^{*(t)} - \bar{\theta}_{\mathcal{A}}^{*(t)}\| \leq \sqrt{M}R$, then we have $\|\bar{\theta}_{\mathcal{A}}^{*(t+1)} - \widehat{\theta}_{\mathcal{A}}^*\| \leq \sqrt{M}R$ and $\|\widehat{\theta}_{\mathcal{A}}^{*(t+1)} - \bar{\theta}_{\mathcal{A}}^{*(t+1)}\| \leq \sqrt{M}R$.

Note that now we have $\ddot{\mathcal{L}}_\omega(\theta) \geq \frac{1}{2}\bar{\omega}_N^\varrho\lambda_{\min}$, using similar technique skills in the proof of Theorem 4.1, we have

$$\|\bar{\theta}_{\mathcal{A}}^{*(t+1)} - \widehat{\theta}_{\mathcal{A}}^*\| \leq \left(1 - \frac{\alpha\bar{\omega}_N^\varrho\lambda_{\min}}{2}\right)\|\bar{\theta}_{\mathcal{A}}^{*(t)} - \widehat{\theta}_{\mathcal{A}}^*\| + \left\{2L_{\max}\alpha\rho + \mathrm{SE}(W)\right\}\|\widehat{\theta}_{\mathcal{A}}^{*(t)} - \bar{\theta}_{\mathcal{A}}^{*(t)}\|$$

$$+ \alpha\sqrt{M}\widehat{\mathrm{SE}}_\omega.$$

$$\|\widehat{\theta}_{\mathcal{A}}^{*(t+1)} - \bar{\theta}_{\mathcal{A}}^{*(t)}\| \leq \left\{\rho + 2\rho L_{\max}\alpha + \mathrm{SE}(W)\right\}\|\bar{\theta}_{\mathcal{A}}^{*(t)} - \bar{\theta}_{\mathcal{A}}^{*(t)}\| + 2\alpha L_{\max}\|\bar{\theta}_{\mathcal{A}}^{*(t)} - \widehat{\theta}_{\mathcal{A}}^*\|$$

$$+ \alpha\sqrt{M}\widehat{\mathrm{SE}}. \tag{B.16}$$

We start with $\widehat{\theta}^{*(1)}$. For simplicity, we consider all the clients start from the same initializer with $\|\widehat{\theta}^{(0,m)} - \widehat{\theta}_{\mathcal{A}}\| \equiv \widehat{\delta}^{(0)}$, and assume $\widehat{\delta}^{(0)}$ is sufficiently small with $\delta^{(0)} < R$. Then equation (B.16) could be simplified as

$$\|\bar{\theta}_{\mathcal{A}}^{*(1)} - \widehat{\theta}_{\mathcal{A}}^*\| \leq \left(1 - \frac{\alpha\bar{\omega}_N^{\varrho}\lambda_{\min}}{2}\right)\|\bar{\theta}_{\mathcal{A}}^{*(0)} - \widehat{\theta}_{\mathcal{A}}^*\| + \alpha\sqrt{M}\overline{\mathrm{SE}}_{\omega}.$$

$$\|\widehat{\theta}_{\mathcal{A}}^{*(1)} - \bar{\theta}_{\mathcal{A}}^{*(1)}\| \leq 2\alpha L_{\max}\|\widehat{\theta}_{\mathcal{A}}^{*(0)} - \widehat{\theta}_{\mathcal{A}}^*\| + \alpha\sqrt{M}\widehat{\mathrm{SE}}.$$

if $\alpha$ and $\delta^0$ are suffciently small such that $(1-\alpha\bar{\omega}_N^{\varrho}\lambda_{\min}/2)\delta^0 + \alpha\overline{\mathrm{SE}}_{\omega} \leq R$, and $\alpha(2L_{\max}R + \widehat{\mathrm{SE}}) \leq R$, then we have $\|\bar{\theta}_{\mathcal{A}}^{*(1)} - \widehat{\theta}_{\mathcal{A}}^*\| \leq \sqrt{M}R$ and $\|\widehat{\theta}_{\mathcal{A}}^{*(1)} - \bar{\theta}_{\mathcal{A}}^{*(1)}\| \leq \sqrt{M}R$.

Subsequently, assume that for any $t > 0$, we have $\|\bar{\theta}_{\mathcal{A}}^{*(t)} - \widehat{\theta}_{\mathcal{A}}^*\| \leq \sqrt{M}R$ and $\|\widehat{\theta}_{\mathcal{A}}^{*(t)} - \bar{\theta}_{\mathcal{A}}^{*(t)}\| \leq \sqrt{M}R$. Similar to equation (B.8), we have

$$\begin{pmatrix} \widehat{\delta}^{*(t+1)} \\ \bar{\delta}^{*(t+1)} \end{pmatrix} \leq \mathbf{H}^{t+1}\begin{pmatrix} \widehat{\delta}^{*(0)} \\ 0 \end{pmatrix} + \left(\frac{8\sqrt{M}}{(1-\rho)\bar{\omega}_N^{\varrho}\lambda_{\min}}\right)\begin{pmatrix} \alpha\big(\bar{\omega}_N^{\varrho}\lambda_{\min}\widehat{\mathrm{SE}}/2 + 2L_{\max}\overline{\mathrm{SE}}_{\omega}\big) \\ \big\{2\alpha L_{\max} + \mathrm{SE}(W)\big\}\widehat{\mathrm{SE}} + (1-\rho)\overline{\mathrm{SE}}_{\omega} \end{pmatrix}.$$

We first analyze $\widehat{\delta}^{*(t+1)}$, it could be verified that

$$\widehat{\delta}^{*(0)}\widehat{\delta}^{*(t+1)} \leq (\widehat{\delta}^{*(0)})^2\,(1,0)\,\mathbf{H}^{t+1}\begin{pmatrix} 1 \\ 0 \end{pmatrix} + \widehat{\delta}^{*(0)}\left(\frac{8\sqrt{M}}{(1-\rho)\bar{\omega}_N^{\varrho}\lambda_{\min}}\right)\alpha\big(\bar{\omega}_N^{\varrho}\lambda_{\min}\widehat{\mathrm{SE}}/2 + 2L_{\max}\overline{\mathrm{SE}}_{\omega}\big).$$

This yields

$$\widehat{\delta}^{*(t+1)} \leq \left(1 - \frac{\alpha\bar{\omega}_N^{\varrho}\lambda_{\min}}{8}\right)^{t+1}\sqrt{M}\widehat{\delta}^0 + \sqrt{M}\alpha\left(\frac{8}{(1-\rho)\bar{\omega}_N^{\varrho}\lambda_{\min}}\right)\big(\bar{\omega}_N^{\varrho}\lambda_{\min}\widehat{\mathrm{SE}}/2 + 2L_{\max}\overline{\mathrm{SE}}_{\omega}\big).$$

As long as we have

$$\left(1 - \frac{\alpha\bar{\omega}_N^{\varrho}\lambda_{\min}}{8}\right)^{t+1}\widehat{\delta}^0 + \alpha\left(\frac{8}{(1-\rho)\bar{\omega}_N^{\varrho}\lambda_{\min}}\right)\big(\bar{\omega}_N^{\varrho}\lambda_{\min}\widehat{\mathrm{SE}}/2 + 2L_{\max}\overline{\mathrm{SE}}_{\omega}\big) \leq R.$$

We have $\widehat{\delta}^{*(t+1)} \leq \sqrt{M}R$. We subsequently analyze $\bar{\delta}^{*(t+1)}$, by equation (B.16), it could be shown that

$$\|\widehat{\theta}_{\mathcal{A}}^{*(t+1)} - \bar{\theta}_{\mathcal{A}}^{*(t)}\| \leq \rho\sqrt{M}R + \big\{2\rho L_{\max}\alpha + \mathrm{SE}(W)\big\}\sqrt{M}R +$$
$$2\alpha\sqrt{M}L_{\max}\Big\{\Big(1 - \frac{\alpha\bar{\omega}_N^{\varrho}\lambda_{\min}}{8}\Big)^t\widehat{\delta}^0 + O(\alpha)\Big\} + \sqrt{M}\alpha\widehat{\mathrm{SE}}.$$

As long as we have

$$\big\{2L_{\max}\alpha + \mathrm{SE}(W)\big\}R + \alpha\Big\{2L_{\max}\Big(1 - \frac{\alpha\bar{\omega}_N^{\varrho}\lambda_{\min}}{8}\Big)^t\widehat{\delta}^0 + \widehat{\mathrm{SE}}\Big\} \leq (1-\rho)R/2.$$

This leads to $\bar{\delta}^{*(t+1)} \leq \sqrt{M}R$.

Combining the above results, we have proved that $\|\widehat{\theta}_{\mathcal{A}}^{*(t)} - \widehat{\theta}_{\mathcal{A}}^*\| \leq \delta/2$ for any $t$. Applying the above results, using similar techniques as those utilized in Appendix B.3, we have

$$M^{-1/2}\|\widehat{\theta}_{\mathcal{A}}^{*(t)} - \widehat{\theta}_{\mathcal{A}}^*\| \lesssim \Big(1 - \frac{\alpha\bar{\omega}_N^{\varrho}\lambda_{\min}}{8}\Big)^t\widehat{\delta}_0 + \frac{\alpha L_{\max} + \mathrm{SE}(W)}{(1-\rho)\lambda_{\min}\bar{\omega}_N^{\varrho}}\Big\{\Big(\frac{\log n}{n}\Big)^{1/2}L_{\max} + \Big(\frac{|\mathcal{A}|}{M}\Big)^{1/2}\bar{\mathfrak{b}}_2^{\mathcal{A}}\Big\}$$

$$+ (\bar{\omega}_N^{\varrho}\lambda_{\min})^{-1}\Big\{\frac{|\mathcal{A}|}{M}\bar{\mathfrak{b}}_2^{\mathcal{A}}\overline{\omega}_2^{\mathcal{A}} + \bar{\Delta}_2\Big(\frac{\log N}{N}\Big)^{1/2} + L_{\max}\bar{\Delta}_2\|\widehat{\theta}_{\mathcal{A}} - \theta_0\|\Big\}.$$

$$M^{-1/2}\lim_{t\to\infty}\|\widehat{\theta}_{\mathcal{A}}^{*(t)} - I^*\widehat{\theta}_{\mathcal{A}}\| \lesssim \frac{1}{\bar{\omega}_N^{\varrho}}\Big[\big\{\alpha + \mathrm{SE}(W)\big\}\Big\{\frac{1}{\sqrt{n}} + \Big(\frac{|\mathcal{A}|}{M}\Big)^{1/2}\Big\} + \Big(\frac{|\mathcal{A}|}{M}\overline{\omega}_2^{\mathcal{A}} + \frac{\bar{\Delta}_2}{\sqrt{N}}\Big)\Big]$$

with probability tending to 1. This finishes the proof.

### B.8 PROOF OF THEOREM 4.2

To prove Theorem 4.2, it suffices to show that under the local strong convexity assumption 3, the following term, defined in PART 1 of the proof of Theorem A.3 in Appendix B.4, satisfies:

$$\Delta_1^{(2)} = \left\{ \frac{1}{|\mathcal{A}|} \sum_{m \in \mathcal{A}} \exp\left( - 6\lambda_n \|\dot{\mathcal{L}}_m(\theta_{\mathcal{A}}) - \dot{\mathcal{L}}_m(\theta_m)\| \right) \right\}^{1/3} \leq \exp(-c),$$

for some positive constant $c > 0$ with high probability.

To this end, we begin by verifying the strong convexity of $\mathcal{L}_m(\theta)$ in a neighborhood of $\theta_m$. For all $\theta \in B(\theta_m, \widetilde{\delta})$, with $\widetilde{\delta} = \lambda_{\min}/(4L_{\max})$, we have:

$$\lambda_{\min}\big(\Omega_m(\theta)\big) \geq \lambda_{\min}\big(\Omega_m(\theta_m)\big) - L_{\max}\|\theta - \theta_m\| \geq \frac{3}{4}\lambda_{\min}.$$

Recall the events:

$$\mathcal{E}_{2,m} = \big\{ \|\ddot{\mathcal{L}}_m(\theta) - \Omega_{\mathcal{A}}(\theta)\| \leq \lambda_{\min}/4 \big\}, \quad \mathcal{E}_6 = \big\{ \|\theta_{\mathcal{A}} - \theta_0\| \leq \min_{m \in \mathcal{A}} \|\theta_m - \theta_0\|/2 \big\},$$

which jointly occur with probability at least $1 - O(M/n^4)$. Hence, it suffices to analyze $\Delta_1^{(2)}$ under the joint occurrence of these events. Under these good events, we have:

$$\ddot{\mathcal{L}}_m(\theta) \geq \frac{1}{2}\lambda_{\min}, \quad \text{for all } \theta \in B(\theta_m, \widetilde{\delta}).$$

Next, define:

$$\widetilde{\theta} = \theta_m + \widetilde{\eta} u, \quad \text{where } u = \theta_{\mathcal{A}} - \theta_m, \quad \widetilde{\eta} = \min\{1, \delta/\|u\|\}.$$

If $\|u\| \leq \delta$, then clearly $\widetilde{\theta} = \theta_{\mathcal{A}}$. Otherwise, if $\|u\| > \delta$, then $\widetilde{\theta}$ lies on the line joining $\theta_m$ and $\theta_{\mathcal{A}}$ with $\|\widetilde{\theta} - \theta_m\| = \delta$. Under Assumption 3, it can be verified that:

$$\big[\dot{\mathcal{L}}_m(\widetilde{\theta}) - \dot{\mathcal{L}}_m(\theta_m)\big]^\top u \geq \frac{\lambda_{\min}}{2}\widetilde{\eta}\|u\|^2 = \frac{\lambda_{\min}}{2}\|u\| \min\{\delta, \|u\|\}.$$

Define a scalar function: $g_m(\eta) = L_m(\theta_m + \eta u)$. It is straightforward to verify that $g_m(\eta)$ is a convex function on $\mathbb{R}$. Hence, we have: $\dot{g}_m(1) \geq \dot{g}_m(\widetilde{\eta})$. Since $\dot{g}_m(\eta) = \big\{\dot{\mathcal{L}}_m(\theta_m + \eta u)\big\}^\top u$, it follows directly that:

$$\big\{\dot{\mathcal{L}}_m(\theta_{\mathcal{A}})\big\}^\top u \geq \big\{\dot{\mathcal{L}}_m(\widetilde{\theta})\big\}^\top u.$$

This leads to

$$\big\{\dot{\mathcal{L}}_m(\theta_{\mathcal{A}}) - \dot{\mathcal{L}}_m(\theta_m)\big\}^\top u \geq \frac{\lambda_{\min}}{2}\|u\| \min\{\delta, \|u\|\}.$$

From the above inequality, we obtain:

$$\|\dot{\mathcal{L}}_m(\theta_{\mathcal{A}}) - \dot{\mathcal{L}}_m(\theta_m)\| \geq \frac{\lambda_{\min}}{2} \min\{\delta, \|u\|\} \geq \frac{\lambda_{\min}}{2} \min\left\{\delta, \frac{\min_{m \in \mathcal{A}} \|\theta_m - \theta_0\|}{2}\right\}.$$

Combining the above results, we finally arrive at:

$$\Delta_1^{(2)} \leq \exp\left(-\frac{\lambda_n \lambda_{\min}}{2} \min\left\{\delta, \frac{\min_{m \in \mathcal{A}} \|\theta_m - \theta_0\|}{2}\right\}\right),$$

which completes the proof.

## C TECHNICAL LEMMAS

In this Appendix, we define several "good events" and provide some useful technical lemmas for the theoretical analysis.

We first define the following "good events", where the definitions of $\ddot{\mathcal{L}}_w(\theta)$ and $\Omega_{\mathcal{A}}^w(\theta)$ are provided in equation (B.3).

$$\mathcal{E}_{1,m} = \left\{ \frac{1}{n} \sum_{i \in \mathcal{S}_m} L(X_i, Y_i) \leq 2L_{\max} \right\}; \quad \mathcal{E}_1 = \left\{ \frac{1}{N} \sum_{i=1}^{N} L(X_i, Y_i) \leq 2L_{\max} \right\};$$

$$\mathcal{E}_2 = \left\{ \left\| \ddot{\mathcal{L}}(\theta) - \Omega_{\mathcal{A}}(\theta) \right\| \leq \frac{\lambda_{\min}}{2} \right\}; \quad \mathcal{E}_2^{\omega} = \left\{ \left\| \ddot{\mathcal{L}}_w(\theta) - \Omega_{\mathcal{A}}^w(\theta) \right\| \leq \frac{\bar{\omega}\lambda_{\min}}{2} \right\};$$

$$\mathcal{E}_{3,m} = \left\{ \left\| \dot{\mathcal{L}}_{(m)}(\theta_0) - \mathbb{E}_m \dot{\ell}(x, y; \theta_0) \right\| \leq \|\flat_m\| \Big| m \in \mathcal{A} \right\};$$

$$\mathcal{E}_4 = \left\{ \left\| \frac{1}{M - |\mathcal{A}|} \sum_{m \notin \mathcal{A}} \Delta_m \dot{\mathcal{L}}_{(m)}(\theta_0) \right\| \leq \frac{\left( \frac{1}{M-|\mathcal{A}|} \sum_{m \notin \mathcal{A}} \Delta_m^2 \right)^{1/2} \sqrt{\log(N - n|\mathcal{A}|)}}{\sqrt{N - n|\mathcal{A}|}} \right\};$$

$$\mathcal{E}_5 = \left\{ M^{-1} \sum_{m=1}^{M} \left\| \dot{\mathcal{L}}_{(m)}(\theta_0) - \mathbb{E}_m \dot{\ell}(x, y; \theta_0) \right\|^2 \leq \frac{\log n}{n} \right\}. \tag{C.1}$$

**Lemma C.1.** *For the good events we defined in* (C.1)*, the following inequalities hold:*

$$(i): \mathbb{P}(\mathcal{E}_1^c) \lesssim \frac{1}{N^4}; \quad \mathbb{P}\left( \bigcup_m \mathcal{E}_{1,m}^c \right) \lesssim \frac{M}{n^4}. (ii): \mathbb{P}(\mathcal{E}_2^c) \lesssim \frac{C_{\max}^8}{\lambda_{\min}^8 N^4}; \mathbb{P}\left\{ (\mathcal{E}_2^{\omega})^c \right\} \lesssim \frac{C_{\max}^8}{\bar{\omega}^8 \lambda_{\min}^8 N^4}.$$

$$(iii): \mathbb{P}\left( \bigcup \mathcal{E}_{3,m}^c \right) \lesssim \frac{|\mathcal{A}|}{n^4}. (iv): \mathbb{P}(\mathcal{E}_4^c) \lesssim \frac{C_{\max}^8}{\left\{ \log(N - n|\mathcal{A}|) \right\}^4}. (v): \mathbb{P}(\mathcal{E}_5^c) \lesssim \frac{C_{\max}^8}{(\log n - C_{\max}^2)^4}.$$

*Proof.* We omit similar proofs for brevity.

**Proof of (i):** It could be proved that

$$\mathbb{P}\left( \bigcup \mathcal{E}_{1,m}^c \right) \leq \sum_{m=1}^{M} \mathbb{P}(\mathcal{E}_{1,m}^c) \lesssim \frac{M}{n^4}.$$

The second inequality follows from Markov's inequality and Assumptions $1 - 4$.

**Proof of (ii):** Assumptions $1 - 4$ hold and note that $E\left\{ \|\ddot{\mathcal{L}}(\theta) - \Omega_{\mathcal{A}}(\theta)\|^8 \right\} \lesssim N^{-4} C_{\max}^8$ (see detailed proof in Lemma 7 in Zhang et al. (2013)). It could be verified that $\mathbb{P}(\mathcal{E}_2^c) \lesssim C_{\max}^8/(\lambda_{\min}^8 N^4)$ by Markov's inequality. In addition, note that $\lambda_{\min}\left\{ \Omega_{\mathcal{A}}^{\omega}(\theta) \right\} \geq \bar{\omega}\lambda_{\min}$ since for any vector $x \in \mathbb{R}^p$ satisfies $x^\top x = 1$, we have

$$x^\top \Omega_{\mathcal{A}}^{\omega}(\theta)x = M^{-1} \sum_{m=1}^{M} \omega_m \left\{ x^\top \Omega_m(\theta)x \right\} \geq M^{-1} \sum_{m=1}^{M} \omega_m \lambda_{\min} = \bar{\omega}\lambda_{\min}.$$

This leads to $\mathbb{P}\left\{ (\mathcal{E}_2^{\omega})^c \right\} \lesssim C_{\max}^8/(\bar{\omega}^8 \lambda_{\min}^8 N^4)$.

**Proof of (iii) – (v):** Similarly, it could be shown that

$$\mathbb{P}\left( \bigcup \mathcal{E}_{3,m}^c \right) \leq \sum_{m \in \mathcal{A}} \mathbb{P}\left( \|\dot{\mathcal{L}}_{(m)}(\theta_0) - \mathbb{E}_m \dot{\ell}(x, y; \theta_0)\| > \|\flat_m\| \right) \lesssim \frac{|\mathcal{A}|}{n^4}.$$

$$\mathbb{P}(\mathcal{E}_4^c) \lesssim \frac{\left( \frac{1}{M-|\mathcal{A}|} \sum_{m \notin \mathcal{A}} \Delta_m^2 \right)^4 C_{\max}^8}{\left\{ \log(N - n|\mathcal{A}|) \right\}^4 \left( \frac{1}{M-|\mathcal{A}|} \sum_{m \notin \mathcal{A}} \Delta_m^2 \right)^4}.$$

$$\mathbb{P}(\mathcal{E}_5^c) \leq \mathbb{P}\left( M^{-1} \sum_{m=1}^{M} \left( \|Z_m\|^2 - \mathbb{E}\|Z_m\|^2 \right) > \frac{\log n}{n} - \frac{C_{\max}^2}{n} \right) \lesssim \frac{C_{\max}^8}{(\log n - C_{\max}^2)^4}.$$

Here $Z_m = \dot{\mathcal{L}}_{(m)}(\theta_0) - \mathbb{E}_m \dot{\ell}(x, y; \theta_0)$. This finishes the lemma proof. $\square$

**Lemma C.2.** *Let $\widehat{\theta}^{(t,m)}$ denote the DFL estimator at the $t$th iteration on the $m$th client. Define $\widetilde{\theta}^{(t,m)} = \sum w_{mk} \widehat{\theta}^{(t,k)}$ as the neighborhood-averaged estimator, and let $\bar{\theta}^{(t)} = M^{-1} \sum_{m=1}^{M} \widehat{\theta}^{(t,m)}$*

*represent the averaged estimator at the tth iteration. The stacked DFL estimator and averated estimator at the tth iteration are denoted by $\widehat{\theta}^{*(t)} = ((\widehat{\theta}^{(t,1)})^\top, \ldots, (\widehat{\theta}^{(t,M)})^\top)^\top$, and $\bar{\theta}^{*(t)} = I^*\bar{\theta}^{(t)}$, respectively. For any $t$, we have*

$$(i): \|(W \otimes I_q)(\widehat{\theta}^{*(t)} - \bar{\theta}^{*(t)})\| \leq \rho \|\widehat{\theta}^{*(t)} - \bar{\theta}^{*(t)}\|.$$

$$(ii): \|M^{-1} \sum_{m=1}^M \widetilde{\theta}^{(t,m)} - \bar{\theta}^{(t)}\| \leq M^{-1/2} \operatorname{SE}(W) \|\widehat{\theta}^{*(t)} - \bar{\theta}^{*(t)}\|.$$

*Proof.* The proof is given in Lemmas 6 and 7 in Wu et al. (2024). $\square$

**Lemma C.3.** *Define a $2 \times 2$ matrix*

$$\mathbf{H} = \begin{bmatrix} \rho + 2\alpha\rho L_{\max} + \operatorname{SE}(W) & 2\alpha L_{\max} \\ 2\alpha\rho L_{\max} + \operatorname{SE}(W) & 1 - \alpha\bar{\omega}\lambda_{\min}/2 \end{bmatrix}$$

*with $\alpha, \lambda_{\min}, L_{\max}, \operatorname{SE}(W) > 0$, $\bar{\omega} \in [0,1]$, and $\rho \in (0,1)$. Denote $\rho_H = \max|\lambda(\mathbf{H})|$. For the sake of simplicity in the proof, we choose a sufficiently large $L_{\max}$ in Assumption 4 such that $\lambda_{\min} \leq L_{\max}$. Subsequently, we assume that*

$$2\alpha L_{\max} + \operatorname{SE}(W) < \frac{(1-\rho)\bar{w}\lambda_{\min}}{16 L_{\max}}. \tag{C.2}$$

*Then*

$$0 < \rho_H < 1 - \frac{\alpha\bar{\omega}\lambda_{\min}}{8}. \tag{C.3}$$

*Proof.* Note that for any $2 \times 2$ matrix $(a, b; c, d)$ with $a, b, c, d > 0$, its maximum eigenvalue is given by $\{(a + d) + \sqrt{(a-d)^2 + 4bc}\}/2$. Then we have $\rho_H = \mathcal{O}_1/2 + \mathcal{O}_2/2$, where $\mathcal{O}_1 = \rho + 2\alpha\rho L_{\max} + \operatorname{SE}(W) + (1 - \alpha\bar{\omega}\lambda_{\min}/2)$, and $\mathcal{O}_2 = \{\rho + 2\alpha\rho L_{\max} + \operatorname{SE}(W) - (1 - \alpha\bar{\omega}\lambda_{\min}/2)\}^2 + 8\alpha L_{\max}\{2\alpha\rho L_{\max} + \operatorname{SE}(W)\}$. Then it fould be verified that

$$\frac{\mathcal{O}_1}{2} = \frac{\rho+1}{2} + \alpha\rho L_{\max} + \frac{\operatorname{SE}(W)}{2} - \frac{\alpha\bar{\omega}\lambda_{\min}}{4}. \tag{C.4}$$

$$\mathcal{O}_2^2 = \left\{1 - \rho - 2\alpha\rho L_{\max} - \operatorname{SE}(W)\right\}^2 + \left(\frac{\alpha\bar{\omega}\lambda_{\min}}{2}\right)^2 - \left\{1 - \rho - 2\alpha\rho L_{\max} - \operatorname{SE}(W)\right\} \times \alpha\bar{\omega}\lambda_{\min} + 8\alpha L_{\max}\{2\alpha\rho L_{\max} + \operatorname{SE}(W)\}.$$

To show equation (C.3), we first verify the following inequality:

$$\mathcal{O}_2^2 \leq \left\{1 - \rho - 2\alpha\rho L_{\max} - \operatorname{SE}(W)\right\}^2 + \left(\frac{\alpha\bar{\omega}\lambda_{\min}}{2}\right)^2. \tag{C.5}$$

Under assumption (C.2), it could be proved that

$$\left\{2\alpha\rho L_{\max} + \operatorname{SE}(W)\right\}\left(1 + \frac{\bar{\omega}\lambda_{\min}}{8L_{\max}}\right) < \frac{\bar{\omega}\lambda_{\min}}{8L_{\max}}(1 - \rho)$$

$$\implies 2\alpha\rho L_{\max} + \operatorname{SE}(W) < \frac{\bar{\omega}\lambda_{\min}}{8L_{\max}}\left\{1 - \rho - 2\alpha\rho L_{\max} - \operatorname{SE}(W)\right\}$$

$$\implies 8\alpha L_{\max}\left\{2\alpha\rho L_{\max} + \operatorname{SE}(W)\right\} < \left\{1 - \rho - 2\alpha\rho L_{\max} - \operatorname{SE}(W)\right\}\alpha\bar{\omega}\lambda_{\min}.$$

The first inequality holds because $\bar{\omega}\lambda_{\min}/(8L_{\max}) < 1$, which follows from the facts that $\bar{\omega} \leq 1$ and $\lambda_{\min} \leq L_{\max}$. This yields equation (C.5).

Combining the results of (C.4) and (C.5), we have

$$\rho_H < \frac{1+\rho}{2} + \alpha\rho L_{\max} + \frac{\operatorname{SE}(W)}{2} - \frac{\alpha\bar{\omega}\lambda_{\min}}{4} + \left[\left\{\frac{1-\rho}{2} - \alpha\rho L_{\max} - \frac{\operatorname{SE}(W)}{2}\right\}^2 + \left(\frac{\alpha\bar{\omega}\lambda_{\min}}{4}\right)^2\right]^{1/2}.$$

Under assumption (C.2), we have $1 - \alpha\bar{\omega}\lambda_{\min}/2 > 0$, which yields $\rho_H > 0$. Next, we are going to prove equation (C.3). To this end, it could be proved that under assumption (C.2), we have

$$\frac{3\alpha\bar{\omega}\lambda_{\min}}{16} < \frac{1-\rho}{4} \text{ and } \alpha\rho L_{\max} + \frac{\operatorname{SE}(W)}{2} < \frac{1-\rho}{4}.$$

This yields

$$\frac{3}{4}\frac{\alpha\bar{\omega}\lambda_{\min}}{4} < \frac{1-\rho}{2} - \alpha\rho L_{\max} - \frac{\mathrm{SE}(W)}{2}.$$

Define $\delta_1 = (1-\rho)/2 - \alpha\rho L_{\max} - \mathrm{SE}(W)/2$, and $\delta_2 = \alpha\bar{\omega}\lambda_{\min}/8$. Then it could be verified that

$$\frac{3}{4}(2\delta_2) < \delta_1 \implies 2\delta_1\delta_2 + 4\delta_2^2 - \delta_2^2 < 4\delta_1\delta_2 \implies 2\delta_1\delta_2 + 2\frac{\alpha\bar{\omega}\lambda_{\min}}{4}\delta_2 - \delta_2^2 < 2\delta_1\frac{\alpha\bar{\omega}\lambda_{\min}}{4} \implies$$

$$\delta_1^2 + \left(\frac{\alpha\bar{\omega}\lambda_{\min}}{4}\right)^2 < \left(\delta_1 + \frac{\alpha\bar{\omega}\lambda_{\min}}{4} - \delta_2\right)^2 = \left(1 - \delta_2 - \frac{1+\rho}{2} - \alpha\rho L_{\max} - \frac{\mathrm{SE}(W)}{2} + \frac{\alpha\bar{\omega}\lambda_{\min}}{4}\right)^2.$$

This yields

$$\frac{1+\rho}{2} + \alpha\rho L_{\max} + \frac{\mathrm{SE}(W)}{2} - \frac{\alpha\bar{\omega}\lambda_{\min}}{4} + \left[\left\{\frac{1-\rho}{2} - \alpha\rho L_{\max} - \frac{\mathrm{SE}(W)}{2}\right\}^2 + \left(\frac{\alpha\bar{\omega}\lambda_{\min}}{4}\right)^2\right]^{1/2} < 1 - \delta_2,$$

which leads to $\rho_H < 1 - \delta_2$. This finishes the proof. $\qquad\square$

# D  COMPLETE EXPERIMENTAL DETAILS AND RESULTS

## D.1  IMPLEMENTATION DETAILS

**Experiments compute resources.** For all experiments, we use an NVIDIA Tesla P100 GPU with 16GB GPU memory and 8 Intel(R) Xeon(R) Gold 6271 CPUs, equipped with a total of 64GB of RAM and 500GB of storage. The experiments are implemented using Python 3.8 and PyTorch 1.7.1, and the computational time required to generate each figure is approximately 6 to 8 hours.

**Implementation details of competitors.**

- **DFL** (Wu et al., 2023a): The standard DFL algorithm without considering abnormal clients. The resulting DFL estimator is used as the initial value for aDFL.
- **BRIDGE** (Fang et al., 2022): A robust method to aggregate neighbors' parameter estimates and run local gradient descent. Coordinate-wise median (BRIDGE-M) and coordinate-wise trimmed mean (BRIDGE-T) are used for robustness.
- **SLBRN** (Zhang & Wang, 2024): A robust gradient tracking algorithm, which aggregates both the parameter estimates and gradients from neighbors. Coordinate-wise median (SLBRN-M) and coordinate-wise trimmed mean (SLBRN-T) as used.
- **ClippedGossip** (Karimireddy et al., 2021; He et al., 2022): This algorithm utilizes ClippedGossip as the robust aggregation rule. Furthermore, local momentums are also used.

For the BRIDGE-T and SLBRN-T algorithms, we use $\varrho$ as the trimming proportion for the trimmed mean operation. For the ClippedGossip algorithm, we employ the adaptive clipping strategy proposed by He et al. (2022) to determine the clipping radius, with the hyperparameter $\delta_{\max} = 2\varrho$. Additionally, the momentum parameter is set to 0.9 to align with that in He et al. (2022).

**Implementation details in simulation.** For all algorithms, we adopt a fixed learning rate of $\alpha = 0.01$ and set the maximum number of iterations to $T = 500$. For the aDFL algorithm, we implement the two-stage aDFL algorithm as detailed in Appendix A.4. In addition, we apply the DCV algorithm in Appendix A.5 to select $\lambda_n$ from a candidate set of 5 grid points in the range $[\log(n)/25, \log(n)/5]$.

**Networks.** The details of the two network structures are as follows:

- **Directed Circle Network:** Assume that the clients are arranged in a fixed sequence with an in-degree $d_{m_1} = D > 0$ for each $1 \le m_1 \le M$. The network adjacency matrix $A = (a_{m_1 m_2})$ is then defined with $a_{m_1 m_2} = 1$ if $m_2 = \{(m_1 + d - 1) \mod M\} + 1$ for $1 \le d \le D$, and $a_{m_1 m_2} = 0$ otherwise. Here, $a \mod b$ denotes the remainder when the integer $a$ is divided by the integer $b$. The resulting network structure should be of a circle type Wu et al. (2023a).
- **Undirected Erdős–Rényi Graph:** Consider an undirected Erdős–Rényi graph represented by a symmetric adjacency matrix $A = (a_{m_1 m_2})$, where $a_{m_1 m_2} = a_{m_2 m_1}$ for all $1 \le m_1, m_2 \le M$. We generate each entry $a_{m_1 m_2}$ for $1 \le m_1 < m_2 \le M$ independently, with $P(a_{m_1 m_2} = 1) = q$ and $P(a_{m_1 m_2} = 0) = 1 - q$, where $q \in (0, 1]$ is the link probability. Subsequently, we ensure symmetry by setting $a_{m_2 m_1} = a_{m_1 m_2}$ for $m_1 > m_2$, and set $a_{m_1 m_2} = 0$ for $m_1 = m_2$.

## D.2 ADDITIONAL SIMULATION RESULTS

In the simulation experiments on synthetic data, the averaged values and confidence bands of MSEs under the Undirected Erdős–Rényi Graph are present in Figure D.1.

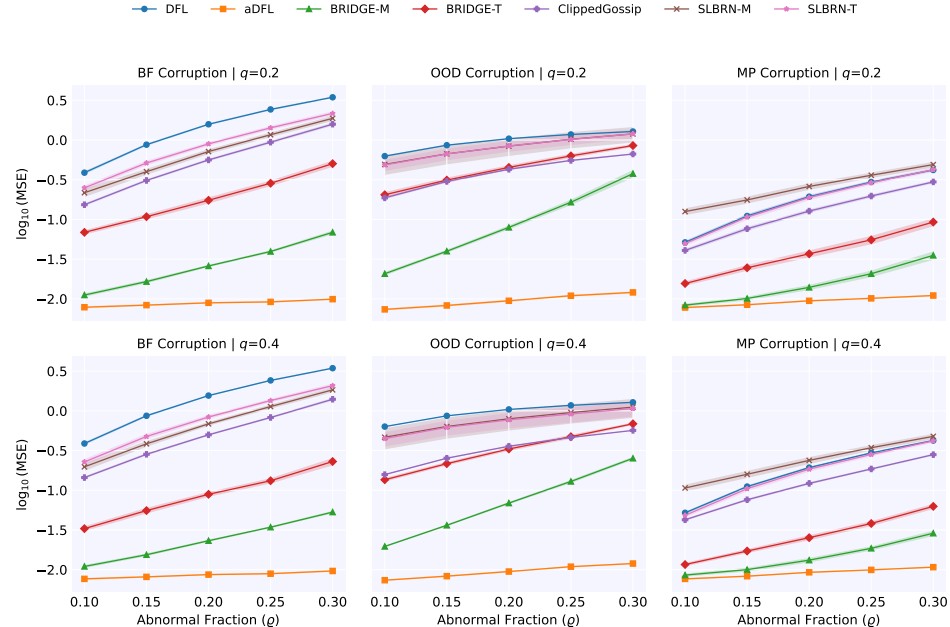

Figure D.1: The logarithm of MSE values versus the fraction of abnormal clients ($\varrho$) under the Undirected Erdős–Rényi Graph in the homogeneous scenario. Different algorithms are evaluated under different corruption types and two link probabilities ($q$).

To further evaluate performance under heterogeneous data distributions, we conduct simulation experiments using synthetic data distributed in a heterogeneous manner across clients. Specifically, for each client $m$ ($1 \leq m \leq M$), feature vectors $X_i$s are generated from the multivariate normal distribution $N_p(\mu_m, \Sigma_m)$. Here, the mean vector $\mu_m \in \mathbb{R}^p$ is constructed by sampling each element independently from the uniform distribution $\mathcal{U}(-0.5, 0.5)$. The covariance matrix $\Sigma_m \in \mathbb{R}^{p \times p}$ is defined as $\Sigma_m = \left(\rho_m^{|j_1 - j_2|}\right)$, where $\rho_m$ is sampled from $\mathcal{U}(0.2, 0.3)$. All other simulation settings remain identical to those described previously. The simulation results under two different network structures in the heterogeneous scenario are presented in Figures D.2 and D.3, respectively. Similar to the findings from the homogeneous scenario, the aDFL algorithm exhibits competitive performance compared to the competing methods.

We next consider two more realistic network structures. The first one is the scale-free network generated by the Barabasi-Albert (BA) model (Barabási & Albert, 1999), a standard topology for modeling real-world networks due to its power-law degree distribution. This structure is commonly used in federated learning research (Bhattacharya et al., 2024; Palmieri et al., 2023). Table D.1 presents the results under this network topology with $M = 200$ clients. We also consider a larger-scale case with $M = 500$ clients, whose results are shown in Table D.2. The second one is the stochastic block structure network. To demonstrate this part, we conduct a simulation experiment with $M = 100$ and $\varrho = 0.3$. We next construct a stochastic block structure network with two equal-sized blocks. To mimic the unevenly distributed case of abnormal clients, we put all abnormal clients in one block. The results are presented in Table D.3. These results show that our aDFL method performs excellently in more complex network structures and larger scales.

We also consider two more complex data corruption types. They are FGSM (Goodfellow et al., 2014) and PGD/iFGSM (Madry et al., 2017; Kurakin et al., 2018) attacks. We conduct experiments using the heterogeneous scenario under the scale-free network generated by the BA model with parameter $m = 5$. The results for two corruption types are shown in Tables D.4 and D.5, respectively. The results show that our aDFL method remains robust against these novel corruption types.

Last, we explore the effectiveness of our method for the dynamic corruption case. Specifically, for every round of gradient descent (GD) iteration, we randomly reassigned clients as either normal or abnormal. The fraction of abnormal clients is set to be $\varrho = 0.3$. Concurrently, the aDFL algorithm has to dynamically update its weights $\widehat{\omega}_m$ by revising Equation (5) from $\widehat{\omega}_m = \pi\{\lambda_n \|\dot{\mathcal{L}}_{(m)}(\widehat{\theta}^{(m)}_{\text{init}})\|\}$ to $\widehat{\omega}^{(t)}_m = \pi\{\lambda_n \|\dot{\mathcal{L}}_{(m)}(\widehat{\theta}^{(t,m)}_{\text{aDFL}})\|\}$. The corresponding results are presented in Table D.6, which demonstrate the robustness of our aDFL method against dynamic corruption.

Table D.1: The averaged MSE (standard deviation in parentheses) of different methods under the scale-free network generated by the Barabasi-Albert (BA) model with $M = 200$ (using the heterogeneous model setting with BA parameter $m = 5$).

| Method | $\varrho = 0.1$ | $\varrho = 0.15$ | $\varrho = 0.2$ | $\varrho = 0.25$ | $\varrho = 0.3$ |
|---|---|---|---|---|---|
| DFL | 0.250 (0.070) | 0.479 (0.124) | 0.719 (0.129) | 0.961 (0.155) | 1.235 (0.138) |
| aDFL | **0.004** (0.001) | **0.005** (0.001) | **0.005** (0.001) | **0.005** (0.001) | **0.006** (0.001) |
| BRIDGE-M | 0.008 (0.001) | 0.011 (0.003) | 0.017 (0.005) | 0.030 (0.014) | 0.082 (0.092) |
| BRIDGE-T | 0.008 (0.002) | 0.013 (0.005) | 0.030 (0.024) | 0.099 (0.096) | 0.285 (0.189) |
| ClippedGossip | 0.104 (0.037) | 0.192 (0.067) | 0.290 (0.076) | 0.403 (0.100) | 0.527 (0.095) |
| SLBRN-M | 0.268 (0.061) | 0.401 (0.067) | 0.590 (0.085) | 0.819 (0.137) | 1.112 (0.113) |
| SLBRN-T | 0.188 (0.036) | 0.337 (0.053) | 0.533 (0.080) | 0.768 (0.118) | 1.070 (0.101) |

Table D.2: The averaged MSE (standard deviation in parentheses) of different methods under the scale-free network generated by the Barabasi-Albert (BA) model with $M = 500$ and $\varrho = 0.3$ (using the heterogeneous model setting with BA parameter $m = 5$).

| DFL | aDFL | BRIDGE-M | BRIDGE-T | ClippedGossip | SLBRN-M | SLBRN-T |
|---|---|---|---|---|---|---|
| 1.301 | **0.003** | 0.062 | 0.483 | 0.564 | 1.020 | 1.056 |
| (0.086) | (0.001) | (0.020) | (0.105) | (0.065) | (0.131) | (0.066) |

Table D.3: The averaged MSE (standard deviation in parentheses) of different methods under the stochastic block network (using the heterogeneous model setting with the MP corruption and $\varrho = 0.3$).

| DFL | aDFL | BRIDGE-M | BRIDGE-T | ClippedGossip | SLBRN-M | SLBRN-T |
|---|---|---|---|---|---|---|
| 0.401 | **0.009** | 0.473 | 0.452 | 0.265 | 0.585 | 0.439 |
| (0.022) | (0.000) | (0.031) | (0.021) | (0.012) | (0.089) | (0.042) |

## D.3 REAL DATA APPLICATION

**Datasets.** We consider the following datasets to evaluate the effectiveness of our proposed aDFL method.

- MNIST (LeCun et al., 1998) consists of 70,000 handwritten digit images (0–9), with approximately 7,000 images per class. Among these, 60,000 are used for training and the remaining 10,000 are reserved for testing. For this dataset, we use the LeNet model with $p = 61,706$ parameters. The initial values of the LeNet model are set using the Xavier uniform initializer.

- CIFAR10 (Krizhevsky et al., 2009) comprises 60,000 color images, evenly distributed across 10 classes. Of these, 50,000 images are used for training and 10,000 are used for validation. For this dataset, we fine-tune the VGG16 model pre-trained on ImageNet dataset, with $p = 5,130$ trainable parameters.

- We further explore a more challenging dataset CINIC10 (Darlow et al., 2018), which consists of 270,000 images drawn from both CIFAR10 and downsampled ImageNet, evenly distributed across 10 classes. We then conduct the experiment for CINIC10 in a similar way as for CIFAR10 in Section 5.2. The corresponding results are shown in Figure D.12 (ii). It shows that the results remain encouraging and are qualitatively similar to those obtained on MNIST and CIFAR10.

**Distribution Pattern.** The following distribution scenarios are considered:

- **Homogeneous Scenario:** Images from the entire training dataset are randomly and uniformly distributed across 50 clients.

Table D.4: The averaged MSE (standard deviation in parentheses) of different methods under the FGSM attack with $\epsilon = 0.5$ (using the heterogeneous model setting).

| Method | $\varrho = 0.1$ | $\varrho = 0.15$ | $\varrho = 0.2$ | $\varrho = 0.25$ | $\varrho = 0.3$ |
|---|---|---|---|---|---|
| DFL | 0.250 (0.070) | 0.479 (0.124) | 0.719 (0.129) | 0.961 (0.155) | 1.235 (0.138) |
| aDFL | **0.004** (0.001) | **0.005** (0.001) | **0.005** (0.001) | **0.005** (0.001) | **0.006** (0.001) |
| BRIDGE-M | 0.008 (0.001) | 0.011 (0.003) | 0.017 (0.005) | 0.030 (0.014) | 0.082 (0.092) |
| BRIDGE-T | 0.008 (0.002) | 0.013 (0.005) | 0.030 (0.024) | 0.099 (0.096) | 0.285 (0.189) |
| ClippedGossip | 0.104 (0.037) | 0.192 (0.067) | 0.290 (0.076) | 0.403 (0.100) | 0.527 (0.095) |
| SLBRN-M | 0.268 (0.061) | 0.401 (0.067) | 0.590 (0.085) | 0.819 (0.137) | 1.112 (0.113) |
| SLBRN-T | 0.188 (0.036) | 0.337 (0.053) | 0.533 (0.080) | 0.768 (0.118) | 1.070 (0.101) |

Table D.5: The averaged MSE (standard deviation in parentheses) of different methods under the PGD/i-FGDM attack with $\epsilon = 0.5$ and $\alpha = 0.05$ (using the heterogeneous model setting with $\varrho = 0.3$).

| DFL | aDFL | BRIDGE-M | BRIDGE-T | ClippedGossip | SLBRN-M | SLBRN-T |
|---|---|---|---|---|---|---|
| 1.316 | **0.006** | 0.150 | 0.505 | 0.565 | 1.245 | 1.191 |
| (0.172) | (0.001) | (0.253) | (0.240) | (0.117) | (0.274) | (0.232) |

Table D.6: The averaged MSE (standard deviation in parentheses) of different methods under dynamic corruption (using the heterogeneous model setting under BF corruption with $\varrho = 0.3$ and a directed circle network with $D = 5$).

| DFL | aDFL | BRIDGE-M | BRIDGE-T | ClippedGossip |
|---|---|---|---|---|
| 3.556 | **0.010** | 1.267 | 2.100 | 3.305 |
| (0.128) | (0.002) | (0.079) | (0.094) | (0.131) |

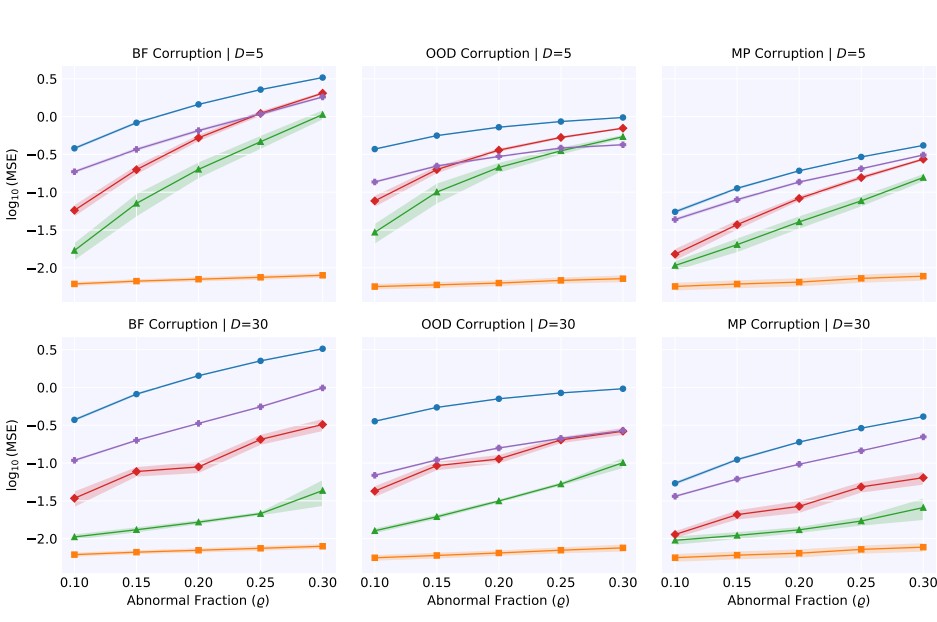

Figure D.2: The logarithm of MSE values versus the fraction of abnormal clients ($\varrho$) under the Directed Circle Network in the heterogeneous scenario. Different algorithms are evaluated under different corruption types and two in-degrees ($D$).

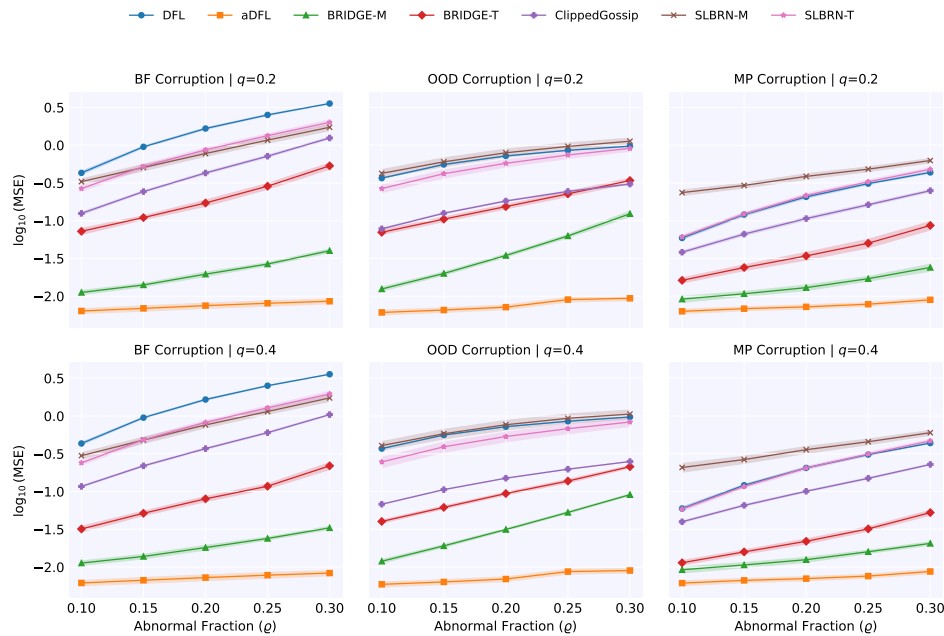

Figure D.3: The logarithm of MSE values versus the fraction of abnormal clients ($\varrho$) under the Undirected Erdős–Rényi Graph in the heterogeneous scenario. Different algorithms are evaluated under different corruption types and two link probabilities ($q$).

- **Heterogeneous Scenario:** Images are first grouped according to their labels, with each label category evenly divided into 25 subsets. These subsets are then assigned such that each client receives data from 5 subsets with different labels, ensuring that every client ultimately holds data associated with 5 distinct labels.

- To increase heterogeneity, we explore a more challenging label distribution. Specifically, we assume each client holds data from 10 different classes and the class distribution is highly imbalanced: one dominant label accounts for 64% of the client's data, while each of the remaining 9 labels contributes only 4%. This introduces significant intra-client label imbalance and inter-client diversity, as the dominant label varies across clients. The detailed results using the CIFAR-10 dataset are present in Figure D.12 (i). We find that, although our aDFL method performs slightly worse than the oracle method, it exceeds all competing methods. These results demonstrate the practical robustness of our method.

**Corruptions.** Two kinds of data corruption are considered.

- **OOD**: The feature vectors $X_i$'s on abnormal machines are replaced by $\widetilde{X_i} = 0.3X_i + 3V_p$, where the entries of $V_p \in \mathbb{R}^p$ are independently generated from a standard normal distribution $N(0, 1)$.

- **Label-Flipping (LF)**: We encode the image labels as numerical values ranging from 0 to 9. Subsequently, the response variables $Y_i$'s on the abnormal machines are replaced by $\widetilde{Y_i} = (Y_i + 1) \bmod 10$.

**Training strategy.** We randomly distribute all training samples equally to $M = 50$ clients. To speed up convergence, we adopt a constant-and-cut learning-rate scheduling strategy (Lang et al., 2019). Specifically, for the MNIST dataset, a total of 6,000 iterations are executed with an initial learning rate of $\alpha = 0.1$. The learning rate decreases to 0.05 after 200 iterations and to 0.01 after 4,000 iterations. For the CIFAR10 dataset, we run a total of 9,000 iterations with an initial learning rate of $\alpha = 1$. The learning rate is reduced to 0.5, 0.2, and 0.01 after 200, 5,000, and 8,000 iterations, respectively. For the proposed aDFL method, we recalculate the weights $\{\widehat{\omega}_m\}_m$ according to equation (4.4) whenever the learning rate is adjusted. In addition, we found that the ClippedGossip and SLBRN methods are highly sensitive to the learning rate in our setting. Therefore, we carefully adjusted the learning rate strategy for these methods separately.

**Results.** We present additional results that were not included in the main text. Specifically, results for real applications on MNIST and CIFAR10 under the homogeneous scenario are shown in Figures D.8 – D.11. Results under the heterogeneous scenario are shown in Figures D.5 – D.7. These results exhibit patterns consistent with those in Figure 2. In addition, we observe that the two SLBRN algorithms failed to converge in the Directed Circle Network. Therefore, the corresponding results are not reported.

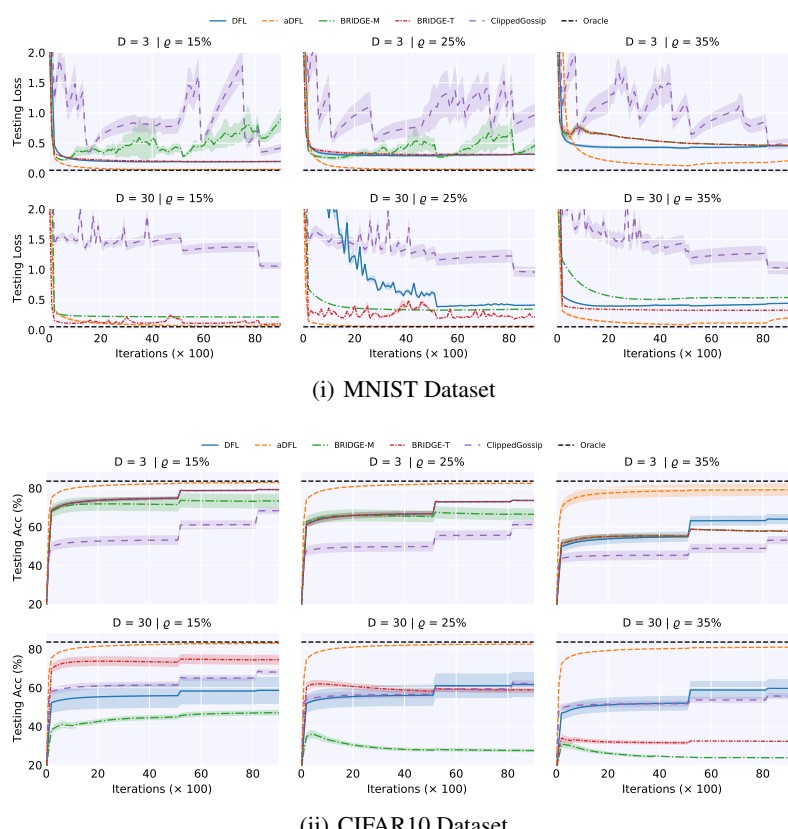

Figure D.4: The testing loss/accuracy over iterations for two datasets in the heterogeneous scenario. Different methods are evaluated with varying in-degrees ($D$) and fraction of abnormal clients ($\varrho$) under the LF corruption and Directed Circle Network.

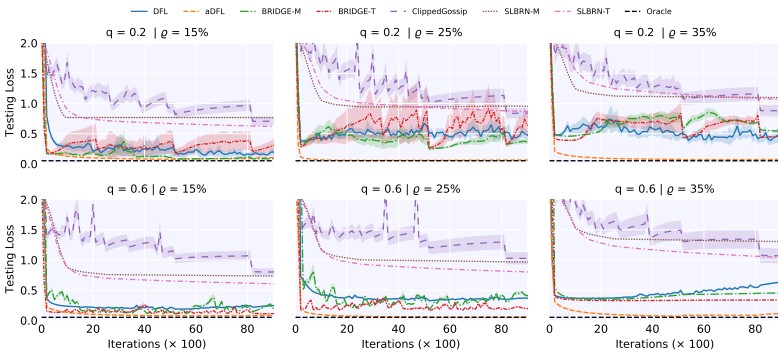

Figure D.5: The testing loss over iterations for MNIST in the heterogeneous scenario. Different methods are evaluated with varying link probabilities ($q$) and the fraction of abnormal clients ($\varrho$) under the LF corruption and Erdős–Rényi Graph.

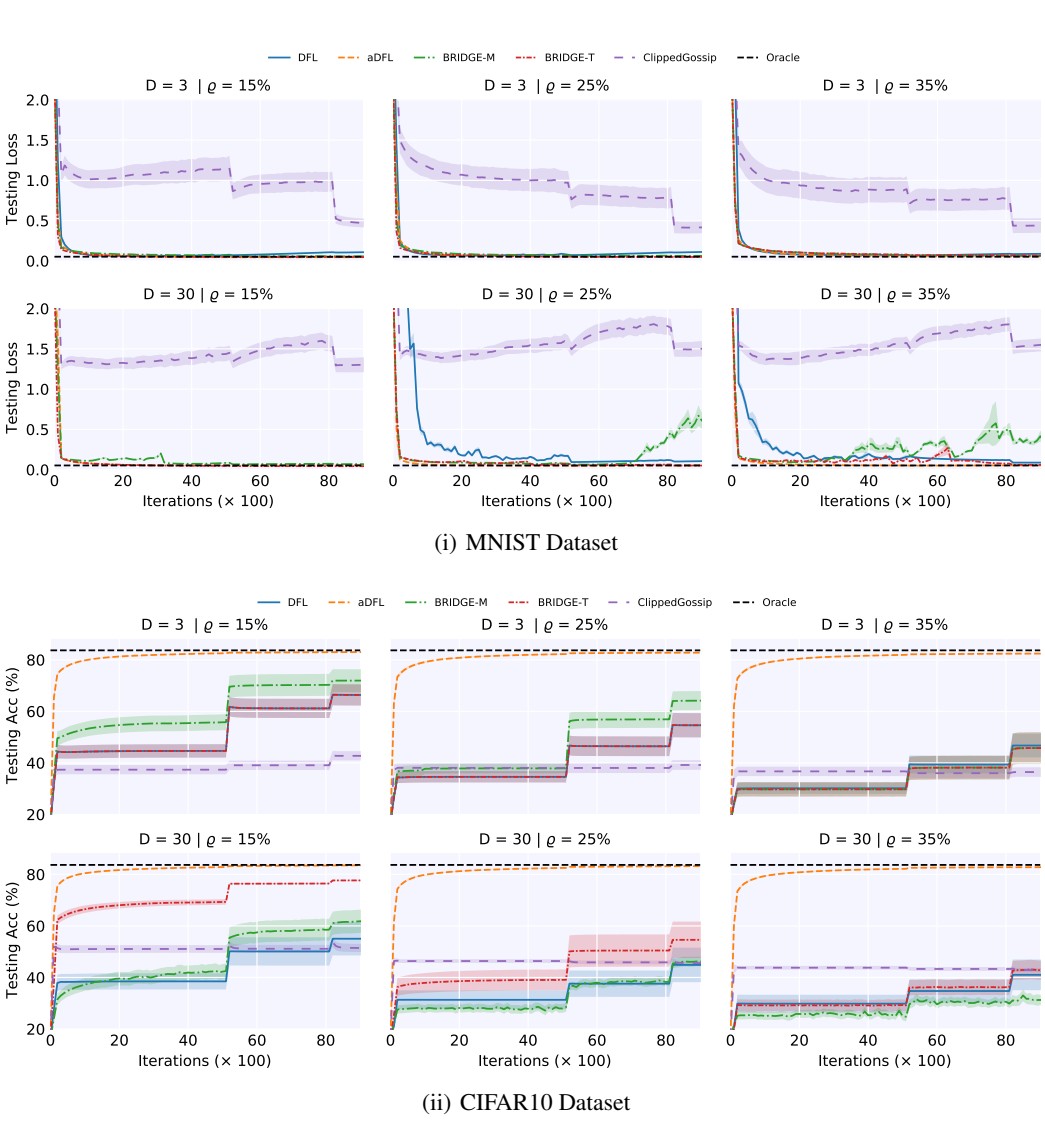

Figure D.6: The testing loss/accuracy over iterations for two datasets in the heterogeneous scenario. Different methods are evaluated with varying in-degrees ($D$) and fractions of abnormal clients ($\varrho$) under the OOD corruption and Directed Circle Network.

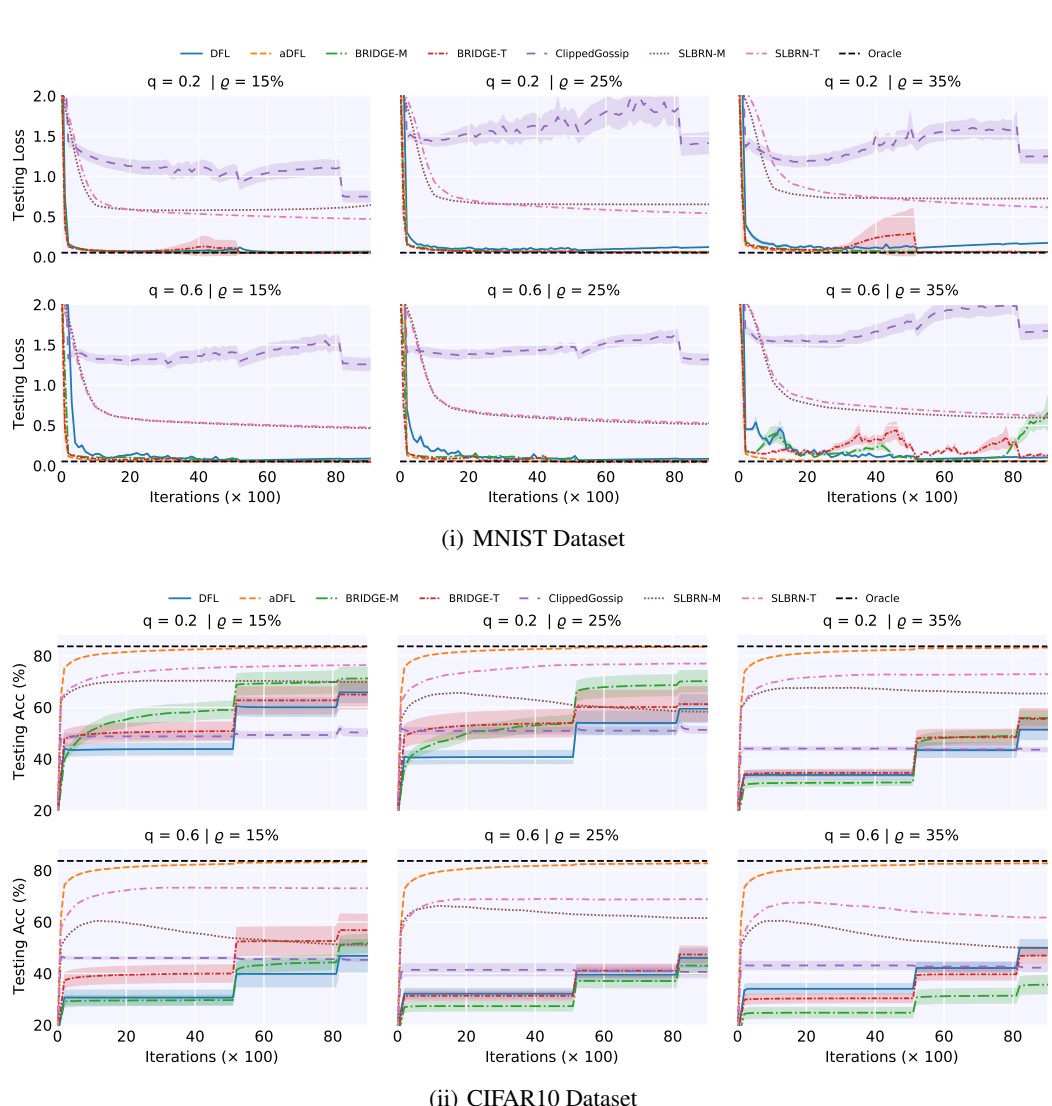

Figure D.7: The testing loss/accuracy over iterations for two datasets in the heterogeneous scenario. Different methods are evaluated with varying link probabilities ($q$) and fractions of abnormal clients ($\varrho$) under the OOD corruption and Erdős–Rényi Graph.

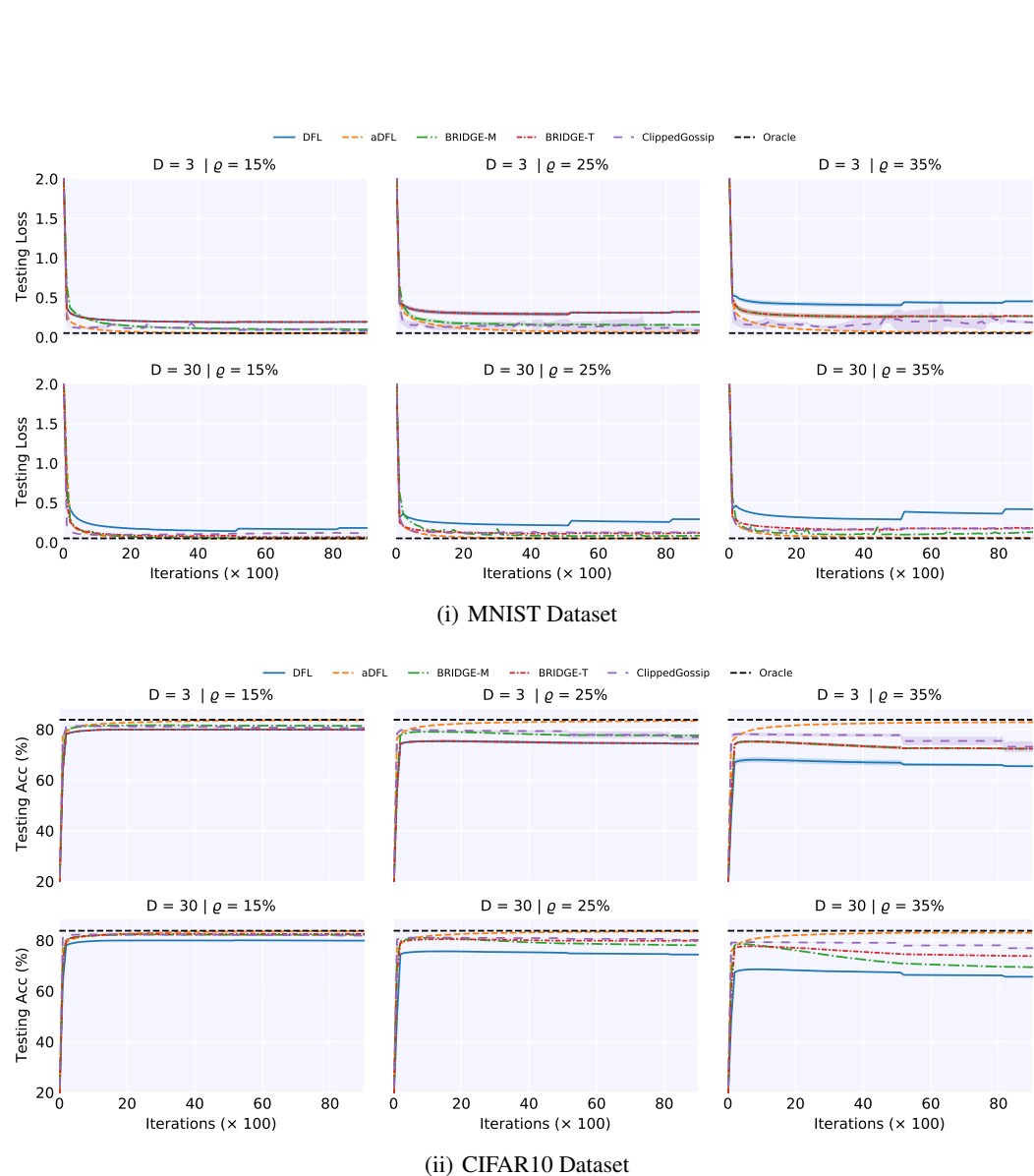

Figure D.8: The testing loss/accuracy over iterations for two datasets in the homogeneous scenario. Different methods are evaluated with varying in-degrees ($D$) and fractions of abnormal clients ($\varrho$) under the LF corruption and Directed Circle Network.

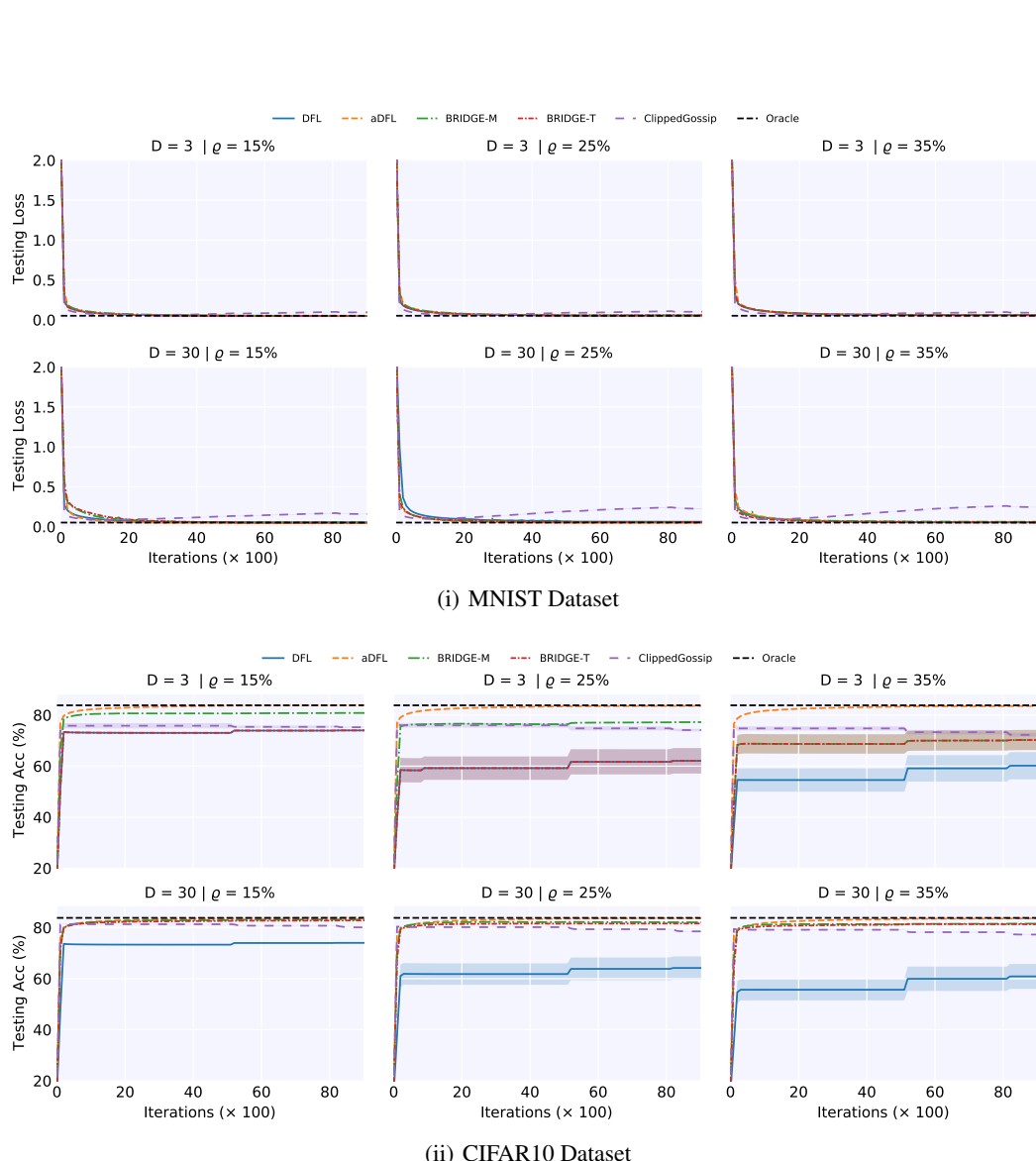

(i) MNIST Dataset

(ii) CIFAR10 Dataset

Figure D.9: The testing loss/accuracy over iterations for two datasets in the homogeneous scenario. Different methods are evaluated with varying in-degrees ($D$) and fractions of abnormal clients ($\varrho$) under the OOD corruption and Directed Circle Network.

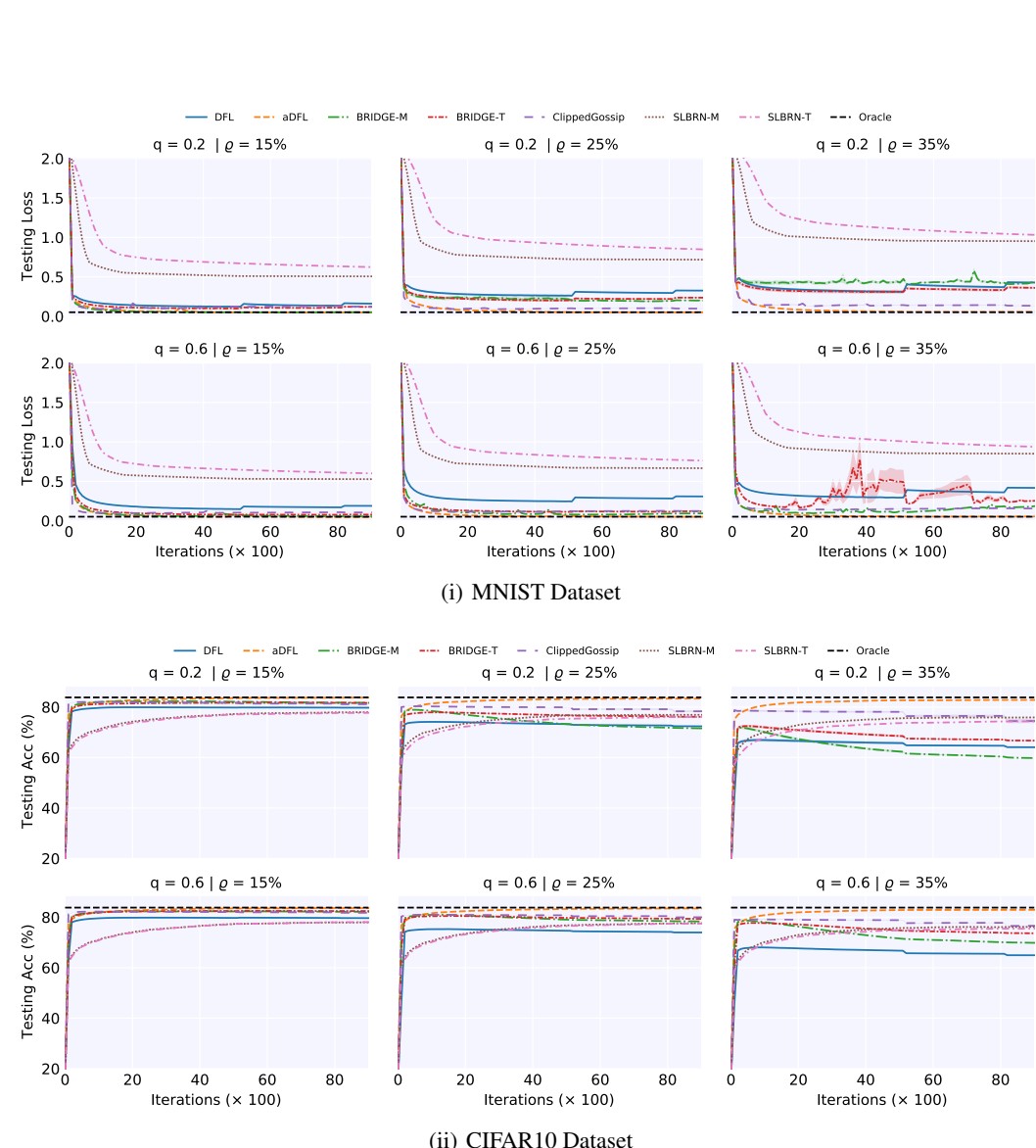

Figure D.10: The testing loss/accuracy over iterations for two datasets in the homogeneous scenario. Different methods are evaluated with varying link probabilities ($q$) and fractions of abnormal clients ($\varrho$) under the LF corruption and Erdős–Rényi Graph.

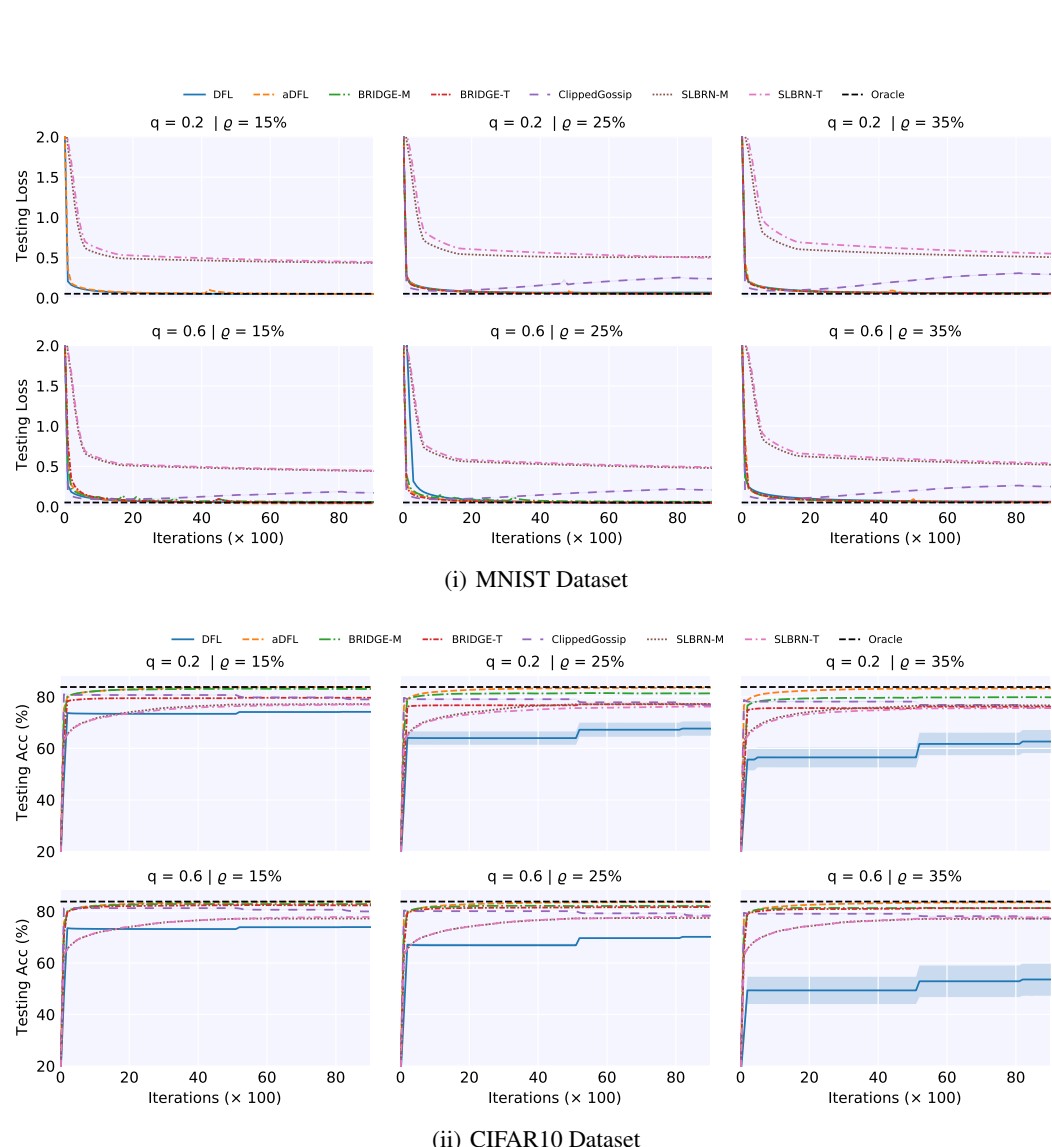

Figure D.11: The testing loss/accuracy over iterations for two datasets in the homogeneous scenario. Different methods are evaluated with varying link probabilities ($q$) and fractions of abnormal clients ($\varrho$) under the OOD corruption and Erdős–Rényi Graph.

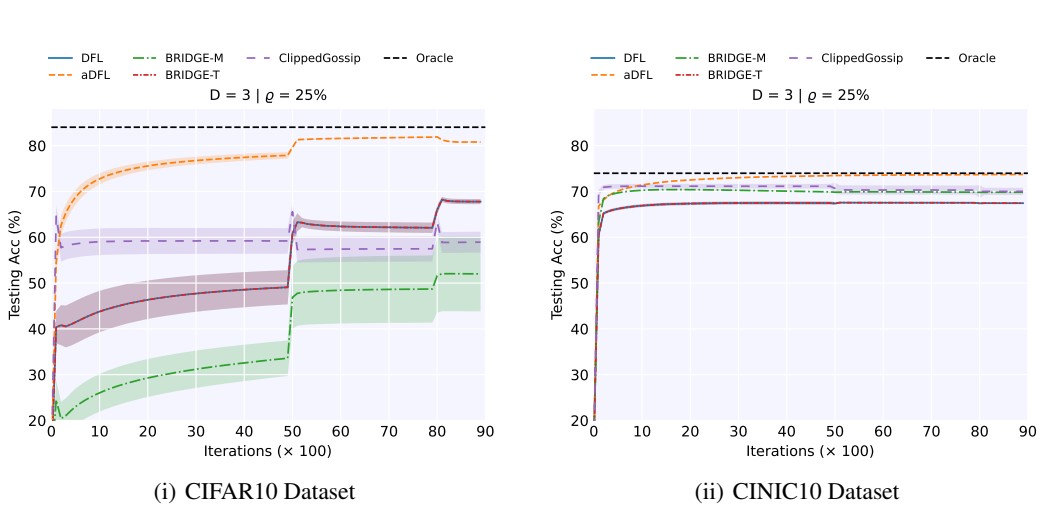

(i) CIFAR10 Dataset          (ii) CINIC10 Dataset

Figure D.12: Testing accuracy over iterations under the increasingly heterogeneous scenario (left panel) and the homogeneous scenario (right panel). We fix the in-degree at $D = 3$ and set the fraction of abnormal clients to $\varrho = 25\%$ under LF corruption on the Directed Circle Network.