# OpenReview forum: "Adaptive Decentralized Federated Learning for Robust Optimization"
_ICLR.cc/2026/Conference — ICLR 2026 Conference Withdrawn Submission_

### Official Review · Reviewer_G32j · 2025-10-25

**Soundness:** 2
**Presentation:** 2
**Contribution:** 2
**Rating:** 2
**Confidence:** 4

**Summary:**

This paper studies fully decentralized federated learning in the presence of abnormal clients. In contrast to the prior work, which requires a sufficiently large number of normal clients or prior knowledge of reliable clients. The proposed method is based on the key idea of adaptively adjusting the learning rate of suspicious and normal clients. Both theoretical and empirical results are provided.

**Strengths:**

* The paper studies an interesting research question where we have abnormal clients with corrupted or low-quality data;
* The intuition is elegant and practical.

**Weaknesses:**

In short, the paper suffers from significant presentation issues, which lead to almost unreadable technical details. Some of the claims in the paper are either incorrect or are not backed by rigorous evidence.
* The flow of the paper can be significantly improved. For example, in the abstract, the authors mention they are free from the stringent conditions on neighboring nodes without discussing what the conditions are or what the prior knowledge is. In lines 55 - 58, the methodology part actually talks about their theoretical contributions.
* The problem definition is vague and unclear. In lines 42 - 46, the authors mention that many prior work requires a sufficiently large number of normal clients. How large is defined as sufficiently large? The statement right after it also remains unclear.
* In the paper, the authors try to address the problem where clients' local data is either of low quality or corrupted. However, the reviewer is quite confused by its connection with Byzantine clients in line 33. The Byzantine problem is that the adversaries might intentionally and unstructuredly modify the communication message to poison the training. If the data is faithfully collected, and we assume the learning process is correct, as stated by the authors in lines 125 - 126. Then, it has nothing to do with Byzantine clients.
* In Section 3.1, the introductions of notations significantly diminish the readability of the paper. For example, it remains unclear what sample set is. Does it mean the set of clients? Does it mean the set of data samples collected by a given client? In addition, many important notations are overlooked in this part. For example, I believe the authors use $\dot{\ell}$ to represent the gradient, e.g., in Eq. (3.1). Yet, it is never mentioned in the notation section, and it is not standard. It is recommended to use $\nabla \ell$ for gradients and $\nabla^2 \ell$ for Hessian.  Furthermore, the choices of notations are, by design, confusing. For instance, $\tilde \theta^{(t,m)}$ in Eq. (3.1) is the model parameter at client $m$ in round $t$, which unifies two characteristics of one client to one dimension. It is recommended to use superscript and subscript to denote them separately. Many notations are also used without specification. For example, in line 156, the authors use $\mathbf{1}_M$ without defining it.
* Lines 123 - 133 describe the threat model in this paper, which is, by definition, vague and confusing.
  * First, the cited papers are about adversarial or Byzantine training, which I have made clear in a previous weakness that is irrelevant to the scope of the paper because it assumes clients faithfully follow the learning protocols.
  * Second, the notation is confusing: what does $\rightarrow_d$ mean? What is $d$? If $\theta_0$ is asymptotically close to each client's local minimizer $\hat{\theta}_m$, isn't the clients ' local data homogeneous? This is in conflict with the statement in lines 114 - 115, which assumes the clients' data is heterogeneous.
  * Third, if under lines 129 - 132, we assume clients' local data is homogeneous. The robust aggregation rule is a potential solution to the problem, which questions the novelty and contributions of the work.
* Assumption 3 requires local strong convexity, which is stronger than the often-used global strong convexity in the prior work. Please clarify the statements in lines 187 - 189.
* Assumption 6 requires clarification. When we have homogeneous clients and when the global model reaches a stationary point, it is natural to obtain zero-valued gradients for all clients in expectation.
* Since the foundations (assumptions) of the paper are unclear, it is hard to trust the analysis. For example, what is convergence with probability ***tending to 1*** in line 264?

**Questions:**

* Can the authors give the paper another round of proofreading to improve the flow of the paper?
* Can the authors clarify the problem definitions?
* Can the authors use standard notations?
* Can the authors clean up the notations to minimize overwhelm?

---

### Official Review · Reviewer_mDKp · 2025-10-30

**Soundness:** 2
**Presentation:** 2
**Contribution:** 2
**Rating:** 2
**Confidence:** 5

**Summary:**

To defend against data poisoning attacks in the decentralized federated learning setting, the authors propose a novel algorithm that adaptively assigns large weights to trustworthy clients and small weights to abnormal clients during iterations. The authors prove that the proposed algorithm converges to the oracle estimator  as the total sample size approaches infinity.

**Strengths:**

1. The investigated problem of robust decentralized federated learning is both important and underexplored in this field.

**Weaknesses:**

1. The idea of assigning trustworthiness weights to clients based on their gradient norms is questionable. It is unclear why trustworthy clients would consistently have small gradient norms while abnormal clients would have large ones, as this does not hold throughout the training process. For instance, during the early stages of training, all clients typically exhibit large gradient norms. This represents a significant weakness of the paper, and the reviewer is not convinced by this approach.

2. The assumptions made in this paper are somewhat strong. For instance, the assumption in line 129 requires all trustworthy clients to share the same optimal solution to their local cost functions, which is difficult to justify in practice. Additionally, Assumption 5 is not commonly used in the analysis of decentralized federated learning algorithms. Could the authors provide examples or further justification to demonstrate that this assumption holds in practical scenarios?

3. The baseline is limited. The authors should compare the proposed algorithm with other state-of-the-art robust decentralized federated learning algorithms, such as IOS in Wu et al. (2023b).

4. The writing of this paper requires further refinement. The excessive use of notations, unclear presentation, and several typographical errors make it challenging to read and follow. The authors are encouraged to streamline the notation, improve the clarity of the presentation, and thoroughly proofread the paper to eliminate any potential typos.

**Questions:**

My detailed questions are outlined in the section above; please refer to them.

---

### Official Review · Reviewer_mcBF · 2025-10-31

**Soundness:** 3
**Presentation:** 3
**Contribution:** 2
**Rating:** 4
**Confidence:** 3

**Summary:**

This paper proposes a method called adaptive Decentralized Federated Learning (aDFL), aimed at enhancing robustness in decentralized federated learning environments with abnormal clients. The core idea is to dynamically adjust each client’s learning rate according to measures of their trustworthiness, specifically using a gradient-norm-based adaptive weighting mechanism. The method is accompanied by rigorous theoretical analysis, including convergence and statistical efficiency results, and validated through extensive experiments on both synthetic and real datasets.

**Strengths:**

1. The paper introduces a flexible and practical approach to robustness in decentralized federated learning by leveraging adaptive, gradient-norm-based client weighting.
2. Theoretical analyses are compelling, providing convergence behavior, oracle property, and clear conditions for robustness.
3. Experimental evaluation is thorough, covering various synthetic corruptions (e.g., bit-flipping), network structures (e.g., Directed Circle), and practical tasks.

**Weaknesses:**

1. The process of learning rate adaptation centers on the mapping $\pi(x) = \exp(-x)$ and the tuning parameter $\lambda_n$ (Equation 4.4). However, there is insufficient elaboration in the main paper on how $\lambda_n$ is chosen, beyond a brief mention of cross-validation and a theoretical interval ($\log N \lesssim \lambda_n \lesssim \sqrt{n} M^{-1/8}$).
2. The paper does not provide an explicit ablation study isolating the impact of the initial estimator’s quality. How do other decentralized robust estimators mentioned in Section 4 affect performance?
3. Typo error: “reqruies” should be corrected to “requires” in the conclusion.

**Questions:**

1. Could the authors provide more details on how the tuning parameter $\lambda_n$ (Equation 4.4) is selected in practice? What empirical strategy or heuristic guided the choice of $\lambda_n$ during experiments?
2. Could the authors describe the impact of the initial estimator’s quality on overall performance? In particular, how would the proposed method compare when using other decentralized robust estimators mentioned in Section 4?
3. In addition, could the authors compare their approach against Sun et al. (2024) — “Byzantine-robust decentralized federated learning via dual-domain clustering and trust bootstrapping” — to better position the proposed adaptive mechanism?

---

### Official Review · Reviewer_Ckec · 2025-11-08

**Soundness:** 3
**Presentation:** 3
**Contribution:** 3
**Rating:** 8
**Confidence:** 4

**Summary:**

This paper proposes a new adaptive decentralized learning algorithm that does not enforce constraints on the number of abnormal neighboring clients in the network. This is achieved by an adaptive learning rate that is related with client's gradient with the assumption that abnormal client would have large gradients, thereby should have smaller learning rate so the learning process would not be corrupted. The proposed work is novel and effective when compared with statistical based methods that involve mean, trimmed mean, etc. The algorithms effectiveness is also supported by rigorous theoretical analysis.

**Strengths:**

1. The proposed algorithm releases the constraints on the neighboring clients' numbers in a decentralized network, making it more flexible in real applications where the adversary is caused by data-contamination. The mechanism is also easy to implement.
2. Extensive experiments have been conducted to show the effectiveness of the algorithm. Both regression and classification tasks are simulated on various network topologies.
3. The algorithm's effective is supported by theoretical analysis.
4. The presentation of the algorithm is in general good with some minor modifications needed in reference.

**Weaknesses:**

1. The literature review and experimental comparison with state-of-the-art neglects one recent work that also does not require constraints on the neighboring clients.
[1] Zhang, K., Basharat, A. and Xu, P., 2024, December. Byzantine-robust decentralized federated learning via local performance checking. In International Conference on Neural Information Processing (pp. 171-185). Singapore: Springer Nature Singapore.
2. Only the data-contaminated adversary setting is consider in this work, which is limited. It would be better to consider other abnormal phenomenon.
3. [minor comment] References are not in consistent format, some using full names and some use abbreviated names of the authors.

**Questions:**

1. In algorithm 1, Line 4, the neighborhood-averaged estimator already combines abnormal clients' parameters, how much would that affect the algorithm's performance, experimentally and theoretically?
2. Why is the fraction of abnormal clients $\varrho$ set to be [0, 1/2)? Could it be larger? Did you test the performance of your algorithm with $\varrho=0.5$ or close to 0.5?

---

### Note · Authors · 2025-12-02

I have read and agree with the venue's withdrawal policy on behalf of myself and my co-authors.